



# Nitrate radicals and biogenic volatile organic compounds: oxidation, mechanisms and organic aerosol

N. L. Ng[1,2], S. S. Brown[3,4], A. T. Archibald[5], E. Atlas[6], R. C. Cohen[7], J. N. Crowley[8], D. A. Day[9,4], N. M. Donahue[10], J. L. Fry[11], H. Fuchs[12], R. J. Griffin[13], M. I. Guzman[14], H. Herrmann[15], A. Hodzic[16], Y. Iinuma[15], J. L. Jimenez[9,4], A. Kiendler-Scharr[12], B. H. Lee[17], D. J. Luecken[18], J. Mao[19,20], R. McLaren[21], A. Mutzel[15], H. D. Osthoff[22], B. Ouyang[23], B. Picquet-Varrault[24], U. Platt[25], H. O. T. Pye[18], Y. Rudich[26], R. H. Schwantes[27], M. Shiraiwa[28], J. Stutz[29], J. A. Thornton[17], A. Tilgner[15], B. J. Williams[30], R. A. Zaveri[31]

[1]School of Chemical and Biomolecular Engineering, Georgia Institute of Technology, Atlanta, GA, USA

[2]School of Earth and Atmospheric Sciences, Georgia Institute of Technology, Atlanta, GA, USA

[3]NOAA Earth System Research Laboratory, Chemical Sciences Division, Boulder, CO, USA

[4]Department of Chemistry and Biochemistry, University of Colorado, Boulder, CO, USA

[5]National Centre for Atmospheric Science, University of Cambridge, Cambridge, UK

[6]Department of Atmospheric Sciences, RSMAS, University of Miami, Miami, FL, USA

[7]Department of Chemistry, University of California at Berkeley, Berkeley, CA, USA

[8]Max-Planck-Institut für Chemie, Division of Atmospheric Chemistry, Mainz, Germany

[9]Cooperative Institute for Research in Environmental Sciences, University of Colorado, Boulder, CO, USA

[10]Center for Atmospheric Particle Studies, Carnegie-Mellon University, Pittsburgh, PA, USA

[11]Department of Chemistry, Reed College, Portland, OR, USA

[12]Institut für Energie und Klimaforschung: Troposphäre (IEK-8), Forschungszentrum Jülich, Jülich, Germany

[13]Department of Civil and Environmental Engineering, Rice University, Houston, TX, USA

[14]Department of Chemistry, University of Kentucky, Lexington, KY, USA

[15]Atmospheric Chemistry Department, Leibniz Institute for Tropospheric Research, Leipzig, Germany

[16]Atmospheric Chemistry Observations and Modeling, National Center for Atmospheric Research, Boulder, CO, USA

[17]Department of Atmospheric Sciences, University of Washington, Seattle, WA, USA





[18]National Exposure Research Laboratory, U.S. Environmental Protection Agency, Research Triangle Park, North Carolina 27711, United States

[19]Program in Atmospheric and Oceanic Sciences, Princeton University, Princeton, New Jersey, USA.

5    [20]Geophysical Fluid Dynamics Laboratory/National Oceanic and Atmospheric Administration, Princeton, New Jersey, USA.

[21]Centre for Atmospheric Chemistry, York University, Toronto, ON, Canada

[22]Department of Chemistry, University of Calgary, Calgary, Alberta, Canada.

[23]Department of Chemistry, University of Cambridge, Cambridge, UK.

10    [24] Laboratoire Interuniversitaire des Systemes Atmospheriques (LISA), CNRS, Universities of Paris-Est Créteil and ́ Paris Diderot, Institut Pierre Simon Laplace (IPSL),  Créteil, France

[25]Institute of Environmental Physics, University of Heidelberg, Heidelberg, Germany

[26]Department of Earth and Planetary Sciences, Weizmann Institute, Rehovot, Israel

[27]Division of Geological and Planetary Sciences, California Institute of Technology, Pasadena, 15    CA, USA

[28]Department of Chemistry, University of California Irvine, CA, USA

[29]Department of Atmospheric and Oceanic Sciences, University of California, Los Angeles, CA, USA

[30]Department of Energy, Environmental and Chemical Engineering, Washington University in St. 20    Louis, St. Louis, MO, USA

[31]Atmospheric Sciences and Global Change Division, Pacific Northwest National Laboratory, Richland,WA, USA

*Correspondence to:* N. L. Ng (ng@chbe.gatech.edu), S. S. Brown (steven.s.brown@noaa.gov)





**Abstract.** Oxidation of biogenic volatile organic compounds (BVOC) by the nitrate radical ($NO_3$) represents one of the important interactions between anthropogenic emissions related to combustion and natural emissions from the biosphere. This interaction has been recognized for more than three decades, during which time a large body of research has emerged from laboratory, field and modeling studies. $NO_3$-BVOC reactions influence air quality, climate and visibility through regional and global budgets for reactive nitrogen (particularly organic nitrates), ozone and organic aerosol. Despite its long history of research and the significance of this topic in atmospheric chemistry, a number of important uncertainties remain. These include an incomplete understanding of the rates, mechanisms and organic aerosol yields for $NO_3$-BVOC reactions, lack of constraints on the role of heterogeneous oxidative processes associated with the $NO_3$ radical, the difficulty of characterizing the spatial distributions of BVOC and $NO_3$ within the poorly mixed nocturnal atmosphere and the challenge of constructing appropriate boundary layer schemes and non-photochemical mechanisms for use in state-of-the-art chemical transport and chemistry-climate models.

This review is the result of a workshop of the same title held at the Georgia Institute of Technology in June 2015. The first section summarizes the current literature on $NO_3$-BVOC chemistry, with a particular focus on recent advances in instrumentation and models, and in organic nitrate and secondary organic aerosol (SOA) formation chemistry. Building on this current understanding, the second half of the review outlines impacts of $NO_3$-BVOC chemistry on air quality and climate, and suggests critical research needs to better constrain this interaction to improve the predictive capabilities of atmospheric models.

## 1 Introduction

The emission of hydrocarbons from the terrestrial biosphere represents an enormous natural input of chemically-reactive compounds to Earth's atmosphere (Guenther et al., 1995;Goldstein and Galbally, 2007). Understanding the atmospheric degradation of these species is a critical area of current research that influences models of oxidants and aerosols on regional and global scales. Nitrogen oxides ($NO_x = NO + NO_2$) arising from combustion and microbial action on fertilizer are one of the major anthropogenic inputs that perturb the chemistry of the atmosphere (Crutzen, 1973). Nitrogen oxides have long been understood to influence oxidation cycles of biogenic





volatile organic compounds (BVOC), especially through photochemical reactions of organic and hydroperoxy radical intermediates ($RO_2$ and $HO_2$) with nitric oxide (NO) (Chameides, 1978).

The nitrate radical ($NO_3$) arises from the oxidation of nitrogen dioxide ($NO_2$) by ozone ($O_3$) and occurs principally in the nighttime atmosphere due to its rapid photolysis in sunlight and its

reaction with NO (Wayne et al., 1991;Brown and Stutz, 2012). The nitrate radical is a strong oxidant, reacting with a wide variety of volatile organic compounds, including alkanes, aromatics, and oxygenates as well as with reduced sulfur compounds. Reactions of $NO_3$ are particularly rapid with unsaturated compounds (alkenes) (Atkinson and Arey, 2003). BVOC such as isoprene, monoterpenes and sesquiterpenes typically have one or more unsaturated functionalities such that

they are particularly susceptible to oxidation by $O_3$ and $NO_3$.

The potential for $NO_3$ to serve as a large sink for BVOC was recognized more than three decades ago (Winer et al., 1984). Field studies since that time have shown that in any environment with moderate to large BVOC concentrations, a majority of the $NO_3$ radical oxidative reactions are with BVOC rather than VOC of anthropogenic origin (Brown and Stutz, 2012). This interaction

gives rise to a mechanism that couples anthropogenic $NO_x$ emissions with natural BVOC emissions (Fry et al., 2009;Xu et al., 2015a). Although it is one of several such anthropogenic – biogenic interactions (Hoyle et al., 2011), reactions of $NO_3$ with BVOC is an area of intense current interest and one whose study has proven challenging. These challenges arise from the more limited current database of laboratory data for $NO_3$ oxidation reactions relative to those of

other common atmospheric oxidants such as hydroxyl radical (OH) and $O_3$. The mixing state of the nighttime atmosphere and the limitations it imposes for characterization of nocturnal oxidation chemistry during field measurements and within atmospheric models present a second challenge to this field of research.

Reactions of $NO_3$ with BVOC have received increased attention in the recent literature as a

potential source of secondary organic aerosol (SOA) (Pye et al., 2010;Fry et al., 2014;Boyd et al., 2015). This SOA source is intriguing for several reasons. First, although organics are now understood to comprise a large fraction of total aerosol mass, and although much of these organics are secondary, sources of SOA remain difficult to characterize in part due to a large number of emission sources and potential chemical mechanisms (Zhang et al., 2007;Hallquist et

al., 2009;Jimenez et al., 2009;Ng et al., 2010). Analysis of aerosol organic carbon shows that a large fraction is modern, arising either from biogenic hydrocarbon emissions or biomass burning sources (e.g. (Schichtel et al., 2008;Hodzic et al., 2010)). Conversely, field data in regionally polluted areas indicate strong correlations between tracers of anthropogenic emissions and SOA,



which suggests that anthropogenic influences lead to production of SOA from modern (i.e., non-fossil) carbon (e.g. (Weber et al., 2007)). Model studies confirm that global observations are best simulated with a biogenic carbon source in the presence of anthropogenic pollutants (Spracklen et al., 2011). Reactions of $NO_3$ with BVOC are one such mechanism that may lead to

anthropogenically influenced biogenic SOA (Hoyle et al., 2007), and it is important to quantify the extent to which such reactions can explain sources of SOA.

Second, some laboratory and chamber studies suggest that SOA yields from $NO_3$ oxidation of common BVOC such as isoprene and selected monoterpenes are greater than that for OH or $O_3$ oxidation (Hallquist et al., 1997b;Griffin et al., 1999;Spittler et al., 2006;Ng et al., 2008;Fry et al.,

2009;Rollins et al., 2009;Fry et al., 2011;Fry et al., 2014;Boyd et al., 2015). However, among the monoterpenes, the SOA yields may be much more variable for $NO_3$ oxidation than for other oxidants, with anomalously low SOA yields in some cases and high SOA yields in others (Draper et al., 2015;Nah et al., 2016b).

Third, not only is $NO_3$-BVOC chemistry a potentially efficient SOA formation mechanism, it is

also a major pathway for the production of organic nitrates (von Kuhlmann et al., 2004;Horowitz et al., 2007), a large component of oxidized reactive nitrogen that may serve as either a $NO_x$ reservoir or $NO_x$ sink. Results from recent field measurements have shown that organic nitrates are important components of ambient OA (Day et al., 2010;Rollins et al., 2012;Fry et al., 2013;Ayres et al., 2015;Xu et al., 2015a;Xu et al., 2015b;Kiendler-Scharr et al., 2016;Lee et al.,

2016). Furthermore, within the last several years, the capability to measure both total and speciated gas-phase and particle-phase organic nitrates has been demonstrated (Fry et al., 2009;Rollins et al., 2010;Fry et al., 2013;Rollins et al., 2013;Fry et al., 2014;Lee et al., 2016;Nah et al., 2016b). The lifetimes of BVOC-$NO_3$-derived organic nitrates with respect to hydrolysis, photooxidation, and deposition play an important role in the $NO_x$ budget and formation of $O_3$ and

SOA. These processes appear to depend strongly on the parent VOCs and oxidation conditions and must be better constrained for understanding organic nitrate lifetimes in the atmosphere (Darer et al., 2011;Hu et al., 2011;Liu et al., 2012b;Boyd et al., 2015;Pye et al., 2015;Rindelaub et al., 2015;Lee et al., 2016;Nah et al., 2016b).

Fourth, incorporation of SOA yields for $NO_3$-BVOC reactions into regional and global models

indicates that these reactions could be a significant, or in some regions even dominant, SOA contributor (Hoyle et al., 2007;Pye et al., 2010;Chung et al., 2012;Fry and Sackinger, 2012;Pye et al., 2015;Kiendler-Scharr et al., 2016). Model predictions of organic aerosol formation from $NO_3$-BVOC until recently have been difficult to verify directly from field measurements. Recent



progress in laboratory and field studies have provided some of the first opportunities to develop coupled gas and particle systems to describe mechanistically and predict SOA and organic nitrate formation from $NO_3$-BVOC reactions (Pye et al., 2015).

Finally, analyses from several recent field studies examining diurnal variation in the organic and/or nitrate content of aerosols conclude that nighttime BVOC oxidation through $NO_3$ radicals constitutes a large organic aerosol source (Rollins et al., 2012;Fry et al., 2013;Xu et al., 2015a;Xu et al., 2015b;Kiendler-Scharr et al., 2016). Although such analyses may correct their estimates of aerosol production for the variation in boundary layer depth, field measurements at surface level are necessarily limited in their ability to assess accurately the atmospheric chemistry in the overlying residual layer, or even the gradients that may exist within the relatively shallow nocturnal boundary layer (Stutz et al., 2004;Brown et al., 2007b). Thus, although there is apparent consistency between recent results from both modeling and field studies, the vertically stratified structure of the nighttime atmosphere makes such comparisons difficult to evaluate critically. There is a limited database of nighttime aircraft measurements that has probed this vertical structure with sufficient chemical detail to assess $NO_3$-BVOC reactions (Brown et al., 2007a;Brown et al., 2009), and some of these data show evidence for an OA source related to this chemistry, especially at low altitude (Brown et al., 2013). A larger database of aircraft and/or vertically resolved measurements is required, however, for comprehensive comparisons to model predictions.

The purpose of this article is to review the current literature on the chemistry of $NO_3$ and BVOC to assess critically the current state of the science. The review focuses on BVOC emitted from terrestrial vegetation. The importance of $NO_3$ reactions with reduced sulfur compounds such as dimethyl sulfide in marine ecosystems is well known (Platt et al., 1990;Yvon et al., 1996;Allan et al., 1999;Allan et al., 2000;Vrekoussis et al., 2004;Stark et al., 2007;Osthoff et al., 2009) but outside of the scope of this review. Key uncertainties include chemical mechanisms, yields of major reaction products such as SOA and organic nitrogen, the potential for $NO_3$ and BVOC to interact in the ambient atmosphere and the implications of that interaction for current understanding of air quality and climate. The review stems from an International Global Atmospheric Chemistry (IGAC) and U.S. National Science Foundation (NSF) sponsored workshop of the same name held in June 2015 at the Georgia Institute of Technology, Atlanta, GA, USA. Following this introduction, section 2 of this article reviews the current literature in several areas relevant to the understanding of $NO_3$-BVOC atmospheric chemistry. Section 3 outlines perspectives on the implications of this chemistry for understanding climate and air





quality, its response to current emission trends and its relevance to implementation of control strategies. Finally, the review concludes with an assessment of the impacts of $NO_3$-BVOC reactions on air quality, visibility and climate.

## 2 Review of current literature

This section contains a literature review of the current state of knowledge of $NO_3$-BVOC chemistry with respect to 1) reaction rate constants and mechanisms from laboratory and chamber studies; 2) secondary organic aerosol yields, speciation and particle-phase chemistry; 3) heterogeneous reactions of both $NO_3$ and $N_2O_5$ and their implications for $NO_3$-BVOC chemistry; 4) instrumental methods for analysis of reactive nitrogen compounds, including $NO_3$, organic

nitrates and nitrogen containing particulate matter; 5) field observations relevant to the understanding of $NO_3$ and BVOC; and 6) models of $NO_3$-BVOC chemistry.

### 2.1 $NO_3$-BVOC reaction rate constants and chemical mechanisms

### 2.1.1 Reaction rate constants

Among the numerous BVOC emitted into the troposphere, kinetic data for $NO_3$-oxidation have

been provided for more than 40 compounds. The most emitted/important BVOC have been subjected to several kinetic studies, using both absolute and relative methods, which provides quite precise recommended rate coefficients by IUPAC (Table 1). This is the case for isoprene, α-pinene, β-pinene and 2-methyl-3-buten-2-ol (MBO). However, for isoprene, β-pinene and MBO, rate coefficients obtained by different studies range over a factor of two. For some other terpenes,

only few kinetic studies have been carried out, with at least one absolute rate determination. This is the case for sabinene, 2-carene, camphene, d-limonene, α-phellandrene, myrcene, γ-terpinene, terpinolene. For these compounds, experimental data agree within 30-40%, except α-phellandrene and terpinolene for which discrepancies are larger. For other BVOC, including other terpenes, sesquiterpenes and oxygenated species, rate constants are mostly based on a single determination

and highly uncertain. For these compounds, further rate constant determinations and end-product measurements are essential to better evaluate the role of $NO_3$ in their degradation. The ability to predict the $NO_3$-BVOC rate constants using Structure-Activity Relationships (SAR) has been improved. A recent study (Kerdouci et al., 2010;Kerdouci et al., 2014) presented a new SAR



parameterization based on 180 $NO_3$-VOCs reactions. The method is capable of predicting 90% of the rate constants within a factor of two.

**Table 1** Reaction rate constants of $NO_3$+BVOC.

| Compound | Chemical structure | Rate coefficient data [a] | |
|---|---|---|---|
| | | $k(NO_3+BVOC)$ cm$^3$ molecule$^{-1}$ s$^{-1}$ | Technique/Reference |
| Isoprene |  | | RR/ (Atkinson et al., 1984) |
| | | | RR/ (Barnes et al., 1990) |
| | | $(5.94 \pm 0.16) \times 10^{-13}$ | RR/ (Berndt and Boge, 1997a) |
| | | $(1.21 \pm 0.20) \times 10^{-13}$ | DF-MS/ (Benter and Schindler, |
| | | $(6.86 \pm 0.55) \times 10^{-13}$ | 1988) |
| | | $(1.30 \pm 0.14) \times 10^{-12}$ | F-LIF/ (Dlugokencky and |
| | | $3.03 \times 10^{-12} \exp[-(450 \pm 70)/T]$ | Howard, 1989) |
| | | $(6.52 \pm 0.78) \times 10^{-13}$ | F-LIF/ (Dlugokencky and |
| | | $(7.30 \pm 0.44) \times 10^{-13}$ | Howard, 1989)DF-MS/ (Wille et |
| | | $(8.26 \pm 0.60) \times 10^{-13}$ | al., 1991) |
| | | $(1.07 \pm 0.20) \times 10^{-12}$ | DF-MS/ (Wille et al., 1991) |
| | | $(7.3 \pm 0.2) \times 10^{-13}$ | PR-A/ (Ellermann et al., 1992) |
| | | $(6.24 \pm 0.11) \times 10^{-13}$ | F-CIMS/ (Suh et al., 2001) |
| | | **$6.5 \times 10^{-13}$** | RR/ (Zhao et al., 2011a) |
| | | | **IUPAC** |
| α-pinene | | $1.19 \times 10^{-12} \exp[(490 \pm 70)/T]$ | F-LIF/ (Dlugokencky and |
| | | $(6.18 \pm 0.94) \times 10^{-12}$ | Howard, 1989) |
| | | $3.5 \times 10^{-13} \exp[(841 \pm 144)/T]$ | F-LIF/ (Dlugokencky and |
| | | $(5.9 \pm 0.8) \times 10^{-12}$ | Howard, 1989) |
| | | $(5.82 \pm 0.16) \times 10^{-12}$ | DF-LIF/ (Martinez et al., 1998) |
| | | $(6.56 \pm 0.94) \times 10^{-12}$ | DF-LIF/ (Martinez et al., 1998) |
| | | $(5.82 \pm ) \times 10^{-12}$ | RR/ (Atkinson et al., 1984) |
| | | $(4.88 \pm 0.56) \times 10^{-12}$ | RR/ (Barnes et al., 1990) |
| | | **$6.2 \times 10^{-12}$** | RR/ (Kind et al., 1998) |





RR/ (Stewart et al., 2013)

**IUPAC**

| | | | |
|---|---|---|---|
| β-pinene |  | $(2.1 \pm 0.4) \times 10^{-12}$ | DF-LIF/ (Martinez et al., 1998) |
| | | $1.6 \times 10^{-10} \exp[(-1248 \pm 36)/T]$ | DF-LIF/ (Martinez et al., 1998) |
| | | $(2.36 \pm 0.10) \times 10^{-12}$ | RR/ (Atkinson et al., 1984) |
| | | $(2.38 \pm 0.05) \times 10^{-12}$ | RR/ (Atkinson et al., 1988) |
| | | $(1.1 \pm 0.4) \times 10^{-12}$ | RR/ (Kotzias et al., 1989) |
| | | $(2.81 \pm 0.47) \times 10^{-12}$ | RR/ (Barnes et al., 1990) |
| | | $(2.81 \pm 0.56) \times 10^{-12}$ | RR/ (Kind et al., 1998) |
| | | **$2.5 \times 10^{-12}$** | **IUPAC** |
| Sabinene | | $(1.07 \pm 0.16) \times 10^{-11}$ | DF-LIF/ (Martínez et al., 1999) |
| | | $2.03 \times 10^{-10} \exp[(-940 \pm 200)/T]$ | DF-LIF/ (Martínez et al., 1999) |
| | | $(1.01 \pm 0.03) \times 10^{-11}$ | RR/ (Atkinson et al., 1990) |
| | | $(6.8 \pm 0.12) \times 10^{-12}$ | RR/ (Atkinson et al., 1990) |
| | | **$1.0 \times 10^{-11}$** | **IUPAC** |
| Camphene | | $3.1 \times 10^{-12} \exp[(-481 \pm 55)/T]$ | DF-LIF/ (Martinez et al., 1998) |
| | | $(6.2 \pm 2.1) \times 10^{-13}$ | DF-LIF/ (Martinez et al., 1998) |
| | | $(6.54 \pm 0.16) \times 10^{-13}$ | RR/ (Atkinson et al., 1990) |
| | | | DF-LIF/ (Martínez et al., 1999) |
| 2-carene | | $(1.66 \pm 0.18) \times 10^{-11}$ | DF-LIF/ (Martínez et al., 1999) |
| | | $1.4 \times 10^{-12} \exp[(741 \pm 190)/T]$ | RR/ (Corchnoy and Atkinson, 1990) |
| | | $(1.87 \pm 0.11) \times 10^{-11}$ | RR/ (Corchnoy and Atkinson, 1990) |
| | | $(2.16 \pm 0.36) \times 10^{-11}$ | |
| 3-carene | | $(1.01 \pm 0.02) \times 10^{-11}$ | RR/ (Atkinson et al., 1984) |
| | | $(8.2 \pm 1.2) \times 10^{-12}$ | RR/ (Barnes et al., 1990) |
| | | **$9.1 \times 10^{-12}$** | **IUPAC** |



| | | | |
|---|---|---|---|
| d-limonene | | $(9.4 \pm 0.9) \times 10^{-12}$ | DF-LIF/ (Martínez et al., 1999) |
| | | $(1.31 \pm 0.04) \times 10^{-11}$ | RR/ (Atkinson et al., 1984) |
| | | $(1.12 \pm 0.17) \times 10^{-11}$ | RR/ (Barnes et al., 1990) |
| | | **$1.2 \times 10^{-11}$** | **IUPAC** |
| α-phellandrene | | $(4.2 \pm 1.0) \times 10^{-11}$ | DF-LIF/ (Martínez et al., 1999) |
| | | $1.9 \times 10^{-19} \exp[-(1158 \pm 270)/T]$ | DF-LIF/ (Martínez et al., 1999) |
| | | $(8.52 \pm 0.63) \times 10^{-11}$ | RR/ (Atkinson et al., 1985) |
| | | $(5.98 \pm 0.20) \times 10^{-11}$ | RR/ (Berndt et al., 1996) |
| | | **$7.3 \times 10^{-11}$** | **IUPAC** |
| β-phellandrene | | $(7.96 \pm 2.82) \times 10^{-12}$ | RR/ (Shorees et al., 1991) |
| α-terpinene | | $(1.82 \pm 0.07) \times 10^{-10}$ | RR/ (Atkinson et al., 1985) |
| | | $(1.03 \pm 0.06) \times 10^{-10}$ | RR/ (Berndt et al., 1996) |
| | | **$1.8 \times 10^{-10}$** | **IUPAC** |
| γ-terpinene | | $(2.4 \pm 0.7) \times 10^{-11}$ | DF-LIF/ (Martínez et al., 1999) |
| | | $(2.94 \pm 0.05) \times 10^{-11}$ | RR/ (Atkinson et al., 1985) |
| | | **$2.9 \times 10^{-11}$** | **IUPAC** |
| Terpinolene | | $(5.2 \pm 0.9) \times 10^{-11}$ | DF-LIF/ (Martínez et al., 1999) |
| | | $(9.67 \pm 0.51) \times 10^{-11}$ | RR/ (Corchnoy and Atkinson, 1990) |
| | | $(6.12 \pm 0.52) \times 10^{-11}$ | RR/ (Stewart et al., 2013) |
| | | **$9.7 \times 10^{-11}$** | **IUPAC** |
| Ocimene (cis, trans) | | $(2.23 \pm 0.06) \times 10^{-11}$ | RR/ (Atkinson et al., 1985) |
| | | **$2.2 \times 10^{-11}$** | **IUPAC** |
| Myrcene | | $(1.28 \pm 0.11) \times 10^{-11}$ | DF-LIF/ (Martínez et al., 1999) |
| | | $2.2 \times 10^{-12} \exp[(523 \pm 35)/T]$ | DF-LIF/ (Martínez et al., 1999) |
| | | $(1.06 \pm 0.02) \times 10^{-11}$ | RR/ (Atkinson et al., 1985) |
| | | **$1.1 \times 10^{-11}$** | **IUPAC** |
| α-cedrene | | $(0.82 \pm 0.30) \times 10^{-11}$ | RR/ (Shu and Atkinson, 1995) |



| | | | |
|---|---|---|---|
| α-copaene | | $(1.6 \pm 0.6) \times 10^{-11}$ | RR/ (Shu and Atkinson, 1995) |
| β-caryophyllene | | $(1.9 \pm 0.8) \times 10^{-11}$ | RR/ (Shu and Atkinson, 1995) |
| α-humulene | | $(3.5 \pm 1.3) \times 10^{-11}$ | RR/ (Shu and Atkinson, 1995) |
| longifolene | | $(6.8 \pm 2.1) \times 10^{-13}$ | RR/ (Shu and Atkinson, 1995) |
| isolongifolene | | $(3.9 \pm 1.6) \times 10^{-12}$ | RR/ (Canosa-Mas et al., 1999b) |
| alloisolongifolene | | $(1.4 \pm 0.7) \times 10^{-12}$ | RR/ (Canosa-Mas et al., 1999b) |
| α-neoclovene | | $(8.2 \pm 4.6) \times 10^{-12}$ | RR/ (Canosa-Mas et al., 1999b) |





| | | | |
|---|---|---|---|
| | | $4.6\times10^{-14}\exp[-(400 \pm 35)/T]$ | F-A/ (Rudich et al., 1996) |
| | | $(1.21 \pm 0.09)\times10^{-14}$ | F-A/ (Rudich et al., 1996) |
| | | $(2.1 \pm 0.3)\times10^{-14}$ | DF-A/ (Hallquist et al., 1996) |
| 2-methyl-3-buten-2-ol | | $(1.55 \pm 0.55)\times10^{-14}$ | RR/ (Hallquist et al., 1996) |
| | | $(8.7 \pm 3.0)\times10^{-15}$ | RR/ (Fantechi et al., 1998a) |
| | | $(1.0 \pm 0.2)\times10^{-14}$ | RR/ (Noda et al., 2002) |
| | | $(1.1 \pm 0.1)\times10^{-14}$ | RR/ (Noda et al., 2002) |
| | | **$1.2\times10^{-14}$** | **IUPAC** |
| 3-methyl-2-buten-1-ol | | $(1.0 \pm 0.1)\times10^{-12}$ | RR/ (Noda et al., 2002) |
| 3-methyl-3-buten-1-ol | | $(2.7 \pm 0.2)\times10^{-13}$ | RR/ (Noda et al., 2002) |
| cis-3-hexen-1-ol | | $(2.72 \pm 0.83)\times10^{-13}$ | RR/ (Atkinson et al., 1995) |
| | | $(2.67 \pm 0.42)\times10^{-13}$ | DF-CEAS/ (Pfrang et al., 2006) |
| trans-3-hexen-1-ol | | $(4.43 \pm 0.91)\times10^{-13}$ | DF-CEAS/ (Pfrang et al., 2006) |
| cis-4-hexen-1-ol | | $(2.93 \pm 0.48)\times10^{-13}$ | DF-CEAS/ (Pfrang et al., 2006) |
| trans-2-hexen-1-ol | | $(1.30 \pm 0.24)\times10^{-13}$ | DF-CEAS/ (Pfrang et al., 2006) |
| cis-2-hexen-1-ol | | $(1.56 \pm 0.24)\times10^{-13}$ | DF-CEAS/ (Pfrang et al., 2006) |
| | | $(1.21 \pm 0.44)\times10^{-14}$ | RR/ (Atkinson et al., 1995) |
| trans-2-hexenal | | $(1.36 \pm 0.29)\times10^{-14}$ | RR/ (Zhao et al., 2011b) |
| | | $(4.7 \pm 1.5)\times10^{-15}$ | AR/ (Kerdouci et al., 2012) |
| 4-methylenehex-5-enal | | $(4.75 \pm 0.35)\times10^{-13}$ | RR/ (Baker et al., 2004) |
| (3Z)-4-methylhexa-3,5-dienal | | $(2.17 \pm 0.30)\times10^{-12}$ | RR/ (Baker et al., 2004) |
| (3E)-4-methylhexa-3,5-dienal | | $(1.75 \pm 0.27)\times10^{-12}$ | RR/ (Baker et al., 2004) |



| Compound | Structure | Rate coefficient | Method / Reference |
|---|---|---|---|
| 4-methylcyclohex-3-en-1-one | | $(1.81 \pm 0.35) \times 10^{-12}$ | RR/ (Baker et al., 2004) |
| cis-3-hexenyl acetate | | $(2.46 \pm 0.75) \times 10^{-13}$ | RR/ (Atkinson et al., 1995) |
| methyl vinyl ketone | | $(3.2 \pm 0.6) \times 10^{-16}$ | DF-LIF/ (Canosa-Mas et al., 1999a) |
| | | $< 1.2 \times 10^{-16}$ | F-A/ (Rudich et al., 1996) |
| | | $(5.0 \pm 1.2) \times 10^{-16}$ | RR/ (Canosa-Mas et al., 1999a) |
| | | $< 6 \times 10^{-16}$ | RR/ (Kwok et al., 1996) |
| | | **$< 6 \times 10^{-16}$** | **IUPAC** |
| methacrolein | | $(4.46 \pm 0.58) \times 10^{-15}$ | RR/ (Kwok et al., 1996) |
| | | $(3.08 \pm 0.18) \times 10^{-15}$ | RR/ (Chew et al., 1998) |
| | | $(3.50 \pm 0.15) \times 10^{-15}$ | RR/ (Chew et al., 1998) |
| | | $(3.72 \pm 0.47) \times 10^{-15}$ | RR/ (Canosa-Mas et al., 1999a) |
| | | **$3.4 \times 10^{-15}$** | **IUPAC** |
| pinonaldehyde | | $(2.40 \pm 0.38) \times 10^{-14}$ | RR/ (Hallquist et al., 1997a) |
| | | $(6.0 \pm 2.0) \times 10^{-14}$ | RR/ (Glasius et al., 1997) |
| | | $(2.0 \pm 0.9) \times 10^{-14}$ | RR/ (Alvarado et al., 1998) |
| | | **$2.0 \times 10^{-14}$** | **IUPAC** |
| linalool | | $(1.12 \pm 0.40) \times 10^{-11}$ | RR/ (Atkinson et al., 1995) |
| α-terpineol | | $(1.6 \pm 0.4) \times 10^{-11}$ | RR/ (Jones and Ham, 2008) |
| sabinaketone | | $(3.6 \pm 2.3) \times 10^{-16}$ | RR/ (Alvarado et al., 1998) |
| caronaldehyde | | $(2.5 \pm 1.1) \times 10^{-14}$ | RR/ (Alvarado et al., 1998) |

[a] Rate coefficient data at 298 K unless temperature dependence is specified;

RR: Relative Rate; DF-MS: Discharge Flow-Mass Spectrometry; DF-LIF: Discharge Flow-Laser Induced Fluorescence; DF-A: Discharge Flow-Absorption; DF-CEAS: Discharge Flow-Cavity Enhanced Absorption Spectroscopy; F-LIF: Flow System-Laser Induced Fluorescence; F-CIMS: Flow System-Chemical Ionisation Mass





Spectrometry; F-A: Flow System-Absorption; PR-A: Pulse Radiolysis-Absorption; AR: Absolute rate in simulation chamber

### 2.1.2 Mechanisms

In general, $NO_3$ reacts with unsaturated VOCs by addition to a double bond (Wayne et al., 1991), though hydrogen abstraction may occur, most favorably for aldehydic species (Zhang and Morris, 2015). The location and likelihood of the $NO_3$ addition to a double bond depends on the substitution on each end of the double bond, with the favored $NO_3$ addition position being the one resulting in the most substituted carbon radical. In both cases, molecular oxygen adds to the resulting radical to form a peroxy radical ($RO_2$).

The fate of $RO_2$ determines the subsequent chemistry. During the nighttime in the ambient atmosphere, $RO_2$ will react with another $RO_2$, $NO_3$, or $HO_2$. Thus, $NO_3$+BVOC can be a source of nighttime $HO_2$ and OH radicals (Platt et al., 1990). Reaction with NO is a minor peroxy radical fate at night (Pye et al., 2015;Xiong et al., 2015). Few laboratory studies have contrasted the reactivities of $RO_2$ with $NO_3$ and $HO_2$ and their impacts on gas-phase oxidation and aerosol formation (Ng et al., 2008;Boyd et al., 2015;Schwantes et al., 2015). Boyd et al. (2015) examined how $RO_2$ fate influences SOA formation and yields and studied the competition between the $RO_2$-$NO_3$ and $RO_2$-$HO_2$ channels for $\beta$-pinene. Boyd et al. (2015) presented a detailed mechanistic scheme and determined that the SOA yields for both channels are comparable, indicating that the volatility distribution of products may not be very different for the different $RO_2$ fates. In contrast, the results from $NO_3$ oxidation of smaller BVOC, such as isoprene, show large differences in SOA yields depending on the $RO_2$ fate (Ng et al., 2008), with larger SOA yields for second generation $NO_3$ oxidation (Rollins et al., 2009).

For example, the first-generation products from one of the $RO_2$ isomers formed from isoprene oxidation are shown in Figure 1. Some of the products, such as methyl vinyl ketone, are common between all the pathways for this isomer. However, some products are unique to only one channel (e.g., hydroxy nitrates form from $RO_2$-$RO_2$ reactions and nitrooxy hydroperoxides form from $RO_2$-$HO_2$ reactions). In this case, the overall nitrate yield and the specific nitrates formed from isoprene depend on the fate of the $RO_2$. Furthermore, the distribution of gas-phase products will then influence the formation of SOA. For isoprene, the SOA yields from $RO_2$-$RO_2$ reactions are ~2 times greater than the yield from $RO_2$-$NO_3$ reactions (Ng et al., 2008).

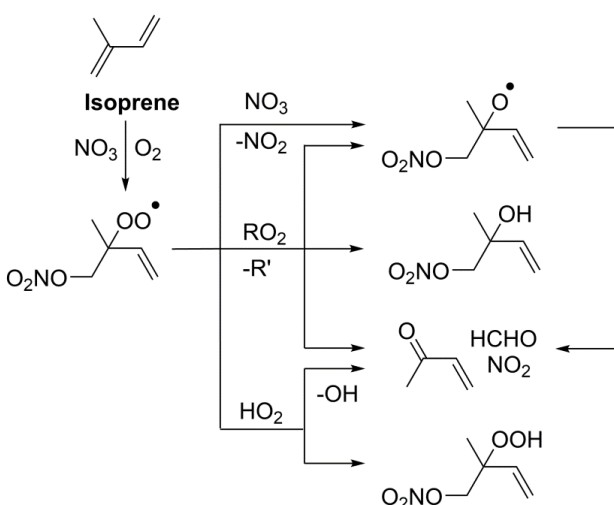

**Figure 1.** Reaction mechanism for one of the $RO_2$ isomers formed from isoprene $NO_3$ oxidation (adapted from (Schwantes et al., 2015))

Given the limited number of studies that have considered the fate of the peroxy radical, generalizations cannot yet be made for all VOCs. Indeed, more studies are needed to determine systematically how gas-phase products and SOA yields are influenced by reactions of $RO_2$. More specifically, for all chamber experiments, constraining the fate and lifetime of $RO_2$ is required to attribute product and SOA yields to a specific pathway. As shown in Table 2 in section 2.2, the

nitrate yields and SOA yields for $NO_3$-induced degradation of many VOCs vary significantly between different studies. This is likely, in part, a result of each experiment having a different distribution of $RO_2$ fates, but may also arise from vapor wall losses.

In general, there are very few mechanistic studies for $NO_3$ relative to other oxidants. Furthermore, the elucidation of mechanisms is limited by the fact that most studies provide

overall yields of organic nitrates (without individual identification of the species) and/or identification (without quantification) due to the lack of standards.

## 2.2 Organic aerosol yields, speciation and particle-phase chemistry

Several papers have reported chamber studies to measure the organic aerosol yield and/or gaseous and aerosol-phase oxidation product distribution from $NO_3$-BVOC reactions. These are

summarized in Table 2. In general, these experimental results show that monoterpenes are





efficient sources of SOA, with reported yields variable but consistently above 20%, with the notable exception of α-pinene (yields 0-15%). This anomalous monoterpene also has a much larger product yield of carbonyls instead of organonitrates compared to the others. This difference among monoterpenes was investigated in the context of the competition between $O_3$ and $NO_3$ oxidation (Draper et al., 2015). The smaller isoprene has substantially lower SOA yield (2-24%), and the only sesquiterpene studied, β-caryophyllene, has a much larger yield (86-150%) than the monoterpenes.

In general, these chamber experiments are conducted at conditions that focus on first generation oxidation only, but further oxidation can continue to change SOA loadings in the real atmosphere (e.g. (Rollins et al., 2009;Chacon-Madrid et al., 2013)). Recent experiments showed that the SOA and organic nitrate fraction from β-pinene-$NO_3$ stays fairly constant upon photochemical aging, while those SOA formed from α-pinene-$NO_3$ evaporates more readily (Nah et al., 2016b).

Other chamber studies have not reported SOA mass yields or gas-phase product measurements but have otherwise demonstrated the importance of $NO_3$-BVOC reactions to SOA production. These studies have identified β-pinene and Δ-carene as particularly efficient sources of SOA upon $NO_3$ oxidation (Hoffmann et al., 1997), confirmed the greater aerosol-forming potential from β-pinene versus α-pinene (Bonn and Moortgat, 2002), and reported Fourier-Transform Infrared (FTIR) spectroscopy and aerosol mass spectrometry (AMS) measurements of the composition of organonitrates detected in aerosol formed from $NO_3$-isoprene, α-pinene, β-pinene, Δ-carene, and limonene reactions (Bruns et al., 2010).

Relative humidity can be an important parameter as it affects the competition between $NO_3$+BVOC reactions and heterogeneous uptake of $N_2O_5$. Among existing laboratory studies, only a few have focused on the effect of relative humidity (RH) on SOA formation from $NO_3$-initiated oxidation (Bonn and Moortgat, 2002;Spittler et al., 2006;Fry et al., 2009;Boyd et al., 2015). The impact of RH might be important, especially at night and during the early morning when RH near the surface is high and $NO_3$ radical chemistry is competitive with $O_3$ and OH reaction. However, observations of the effect of water on SOA formation originating from $NO_3$ oxidation hint at a varied role. Spittler et al. (2006) reported lower SOA yields under humid conditions, but other studies did not observe a significant effect (Bonn and Moortgat, 2002;Fry et al., 2009;Boyd et al., 2015). Among the important effects of water is its role as a medium for hydrolysis. In laboratory studies, primary and secondary organonitrates were found to be less prone to aqueous hydrolysis than tertiary organonitrates (Darer et al., 2011;Hu et al., 2011). First-generation organic nitrates retaining double bonds may also hydrolyze relatively quickly,





especially in the presence of acidity (Jacobs et al., 2014;Rindelaub et al., 2015). Depending on the relative amount of these different types of organic nitrates, the overall hydrolysis rate could be different for organic nitrates formed from $NO_3$ oxidation and photooxidation in the presence of $NO_x$ (Boyd et al., 2015). Recently, there has been increasing evidence from field measurements

that organic nitrates hydrolyze in the particle phase, producing $HNO_3$ (Liu et al., 2012b;Browne et al., 2013). This has been only a limited focus of chamber experiments to date (Boyd et al., 2015). In addition to the effect of RH, particle-phase acidity is known to affect SOA formation from ozonolysis and OH reaction (e.g. (Gao et al., 2004;Tolocka et al., 2004)). Thus far only one study has examined the effect of acidity on $NO_3$-initiated SOA formation and found a negligible effect

(Boyd et al., 2015). Notably, an effect of acidity was observed for the hydrolysis of organonitrates produced in photochemical reactions (Szmigielski et al., 2010;Rindelaub et al., 2015). While much organic nitrate aerosol is formed via $NO_3$ + BVOC reactions, some fraction can also form from $RO_2$ + NO chemistry. Rollins et al. (2010) observed the organic nitrate moiety in 6-15% of total SOA mass generated from high-$NO_x$ photooxidation of limonene, α-

pinene, Δ-3-carene, and tridecane. A very recent study of Berkemeier et al (Berkemeier et al., 2016) showed that organic nitrates accounted for ∼40% of SOA mass during initial particle formation in α-pinene oxidation by $O_3$ in the presence of NO, decreasing to ∼15% upon particle growth to the accumulation-mode size range. They also observed a tight correlation between organic nitrate content and SOA particle-number concentrations. This implies that organic

nitrates may be among the extremely low volatility organic compounds (ELVOC) (Ehn et al., 2014;Tröstl et al., 2016) that play a critical role in the nucleation and nano-particle growth.

**Table 2.** Oxidation products and SOA yields observed in previous studies of $NO_3$-BVOC reactions. Except where noted, carbonyl and organonitrate molar yields represent initial gas-phase

yields were measured by FTIR spectroscopy (carbonyl and organonitrate) or thermal desorption laser-induced fluorescence (TD-LIF) (organonitrate only, (Rollins et al., 2010;Fry et al., 2013)). In some cases, the ranges reported correspond to wide ranges of organic aerosol loading, listed in the rightmost column. Where possible, the mass yield at 10 μg m$^{-3}$ is reported for ease of comparison.

| BVOC | Carbonyl molar yield | Organonitrate molar yield | SOA mass yield | Corresponding OA loading or other relevant information |
| --- | --- | --- | --- | --- |





| isoprene | | 62-78%<br>(Rollins et al., 2009) | 2%<br>(14% after further oxidation)<br>(Rollins et al., 2009) | Nucleation (1 µg m$^{-3}$) |
|---|---|---|---|---|
| | | | 4 – 24%<br>(Ng et al., 2008) | 3 – 70 µg m$^{-3}$; 12% at 10 µg m$^{-3}$ |
| α-pinene | 58-66%<br>(Wangberg et al., 1997)<br>69-81%<br>(Berndt and Boge, 1997a)<br>65-72%<br>(Hallquist et al., 1999)<br>39-58%<br>(Spittler et al., 2006) | 14%<br>(Wangberg et al., 1997)<br>12-18%<br>(Berndt and Boge, 1997b)<br>18-25%<br>(Hallquist et al., 1999)<br>11-29%<br>(Spittler et al., 2006)<br>10%<br>(Fry et al., 2014) | 0.2 – 16%<br>(Hallquist et al., 1999) | Nucleation; 0.5% at 10 ppb N$_2$O$_5$ reacted, 7% at 100 ppb N$_2$O$_5$ reacted* |
| | | | 4 or 16%<br>(Spittler et al., 2006) | Values for 20% RH and dry conditions, respectively, both with organic seed aerosol generated by O$_3$+BVOC. Yields reported correspond to "M$_\infty$": extrapolated value at highest mass loading (see note below at limonene) |
| | | | 1.7 – 3.6%<br>(Nah et al., 2016a) | 1.2 – 2.5 µg m$^{-3}$ |
| | | | 0%<br>(Fry et al., 2014) | Both nucleation and ammonium sulfate seeded |
| β-pinene | 0-2%<br>(Hallquist et al., 1999) | 51-74%<br>(Hallquist et al., 1999)<br>40%<br>(Fry et al., 2009)<br>22%<br>(Fry et al., 2014)<br>45%-74% of OA mass,<br>(Boyd et al., 2015) | 32 – 89%<br>(Griffin et al., 1999) | 32 – 470 µg m$^{-3}$; low end closest to 10 µg m$^{-3}$ |
| | | | 7 – 40% (Moldanova and Ljungstrom, 2000) using new model to reinterpret data from (Hallquist et al., 1999) (10 – 52%) | 7 – 10% at 7 ppb N$_2$O$_5$ reacted, 40 – 52% at 39 ppb N$_2$O$_5$ reacted |
| | | | 50% | 40 µg m$^{-3}$; same yield at both 0% and |





| | | | (Fry et al., 2009) | 60% RH |
|---|---|---|---|---|
| | | | 33-44%<br>(Fry et al., 2014) | 10 µg m$^{-3}$ ** |
| | | | 27 – 104% (Boyd et al., 2015) | 5 – 135 µg m$^{-3}$, various seeds and RO$_2$ fate regimes; 50 % for expts near 10 µg m$^{-3}$ |
| Δ-carene | 0-3%<br>(Hallquist et al., 1999) | 68-74%<br>(Hallquist et al., 1999)<br>77%<br>(Fry et al., 2014) | 13 – 72%<br>(Griffin et al., 1999) | 24 – 310 µg m$^{-3}$; low end closest to 10 µg m$^{-3}$ |
| | | | 12 – 49% (Moldanova and Ljungstrom, 2000) using new model to reinterpret data from (Hallquist et al., 1999) (15 – 62%) | 7 – 395 ppb N$_2$O$_5$ reacted, 12-15% at 6.8 ppb N$_2$O$_5$ reacted |
| | | | 38 – 65%<br>(Fry et al., 2014) | 10 µg m$^{-3}$ ** |
| limonene | 69%<br>(Hallquist et al., 1999)<br>25-33%<br>(Spittler et al., 2006) | 48%<br>(Hallquist et al., 1999)<br>63-72%<br>(Spittler et al., 2006)<br>30%<br>(Fry et al., 2011)<br>54%<br>(Fry et al., 2014) | 14 – 24% (Moldanova and Ljungstrom, 2000) using new model to retinterpret data from (Hallquist et al., 1999) (17%) | 10 ppb N$_2$O$_5$ reacted; higher number in (Moldanova and Ljungstrom, 2000) from an additional injection of 7 ppb N$_2$O$_5$ and accounting for secondary reactions |
| | | | 21 or 40%<br>(Spittler et al., 2006) | Ammonium sulfate or organic seed, respectively. Yields reported correspond to "M$_\infty$": extrapolated value at highest mass loading (full dataset shown only for limonene, where asymptote is 400 µg m$^{-3}$) |
| | | | 25 – 40%<br>(Fry et al., 2011) | Nucleation to 10 µg m$^{-3}$ (second injection of oxidant) |
| | | | 44 – 57%<br>(Fry et al., 2014) | 10 µg m$^{-3}$ ** |
| sabinene | | | 14 – 76%<br>(Griffin et al., 1999) | 24 – 277 µg m$^{-3}$; low end closest to 10 µg m$^{-3}$ |



| | 25 – 45%<br>(Fry et al., 2014) | 10 μg m$^{-3}$ ** |
| β-caryophyllene | 91 – 146%<br>(Jaoui et al., 2013) | 60 – 130 μg m$^{-3}$; low end closest to 10 μg m$^{-3}$ |
| | 86%<br>(Fry et al., 2014) | 10 μg m$^{-3}$ |

*The authors assume that $N_2O_5$ reacted = BVOC reacted.
**Yield range corresponds to two different methods of calculating ΔBVOC.

## 2.3 Heterogeneous and aqueous phase $NO_3$ processes

The $NO_3$ radical is not only a key nighttime oxidant of organic (and especially biogenic) trace
gases, but it can also play an important role in the aqueous phase of tropospheric clouds and
deliquesced particles (Chameides, 1978;Wayne et al., 1991;Herrmann and Zellner, 1998;Rudich
et al., 1998). Whilst the reaction of $NO_3$ with organic particles and aqueous droplets in the
atmosphere is believed to represent only an insignificant fraction of the overall loss rate for $NO_3$,
it can have a substantial impact on the chemical and physical properties of the particle by
modifying its lifetime, oxidation state, viscosity and hygroscopic properties and thus its
propensity to act as a cloud condensation nucleus (Rudich, 2003).

Biogenic VOC include, but are not limited to the isoprenoids (isoprene, mono- and
sesquiterpenes) as well as alkanes, alkenes, carbonyls, alcohols, esters, ethers, and acids
(Kesselmeier and Staudt, 1999). Recent measurements indicate that biogenic emissions of
aromatic trace-gases are also significant (Misztal et al., 2015). The gas-phase degradation of
BVOC leads to the formation of a complex mixture of organic trace gases including hydroxyl-
and nitrate- substituted oxygenates which can transfer to the particle phase by condensation or
dissolution. Our present understanding is that non-anthropogenic SOA has a large contribution
from isoprenoid degradation.

As is generally the case for laboratory studies of heterogeneous processes, most of the
experimental investigations on heterogeneous uptake of $NO_3$ to organic surfaces have dealt with
single component systems that act as surrogates for the considerably more complex mixtures
found in atmospheric SOA. A further level of complexity arises when we consider that initially
reactive systems, e.g. containing condensed or dissolved unsaturated hydrocarbons, can become



deactivated as SOA ages, single bonds replace double bonds and the oxygen-to-carbon ratio increases.

We summarize the results of the laboratory studies to provide a rough guide to $NO_3$ reactivity on different classes of organics which may be present in SOA and note that further studies of $NO_3$

uptake to biogenic SOA which was either generated and aged under well-defined conditions (Fry et al., 2011) or sampled from the atmosphere are required to confirm predictions of uptake efficiency based on the presently available database.

### 2.3.1 Heterogeneous processes

For some particle-phase organics, the reaction with $NO_3$ is at least as important as other

atmospheric oxidants such as $O_3$ and OH (Shiraiwa et al., 2009;Kaiser et al., 2011). The lifetime ($\tau$) of a single component, liquid organic particle with respect to loss by reaction with $NO_3$ at concentration $[NO_3]$ is partially governed by the uptake coefficient ($\gamma$) (Robinson et al., 2006;Gross et al., 2009):

$$\tau_{liquid} = \frac{2\rho_{org}N_A D_p}{3M_{org}\bar{c}\gamma[NO_3]} \qquad (1)$$

Where $D_p$ is the particle diameter, $\rho_{org}$ and $M_{org}$ are the density and molecular weight of the organic component, respectively, and $\bar{c}$ is the mean molecular velocity of gas-phase $NO_3$, and $N_A$ is Avogadro number. Thus defined, $\tau$ is the time required for all the organic molecules in a

spherical (i.e. liquid) particle to be oxidized once.

Recent studies have shown that organic aerosols can adopt semi-solid (highly viscous) or amorphous solid (crystalline or glass) phase states, depending on the composition and ambient conditions (Virtanen et al., 2010;Koop et al., 2011;Renbaum-Wolff et al., 2013). Typically the bulk phase diffusion coefficients of $NO_3$ are $\sim10^{-7} - 10^{-9}$ cm$^2$ s$^{-1}$ in semisolid and $\sim10^{-10}$ cm$^2$ s$^{-1}$ in

solids (Shiraiwa et al., 2011). Slow bulk diffusion of $NO_3$ in a viscous organic matrix can effectively limit rate of uptake (Xiao and Bertram, 2011;Shiraiwa et al., 2012). Similarly, the solubility may be different in a concentrated, organic medium. If bulk diffusion is slow, the reaction may be confined to the near-surface layers of the particle or bulk substrate. The presence of organic coatings on aqueous aerosols was found to suppress heterogeneous $N_2O_5$ hydrolysis by

providing a barrier through which $N_2O_5$ needs to diffuse to undergo hydrolysis (Alvarado et al., 1998;Cosman et al., 2008;Grifiths et al., 2009). Reactive uptake by organic aerosols is expected





to exhibit a pronounced decrease at low RH and temperature, owing to phase transition from viscous liquid to semi-solid or amorphous solid (Arangio et al., 2015). Therefore, the presence of a semi-solid matrix may effectively shield reactive organic compounds from chemical degradation in long-range transport in the free troposphere.

To get an estimate of the processing rate of BVOC-derived SOA we have summarized the results of several laboratory studies to provide a rough guide to $NO_3$ reactivity on different classes of organics that may be present in SOA (Figure 2). We note that further studies of $NO_3$ uptake to biogenic SOA which was either generated and aged under well-defined conditions (Fry et al., 2011) or sampled from the atmosphere are required to confirm predictions of uptake efficiency

based on the presently available database.

A rough estimate of the reactivity of $NO_3$ to freshly generated, isoprenoid-derived SOA, which still contains organics with double bonds (e.g., from di-olefinic montoterpenes such as limonene) may be obtained by considering the data on alkenes and unsaturated acids, where the uptake coefficient is generally close to 0.1.

The classes of organics for which heterogeneous reactions with $NO_3$ have been examined are: alkanoic / alkenoic acids, alkanes and alkenes, alcohols, aldehydes, polyaromatic hydrocarbons (PAHs) and secondary organic aerosol. Laboratory studies have used either pure organic substrates, the organic of interest internally mixed in an aqueous particle or as a surface coating, the reactive organic mixed in a non-reactive organic matrix, or in the form of self-assembling

monolayers. The surrogate surface may be available as a macroscopic bulk liquid (or frozen liquid) or in particulate form and both gas-phase and particle-phase analysis has been used to derive kinetic parameters and investigate products formed.

In the gas-phase, the $NO_3$ radical reacts slowly (by H-abstraction) with alkanes, more rapidly with aldehydes due to the weaker C-H bond of the carbonyl group, and most readily with alkenes and

aromatics via electrophilic addition. This trend in reactivity is also observed in the condensed phase reactions of $NO_3$ with organics so that long chain organics, for which non-sterically hindered addition to a double bond is possible, and aromatics are the most reactive. In very general terms, uptake coefficients are in the range of $1\text{-}10 \times 10^{-3}$ for alkanes, alcohols and acids without double bonds, $2\text{-}200 \times 10^{-3}$ for alkenes with varying numbers of double bonds, $3\text{-}1000 \times$

$10^{-3}$ for acids with double bonds again depending on the number of double bonds, and $100\text{-}500 \times 10^{-3}$ for aromatics. These trends are illustrated in Figure 2 which plots the experimental data for the uptake of $NO_3$ to single component organic surfaces belonging to different classes of



condensable organics. Condensed-phase organic nitrates have been frequently observed following interaction of $NO_3$ with organic surfaces (see below).

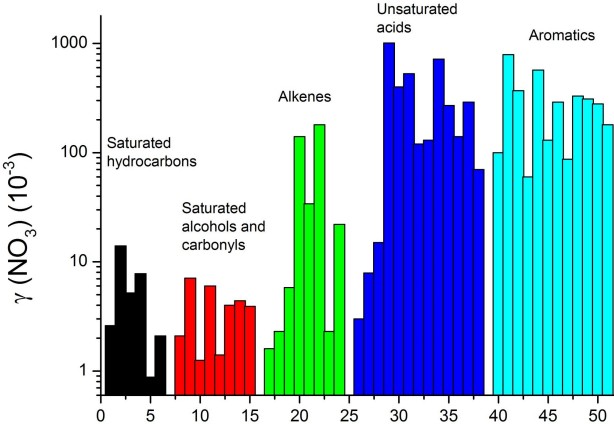

**Figure 2**. Uptake coefficients, $\gamma(NO_3)$, for the interaction of $NO_3$ with single component, organic surfaces. Details of the experiments and the references (corresponding to the x-axis numbers) are given in Table SI-1.

### Saturated hydrocarbons

Uptake of $NO_3$ to saturated hydrocarbons is relatively slow, with uptake coefficients close to $10^{-3}$. Moise et al. (2002) found that (for a solid sample) uptake to a branched chain alkane was more efficient than for a straight-chain alkane, which is consistent with known trends in gas-phase reactivity of $NO_3$. The slow surface reaction with alkanes enables both surface and bulk components of the reaction to operate in parallel. The observation of $RONO_2$ as product is explained (Knopf et al., 2006;Gross and Bertram, 2009) by processes similar to those proceeding in the gas-phase, i.e. abstraction followed by formation of peroxy and alkoxy intermediates which react with $NO_2$ and $NO_3$ to form the organic nitrate.

### Unsaturated hydrocarbons

With exception of the data of Moise et al. (2002), the uptake of $NO_3$ to an unsaturated organic surface is found to be much more efficient than to the saturated analogue. The $NO_3$ uptake coefficient for e.g. squalene is at least an order of magnitude more efficient than for squalane (Xiao and Bertram, 2011;Lee et al., 2013). The location of the double bond is also important and the larger value for $\gamma$ found for a self-assembling monolayer of $NO_3$ + Undec-10-ene-1-thiol compared to liquid, long-chain alkenes is due to the fact that the terminal double bond is located



at the interface and is thus more accessible for a gas-phase reactant (Gross and Bertram, 2009). $NO_3$ uptake to mixtures of unsaturated methyl oleate in a matrix of saturated organic was found to be consistent with either a surface or bulk reaction (Xiao and Bertram, 2011). The formation of condensed phase organic nitrates and simultaneous loss of the vinyl group indicates that the

reaction proceeds, as in the gas-phase, by addition of $NO_3$ to the double bond followed by reaction of $NO_3$ (or $NO_2$) with the resulting alkyl and peroxy-radicals formed (Zhang et al., 2014b).

### Saturated alcohols and carbonyls

Consistent with reactivity trends for $NO_3$ in the gas-phase, the weakening of some C-H bonds in

oxidized, saturated organics results in a more efficient interaction of $NO_3$ than for the non-oxidized counterparts although, as far as the limited data set allows trends to be deduced, the gas-phase reactivity trend of polyalcohol > alkanoate appears to be reversed in the liquid-phase (Gross et al., 2009). For multi-component, liquid particles, the uptake coefficient will also depend on the particle viscosity (Iannone et al., 2011) though it has not been clearly established if the

reaction proceeds predominantly at the surface or throughout the particle (Iannone et al., 2011). The reaction products are expected to be formed via similar pathways as seen in the gas phase, i.e. abstraction of the aldehydic-H atom for aldehydes and abstraction of an H-atom from either the O-H or adjacent $\alpha$-$CH_2$ group for alcohols prior to reaction of $NO_2$ and $NO_3$ with the ensuing alkyl and peroxy radicals (Zhang and Morris, 2015).

### Organic Acids

The efficiency of uptake of $NO_3$ to unsaturated acids is comparable to that found with other oxidized, saturated organics (Moise et al., 2002) suggesting that the reaction proceeds, as in the gas phase, via abstraction rather than addition. Significantly larger uptake coefficients have been observed for a range of unsaturated, long chain acids, with $\gamma$ often between 0.1 and 1 (Gross et

al., 2009;Knopf et al., 2011;Zhao et al., 2011a). $\gamma$ depends on the number and position (steric factors) of the double bond. For example, the uptake coefficient for abietic acid is a factor of 100 lower than for linoleic acid (Knopf et al., 2011). The condensed-phase products formed in the interaction of $NO_3$ with unsaturated acids are substituted carboxylic acids, including hydroxy nitrates, carbonyl nitrates, dinitrates, and hydroxy dinitrates (Hung et al., 2005;Docherty and

Ziemann, 2006;McNeill et al., 2007;Zhao et al., 2011a).

### Aromatics

The interaction of $NO_3$ with condensed-phase aromatics and polycyclic aromatic hydrocarbons (PAH) results in the formation of a large number of nitrated aromatics and nitro-PAHs. Similar to



the gas-phase mechanism, the reaction is initiated by addition of $NO_3$ to the aromatic ring, followed by breaking of an N-O bond to release $NO_2$ to the gas phase and forming a nitrooxycyclohexadienyl-type radical which can further react with $O_2$, $NO_2$ or undergo internal rearrangement to form hydroxyl species (Gross and Bertram, 2008;Lu et al., 2011). The uptake

coefficients are large and comparable to those derived for the unsaturated fatty acids.

The literature results on the interaction of $NO_3$ with organic substrates are tabulated in Table SI-1 of the supplementary information, in which the uptake coefficient is listed (if available) along with the observed condensed- and gas-phase products.

### 2.3.2 Aqueous-phase reactions

The in-situ formation of $NO_3$ (e.g. electron transfer reactions between nitrate anions and other aqueous radical anions, e.g. $SO_x^-$ or $Cl_2^-$), is generally of minor importance and the presence of $NO_3$ in aqueous particles is largely a result of transfer from the gas phase (Herrmann et al., 2005;Tilgner et al., 2013). Concentrations of $NO_3$ in tropospheric aqueous solutions cannot be instantaneously measured and literature values are based on multiphase model predictions

(Herrmann et al., 2010). Model studies with the CAPRAM mechanism (Chemical Aqueous Phase RAdical Mechanism (CAPRAM; (Herrmann et al., 2005;Tilgner et al., 2013)) predicts between $1.6 \cdot 10^{-16}$ mol $L^{-1}$ to $2.7 \cdot 10^{-13}$ mol $L^{-1}$ . The higher values are associated with urban clouds, with rural and marine clouds an order of magnitude lower.

$NO_3$ radicals react with dissolved organic species via three different pathways: (i) by H-atom

abstraction from saturated organic compounds, (ii) by electrophilic addition to double bonds within unsaturated organic compounds, and (iii) by electron transfer from dissociated organic acids (Huie, 1994;Herrmann and Zellner, 1998). For a detailed overview on aqueous-phase $NO_3$ radical kinetics, the reader is referred to several recent summaries (Neta et al., 1988;Herrmann and Zellner, 1998;Ross et al., 1998;Herrmann, 2003;Herrmann et al., 2010;Herrmann et al.,

2015). Compared to the highly reactive and non-selective OH radical, the $NO_3$ radical is characterized by a lower reactivity and represents a more selective aqueous-phase oxidant. The available kinetic data indicate that the reactivity of $NO_3$ radicals with organic compounds in comparison to the two other key radicals (OH, $SO_4^-$) is as follows: OH > $SO_4^-$ >> $NO_3$ (Herrmann et al., 2015).

In Table SI-2, we list kinetic parameters for reaction of $NO_3$ with aliphatic organic compounds as presently incorporated in the CAPRAM database (Bräuer et al., 2016). Typical ranges of rate constants (in $M^{-1}$ $s^{-1}$) for reactions of $NO_3$ in the aqueous phase are $10^6 - 10^7$ for saturated





alcohols, carbonyls and sugars, $10^4-10^6$ for protonated aliphatic mono- and di-carboxylic acids, with higher values for oxygenated acids, $10^6-10^8$ for deprotonated aliphatic mono- and di-carboxylic acids (higher values typically for oxygenated acids), $10^7-10^9$ for unsaturated aliphatic compounds and $10^8-2\times10^9$ for aromatic compounds (without nitro/acid functionality). The

somewhat larger rate constants for deprotonated aliphatic mono- and di-carboxylic acids, unsaturated aliphatic compounds and aromatic compounds is related to the occurrence of electron transfer reactions and addition reaction pathways, which are often faster than H-abstraction reactions.

Many aqueous-phase $NO_3$ reaction rate constants, even for small oxygenated organic compounds,

are not available in the literature and have to be estimated. In the absence of structure–activity relationships (SARs) for $NO_3$ radical reactions with organic compounds, Evans–Polanyi-type reactivity correlations are used to predict kinetic data for H-abstraction $NO_3$ radical reactions. The latest correlation for $NO_3$ reactions in aqueous solution based on 38 H-abstraction reactions of aliphatic alcohols, carbonyl compounds and carboxylic acids was published by (Hoffmann et al.,

2009) (see Eq. 5).

$$\log (k_H) = (39.9 \pm 5.4) - (0.087 \pm 0.014) \cdot BDE \qquad (2)$$

where BDE is the bond dissociation energy (in kJ mol$^{-1}$). The correlation is quite tight, with a

correlation coefficient of R = 0.9.

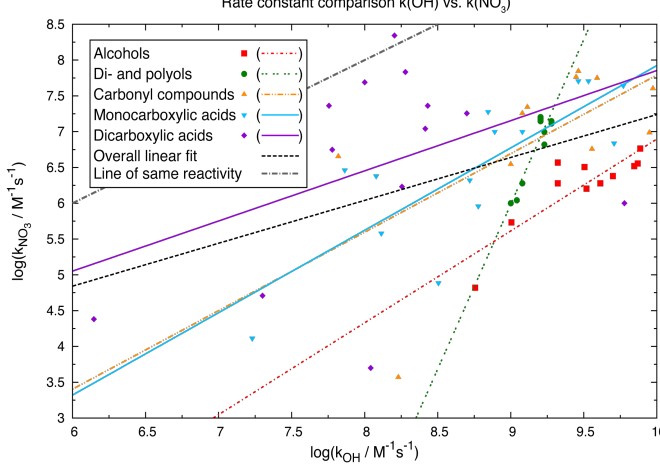





**Figure 3.** Correlation of OH versus $NO_3$ radical rate constants in the aqueous phase for the respective compound classes. The linear regression fits for the different compound classes are presented in the same color as the respective data points. The black line represents the correlation of the overall data

A direct comparison of the aqueous-phase OH and $NO_3$ radical rate constants ($k_{298K}$) of organic compounds from different compound classes is presented in Figure 3, which shows that the $NO_3$ radical reaction rate constants for many organic compounds are about 2 orders of magnitude smaller than respective OH rate constants. In contrast, deprotonated dicarboxylic acids can react

10 with $NO_3$ via electron transfer, and have similar rate constant for OH-reaction. Rate constants for OH and $NO_3$ with alcohols and di-/polyols are well correlated, whereas those rate constants for carbonyl compounds and di-acids have a lower degree of correlation.

Figure 4 shows a comparison of the modeled chemical turnovers of reactions of organic compounds with hydroxyl (OH) versus nitrate ($NO_3$) radicals distinguished for different

15 compound classes. The simulations were performed with the SPACCIM model (Wolke et al., 2005) for the urban summer CAPRAM scenario (see (Tilgner et al., 2013) for details) using the MCM3.2/CAPRAM4.0 mechanism (Rickard, 2015;Bräuer et al., 2016) which has in total 862 $NO_3$ radical reactions with organic compounds.

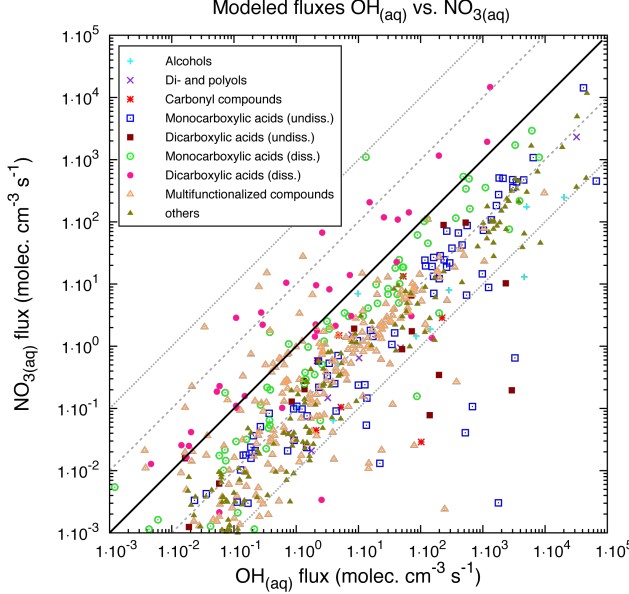



**Figure 4.** Comparison of modeled, aqueous-phase reaction fluxes (mean chemical fluxes in molecule $cm^{-3}$ $s^{-1}$ over a simulation period of 4-5 days) of organic compounds with hydroxyl (OH) versus nitrate ($NO_3$) radicals distinguished by different compound classes (urban CAPRAM summer scenario).

Most of the data lie under the 1:1 line, indicating that, for most of the organic compounds considered, chemical degradation by OH is more important than by $NO_3$, with a significant fraction of the data lying close to a 10:1 line, though OH fluxes sometime exceed $NO_3$ fluxes by a factor of $10^3$ - $10^4$. Approximate, relative flux ratios ($NO_3$/OH) for different classes of organic

are: $10^{-1}$-$10^{-2}$ for alcohols (incl. di- and polyols) and carbonyl compounds, $10^{-1}$-$10^{-4}$ for undissociated mono-acids and di-acids, ~ 1 (or larger) for dissociated mono-acids, $10^{-2}$ - >10 for dissociated diacids, $10^{-2}$ – 1 for organic nitrates. For carboxylate ions, $NO_3$-initiated electron transfer is thus the dominant oxidation pathway. As OH-initiated oxidation proceeds via an H-abstraction, high $NO_3$-OH flux ratios can be observed for carboxylate ions but not for protonated

carboxylic acids.

Overall, Figure 4 shows that, over a four-day, summer cycle, $NO_3$ radical reactions can compete with OH radical reactions in particular for protonated carboxylic acids and multifunctional compounds. Nevertheless, aqueous $NO_3$ radical reactions with organics will become more important during winter or at higher latitudes, where photochemistry as the main source of OH is

less important. Finally, it should be noted that $NO_3$ aqueous phase nighttime chemistry will influence the concentration levels of many aqueous phase reactants available for reaction during the next day.

## 2.4 Instrumental methods

Atmospheric models of the interaction of $NO_3$ with BVOC rely on experimental data gathered in

both the laboratory and the field. These experimental data are used to define model parameters and to evaluate model performance by comparison to observed quantities. Instrumentation for measurements of nitrogen-containing species, oxidants and organic compounds, including $NO_x$, $O_3$, $NO_3$, BVOC and oxidized reactive nitrogen compounds are all important to understand the processes at work. Of particular importance to the subject of this review is the characterization of

organic nitrates, which are now known to exist in both the gas and particle phases and whose





atmospheric chemistry is complex. This section reviews historical and current experimental methods used for elucidating $NO_3$-BVOC atmospheric chemistry.

### 2.4.1 Nitrate radical measurements

Optical absorption spectroscopy has been the primary measurement technique for $NO_3$. It usually

makes use of two prominent absorption features of $NO_3$ near 623 nm and 662 nm. Note that the dissociation limit of the $NO_3$ molecule lies between the two absorption lines (Johnston et al., 1996), thus illumination by measurement radiation at the longer wavelength band does not lead to photolysis of $NO_3$. The room temperature absorption cross section of $NO_3$ at 662 nm is $\sim 2\times10^{-17}$ $cm^2$ molecule$^{-1}$ and increases at lower temperature (Yokelson et al., 1994;Osthoff et al., 2007).

Thus at a typical minimum detectable optical density (reduction of the intensity compared to no absorption) and a light path length of 5 km, a detection limit of $10^7$ molecules/$cm^3$ or $\sim$0.4 ppt (under standard conditions) is achieved.

Initial measurements of $NO_3$ in the atmosphere were "long path" averages using light paths between either the sun or the moon (e.g. (Noxon et al., 1978)) and the receiving spectrometer

(also called passive techniques because natural light sources were used) or between an artificial light source and spectrometer (active techniques, e.g. (Platt et al., 1980)). Passive techniques were later extended to yield $NO_3$ vertical profiles (e.g. (Weaver et al., 1996)). In recent years, resonator cavity techniques allowed construction of very compact instruments capable of performing in situ measurements of $NO_3$ with absorption spectroscopy (see section 2.4.2.2).

An important distinction between the techniques is whether $NO_3$ can be deliberately or inadvertently removed from the absorption path as part of the observing strategy. Long path absorption spectroscopy does not allow control over the sample for obtaining a zero background by removing $NO_3$ (Category 1). Resonator techniques (at least as long as the resonator is encased) allow deliberate removal of $NO_3$ from the absorption path as part of the measurement sequence

and may also result in inadvertent removal during sampling (Category 2).

For instruments of Category 1, the intensity without absorber ($I_0$) cannot be easily detected. Therefore the information about the absorption due to $NO_3$ (and any other trace gas) has to be determined from the structure of the absorption, which is usually done by using Differential Optical Absorption Spectroscopy (DOAS) (Platt and Stutz, 2008), which relies on the

characteristic fingerprint of the $NO_3$ absorption structure in a finite wavelength range (about several 10 nanometers wide). Thus a spectrometer of sufficient spectral range and resolution (around 0.5 nm) is required.



Instruments of Category 2 can determine the $NO_3$ concentration from the difference (or rather log of the ratio) of the intensity with and without $NO_3$ in the measurement volume. In this case only an intensity measurement at a single wavelength (typically of a laser) is necessary. However, enhanced specificity can be gained by combining resonator techniques with DOAS detection. It

should be noted that the advantage of a closed cavity to be able to remove (or manipulate) $NO_3$ comes at the expense of potential wall losses, which have to be characterized. Such instruments have the advantage of being able to also detect $N_2O_5$, which is in thermal equilibrium with $NO_3$ and can be quantitatively converted to $NO_3$ by thermal dissociation (Brown et al., 2001;Brown et al., 2002).

Another complication arises from the presence of water vapor and oxygen lines in the wavelength range of strong $NO_3$ absorptions. To compensate for these potential interferences in open path measurements (where $NO_3$ cannot easily be removed), daytime measurements are frequently used as reference because $NO_3$ levels are typically very low (but not necessarily negligibly low) (Geyer et al., 2003). Thus a good fraction of the reported $NO_3$ data (in particular older data)

represents day-night differences.

### *Passive long-path remote sensing techniques*

Measurements of the $NO_3$ absorption structure using sunlight take advantage of the fact that $NO_3$ is very quickly photolyzed by sunlight (around 5 seconds lifetime during the day) allowing for vertically resolved measurements during twilight (e.g. (Aliwell and Jones, 1998;Allan et al.,

2002;Coe et al., 2002;von Friedeburg et al., 2002)). The fact that the $NO_3$ concentration is nearly zero due to rapid photolysis in the directly sunlit atmosphere, while it is largely undisturbed in a shadowed area, can be used to determine $NO_3$ vertical concentration profiles during sunrise using the moon as a light source (Smith and Solomon, 1990;Smith et al., 1993;Weaver et al., 1996). Alternatively the time series of the $NO_3$ column density derived from scattered sunlight

originating from the zenith (or from a viewing direction away from the sun) during sunrise can be evaluated to yield $NO_3$ vertical profiles (Allan et al., 2002;Coe et al., 2002;von Friedeburg et al., 2002).

Nighttime $NO_3$ total column data have been derived by spectroscopy of moonlight and starlight (Naudet et al., 1981), the intensity of which is about 4-5 orders of magnitude lower than that of

sunlight. Thus photolysis of $NO_3$ by moonlight is negligible. A series of moonlight $NO_3$ measurements have been reported (Noxon et al., 1980;Noxon, 1983;Sanders et al., 1987;Solomon et al., 1989;Solomon et al., 1993;Aliwell and Jones, 1996a, b;Wagner et al., 2000). These measurements yield total column data of $NO_3$, the sum of tropospheric and stratospheric partial





columns. Separation between stratospheric and tropospheric $NO_3$ can be accomplished (to some extent) by the Langley-Plot method (Noxon et al., 1980).

### *Active long-path techniques*

A large number of $NO_3$ measurements have been made using the active long-path DOAS
technique (Platt et al., 1980;Platt et al., 1981;Pitts et al., 1984;Platt et al., 1984;Heintz et al., 1996;Allan et al., 2000;Martinez et al., 2000;Geyer et al., 2001a;Geyer et al., 2001b;Gölz et al., 2001;Stutz et al., 2002;Geyer et al., 2003;Stutz et al., 2004;Asaf et al., 2009;McLaren et al., 2010;Stutz et al., 2010;Crowley et al., 2011;Sobanski et al., 2016). Here, a searchlight-type light source is used to transmit a beam of light across a kilometer-long light path in the open
atmosphere to a receiving telescope-spectrometer combination. The light source typically is a broad-band thermal radiator (incandescent lamp, Xe-arc lamp, laser driven light source). More recently LED light sources were also used (Kern et al., 2006). The telescope (around 0.2 m diameter) collects the radiation and transmits it, usually through an optical fiber, into the spectrometer, which produces the absorption spectrum. Modern instruments now almost
exclusively use transmitter/receiver combinations at one end of the light path and retro-reflector arrays (e.g., cat-eye like optical devices) at the other end. The great advantage of this approach is that power and optical adjustment is only required at one end of the light path while the other end (with the retro-reflector array) is fixed. In this way, several retro-reflector arrays, for instance mounted at different altitudes, can be used sequentially with the same transmitter/receiver unit
allowing determination of vertical profiles of $NO_3$ (and other species measurable by DOAS) (Stutz et al., 2002;Stutz et al., 2004;Stutz et al., 2010).

### *In situ measurement techniques*

Cavity ring-down spectroscopy (CRDS) and cavity enhanced absorption spectroscopy (CEAS) are related techniques for in situ quantification of atmospheric trace gases such as $NO_3$. These
methods are characterized by high sensitivity, specificity, and acquisition speed (Table 3), and they allow for spatially resolved measurements on mobile platforms.

In CRDS, laser light is "trapped" in a high-finesse stable optical cavity, which usually consists of a pair of highly reflective spherical mirrors in a near-confocal arrangement. The concentrations of the optical absorbers present within the resonator are derived from the Beer-Lambert law and the
rate of light leaking from the cavity after the input beam has been switched off (O'Keefe and Deacon, 1988). CRDS instruments are inherently sensitive as they achieve long effective optical absorption paths (up to or in some cases exceeding 100 km) as the light decay is monitored for several 100 µs, and the absorption measurement is not affected by laser intensity fluctuations. For



detection of $NO_3$ at 662 nm, pulsed laser sources such as Nd:YAG pumped dye lasers have been used because of the relative ease of coupling the laser beam to the optical cavity (Brown et al., 2002;Brown et al., 2003;Dubé et al., 2006). Relatively lower cost continuous-wave (cw) diode laser modules that are easily modulated also have been popular choices (e.g., (King et al.,

2000;Simpson, 2003;Ayers et al., 2005;Odame-Ankrah and Osthoff, 2011;Wagner et al., 2011)). In a CEAS instrument (also referred to as integrated cavity output spectroscopy, ICOS or cavity enhanced DOAS, CE-DOAS), the spectrum transmitted through a high-finesse optical cavity is recorded. Mixing ratios of the absorbing gases are derived using spectral retrieval routines similar to those used for open-path DOAS (e.g., (O'Keefe, 1998;O'Keefe et al., 1999;Ball et al.,

2001;Fiedler et al., 2003;Platt et al., 2009;Schuster et al., 2009)).

CRDS and CEAS are, in principle, absolute measurement techniques and do not need to rely on external calibration. In practice, however, chemical losses can occur on the inner walls of the inlet (even when constructed from inert materials such as Teflon) or at the aerosol filters necessary for CRDS instruments. Hence the inlet transmission efficiencies have to be monitored for

measurements to be accurate (Fuchs et al., 2008;Odame-Ankrah and Osthoff, 2011;Fuchs et al., 2012). On the other hand, a key advantage of in situ instruments over open-path instruments is that the sampled air can be manipulated. Deliberate addition of excess NO to the instrument's inlet titrates $NO_3$ and allows measurement of the instrument's zero level and separation of contributions to optical extinction from other species, such as $NO_2$, $O_3$ and $H_2O$. Adding a heated

section to the inlet (usually in a second detection channel) enables (parallel) detection of $N_2O_5$ via the increase in the $NO_3$ signal (Brown et al., 2001;Simpson, 2003).

In addition, non-optical techniques have been used to detect and quantify $NO_3$. Chemical ionization mass spectrometry (CIMS) is a powerful method for sensitive, selective, and fast quantification of a variety of atmospheric trace gases (Huey, 2007). $NO_3$ is readily detected after

reaction with iodide reagent ion as the nitrate anion at *m/z* 62; at this mass, however, there are several known interferences, including dissociative generation from $N_2O_5$, $HNO_3$, and $HO_2NO_2$ (Slusher et al., 2004;Abida et al., 2011;Wang et al., 2014). There has been more success with the quantification of $N_2O_5$, usually as the iodide cluster ion at *m/z* 235 (Kercher et al., 2009), though accurate $N_2O_5$ measurement at *m/z* 62 has been reported from recent aircraft measurements with

large $N_2O_5$ signal (Le Breton et al., 2014).

Two groups have used laser induced fluorescence (LIF) to quantify $NO_3$ (and $N_2O_5$ through thermal dissociation) in ambient air (Wood et al., 2003;Matsumoto et al., 2005a;Matsumoto et al.,





2005b). The major drawback of this method is the relatively low fluorescence quantum yield of $NO_3$, and hence the method has not gained wide use.

Another technique that was demonstrated to be capable of measuring $NO_3$ radicals at atmospheric concentration is matrix isolation electron spin resonance (MIESR) (Geyer et al., 1999). Although

5   the technique allows simultaneous detection of other radicals (including $HO_2$ and $NO_2$), it has not been used extensively, probably because of its complexity.

Recently, a variety of in situ $NO_3$ (Dorn et al., 2013) and $N_2O_5$ (Fuchs et al., 2012) measurement techniques were compared at the SAPHIR chamber in Juelich, Germany. All instruments measuring $NO_3$ were optically based (absorption or fluorescence). $N_2O_5$ was detected as $NO_3$

10   after thermal decomposition in a heated inlet by either CRDS or LIF. Generally, agreement within the accuracy of instruments was found for all techniques detecting $NO_3$ and/or $N_2O_5$ in this comparison exercise. This study showed excellent agreement between the instruments on the single digit ppt $NO_3$ and $N_2O_5$ levels with no noticeable interference due to $NO_2$ and water vapor for instruments based on cavity ring-down or cavity enhanced spectroscopy. Because of the low

15   sensitivity of LIF instruments, $N_2O_5$ measurements by these instruments were significantly noisier compared to the measurements by cavity enhanced methods. The agreement between instruments was less good in experiments with high aerosol mass loadings, specifically for $N_2O_5$ presumably due to enhanced, unaccounted loss of $NO_3$ and $N_2O_5$ demonstrating the need for regular filter changes in closed cavity instruments.

**Table 3**: Selected CRDS and CEAS instruments used to quantify $NO_3$ mixing ratios in ambient air.

| Reference | Principle of measurement (laser pulse rate) | LOD or precision (integration time) |
|---|---|---|
| (Ball et al., 2004) | BB-CEAS | 2.5 pptv (8.6 min) |
| (Bitter et al., 2005) | BB-CRDS | 1 pptv (100 s) |
| (Ayers and Simpson, 2006) | off-axis cw CRDS (500 Hz) | 2 pptv (5 s) |



| Reference | Principle of measurement | LOD or precision (integration time) |
|---|---|---|
| (Dubé et al., 2006) | on-axis pDL-CRDS (33 Hz) | < 1 pptv (1 s) |
| (Venables et al., 2006) | BB-CEAS | 4 pptv (60 s) |
| (Nakayama et al., 2008) | pDL-CRDS (10 Hz) | 2.2 pptv (100 s) |
| (Schuster et al., 2009;Crowley et al., 2010) | off-axis cw CRDS (200 Hz) | 2 pptv (5 s) |
| (Platt et al., 2009;Meinen et al., 2010) | CE-DOAS | 6.3 pptv (300 s) |
| (Langridge et al., 2008;Benton et al., 2010) | BB-CEAS | 2 pptv (15 s) |
| (Kennedy et al., 2011) | BB-CEAS | < 2 pptv (1s) |
| (Wagner et al., 2011) | on-axis cw-CRDS (500 Hz) | < 1 pptv (1 s) |
| (Odame-Ankrah and Osthoff, 2011) | on-axis cw-CRDS (300 Hz) | 8 pptv (10 s) |
| (Le Breton et al., 2014) | BB-CEAS | 1 pptv (1 s) |
| (Wu et al., 2014) | BB-CEAS | 7.9 pptv (60 s) |

CEAS = cavity enhanced absorption spectroscopy; CRDS = cavity ring-down spectroscopy; BB = broadband; pDL = pulsed dye laser; CE-DOAS = cavity-enhanced differential optical absorption spectroscopy; cw = continuous wave diode laser

5    **Table 4**: Selected instruments used to quantify $NO_3$ and $N_2O_5$ mixing ratios in ambient air not using cavity-enhanced absorption spectroscopy.

| Reference | Principle of measurement | LOD or precision (integration time) | Species detected |
|---|---|---|---|



| (Geyer et al., 1999) | MIESR | < 2 pptv (30 min) | $NO_3$ |
| (Slusher et al., 2004) | CIMS | 12 pptv (1 s) | $NO_3 + N_2O_5$ |
| (Matsumoto et al., 2005a;Matsumoto et al., 2005b) | LIF | 11 pptv (10 min) | $NO_3$ |
| (Wood et al., 2005) | LIF | 28 pptv (10 min) | $NO_3$ |
| (Zheng et al., 2008) | CIMS | 30 pptv (30 s) | $N_2O_5$ |
| (Kercher et al., 2009) | CIMS | 5 pptv (1 min) | $N_2O_5$ |
| (Le Breton et al., 2014) | CIMS | 7.4 pptv ( 1 s) | $N_2O_5$ |
| (Wang et al., 2014) | CIMS | 39 pptv (6 s) | $N_2O_5$ |

MIESR = Matrix Isolation Electron Spin Resonance; CIMS = chemical ionization mass spectrometry; LIF = laser induced fluorescence

### 2.4.2 Gas phase organic nitrate measurements

Analytical techniques to detect gaseous organic nitrates have been documented in a recent review by Perring et al. (2013). Sample collection techniques for organic nitrates include preconcentration on solid adsorbents (Atlas and Schauffler, 1991;Schneider and Ballschmiter, 1999;Grossenbacher et al., 2001), cryogenic trapping (Flocke et al., 1991) or collection in stainless steel canisters (Flocke et al., 1998;Blake et al., 1999) or direct sampling (Day et al., 2002;Beaver et al., 2012).

The approaches to the analysis of the organic nitrates fall into three broad categories. First, one or more chemically-speciated organic nitrates are measured by a variety of techniques including liquid chromatography (LC) (Kastler et al., 2000) or gas chromatography (GC) with electron capture detection (Fischer et al., 2000), GC with electron impact or negative-ion-chemical ionization mass spectrometry (GC-MS) (Atlas, 1988;Luxenhofer et al., 1996;Blake et al., 1999;Blake et al., 2003a;Blake et al., 2003b;Worton et al., 2008), GC followed by conversion to NO and chemiluminescent detection (Flocke et al., 1991;Flocke et al., 1998), GC followed by photoionization mass spectrometry (Takagi et al., 1981), GC followed by conversion of organic nitrates to $NO_2$ and luminol chemiluminescent detection (Hao et al., 1994), CIMS (Beaver et al.,





2012;Paulot et al., 2012) and proton transfer reaction MS (PTR-MS) (Perring et al., 2009). Second, the sum of all organic nitrates can be measured *directly* by thermal dissociation to $NO_2$, which is subsequently measured by LIF (TD-LIF) (Day et al., 2002), CRDS (TD-CRDS) (Paul et al., 2009;Thieser et al., 2016) or cavity attenuated phase shift spectroscopy (TD-CAPS) (Sadanaga et al., 2016). Finally, the sum of all organic nitrates can be measured *indirectly* as the difference between all reactive $NO_x$ except for organic nitrates and total oxidized nitrogen ($NO_y$) (Parrish et al., 1993).

Recent advances in adduct ionization, in which the charged cluster of the parent reagent ion with the compound of interest is detected, coupled to high resolution time-of-flight (HR-ToF) mass spectrometers is particularly advantageous owing to its soft ionization approach, thereby minimizing fragmentation and allowing identification of molecular composition. Multifunctional organic nitrates resulting from the oxidation of BVOC have been detected using $CF_3O^-$ (Bates et al., 2014;Nguyen et al., 2015;Schwantes et al., 2015;Teng et al., 2015) and iodide as reagent ions (Lee et al., 2014a;Xiong et al., 2015;Lee et al., 2016;Nah et al., 2016b;Xiong et al., 2016).

### 2.4.3 Online analysis of particulate matter

Total (organic + inorganic) mass of particulate nitrates is routinely quantified using on-line AMS (Jayne et al., 2000;Allan et al., 2004), from which the mass of organic nitrates can be obtained by three techniques. First, the $NO^+/NO_2^+$ ratio (or $NO_2^+/NO^+$ ratio) in the mass spectra is used to distinguish organic from inorganic nitrates (Fry et al., 2009;Farmer et al., 2010;Fry et al., 2013;Xu et al., 2015b;Kiendler-Scharr et al., 2016). It is noted that the $NO_2^+/NO^+$ approaches zero in the case of low or non-existent $NO_2^+$ signal, while $NO^+/NO_2^+$ gives large numbers. Second, positive matrix factorization (PMF) in data matrices including the $NO^+$ and $NO_2^+$ ions in addition to organic ions (Sun et al., 2012;Hao et al., 2014;Xu et al., 2015b) is used. Third, the particulate inorganic nitrate concentration as measured by an independent method such as ion chromatography is subtracted from the total particulate nitrate concentration (Schlag et al., 2015;Xu et al., 2015a;Xu et al., 2015b). A detailed comparison of these three methods is presented in Xu et al. (2015b). As the $NO^+/NO_2^+$ ratio in AMS data is dependent on instruments and the types of nitrates (inorganic, organic nitrates from different VOC oxidations), different strategies were developed when using this method to estimate particulate organic nitrates (Fry et al., 2013;Xu et al., 2015b).

A specialized inlet that selectively scrubs gaseous organic nitrates or collects particulate mass on a filter has been coupled to some of the techniques summarized in section 2.4.3 and utilized to





observe particulate organic nitrates in the ambient atmosphere and laboratory studies. A TD-LIF equipped with a gas-scrubbing denuder (Rollins et al., 2010;Rollins et al., 2012) and the filter inlet for gases and aerosols (FIGAERO) (Lopez-Hilfiker et al., 2014) at the front end of an iodide-adduct HR-ToF-CIMS are examples (Lee et al., 2016;Nah et al., 2016b).

### 2.4.4 Offline analysis of particulate matter

Owing to its ability to analyze polar organic compounds without a prior derivatization step, liquid chromatography coupled to MS (HPLC/MS) is well suited for the characterization of SOA compounds originating from the reactions of BVOC and $NO_3$. Unlike in GC/MS methods, a soft ionization technique such as electrospray ionization (ESI) is utilized to ionize target analytes in the LC/MS technique. In the ESI/MS, target analytes are detected as a cation adduct of a target analyte (e.g. $[M + H]^+$ or $[M + Na]^+$) for a positive mode or a deprotonated form of a target analyte ($[M - H]^-$) for a negative mode. As a biogenic SOA compound typically bears a functional group, such as a carboxylic group or a sulfate group, that easily loses a proton, the negative mode ESI ((-)ESI) is commonly applied to detect SOA compounds. High-resolution MS such as TOF or Fourier transform ion cyclotron (FTICR) MS is commonly used to assign chemical formulas for SOA compounds unambiguously. While the LC or direct infusion (-)ESI-MS techniques have been applied successfully for the detection of the oxidation products from $NO_3$-BVOC reactions, the techniques have been less successful in quantifying these compounds, mainly due to the lack of authentic standard compounds. The synthesis of these compounds should be a priority for future studies.

The LC/(-)ESI-MS technique played a crucial role in relating the formation of organosulfates (OS) and nitrooxy-organosulfates (NOS) to $NO_3$-initiated oxidation of BVOC in laboratory-generated and ambient SOA. Since these earlier works, a number of studies have reported the presence of OS and/or NOS compounds in ambient samples (Table SI 4), though most studies do not connect these compounds explicitly to the $NO_3$ oxidation of BVOC. It should be noted that the direct infusion (-)ESI-MS technique rather than LC/(-)ESI-MS is often used for the analysis of fog, rainwater, and cloud water samples as diluted liquid water samples can be injected into the ion source directly without a sample pre-treatment procedure. However, caution is warranted for the direct infusion technique because it cannot separate isobaric isomers and it is susceptible to ion suppression, especially from the presence of inorganic ions in the samples.

Whilst the LC or direct infusion (-)ESI-MS techniques have been successfully applied for the detection of the oxidation products from $NO_3$-BVOC reactions, the techniques have been less





successful in quantifying these compounds mainly due to the lack of authentic standard compounds. The synthesis of these compounds should be a priority for future studies.

Finally, total organic nitrate functional groups within the particle phase have been quantified in ambient air using FTIR of particles collected on ZnSe impaction disks (low pressure cascade impactor, size segregated) or Teflon filters ($PM_1$) (Mylonas et al., 1991;Garnes and Allen, 2002;Day et al., 2010). The organic nitrate content of particles can be quantified offline as well by collection on quartz fiber filters, extraction into solution (e.g., with water:acetonitrile mixtures) and analysis using standard wet chemistry techniques such as high pressure liquid chromatography coupled to electrospray ionization mass spectrometry (HPLC-ESI-MS) (Angove et al., 2006;Perraud et al., 2010;Draper et al., 2015).

## 2.5 Field observations

This section surveys the current literature on field observations of nitrate radicals and BVOC (section 2.5.1), and organic nitrate aerosol attributable to $NO_3$-BVOC chemistry (section 2.5.2).

## 2.5.1 Nitrate radicals and BVOC

A few years after the first measurement of tropospheric $NO_3$ (Noxon et al., 1980;Platt et al., 1980), it was recognized that the nitrate radical is a significant sink for BVOC, especially monoterpenes in terrestrial ecosystems and dimethyl sulfide (DMS) in maritime air influenced by continental $NO_x$ sources (Winer et al., 1984). The conclusion was based upon computer simulations using $NO_3$ concentrations measured in field studies in western USA and Europe, and measured rate constants of $NO_3$ with olefins. The scenarios in these simulations showed very low monoterpene concentrations in the early morning that were directly attributable to BVOC reactions with $NO_3$. An analysis of $NO_3$ formation rates at several urban and rural sites in Scandinavia (Ljungström and Hallquist, 1996) resulted in the conclusion that while nighttime urban loss of $NO_3$ is dominated by reaction with NO, the loss in rural regions is likely dominated by reactive hydrocarbons, especially monoterpenes.

Due to the fast reactions of $NO_3$ with BVOC, lifetimes of $NO_3$ in biogenically-influenced environments can be very short, making simultaneous detection of VOC and $NO_3$ in biogenic regions very difficult. For this reason, several studies have inferred levels of $NO_3$ and its role in processing BVOC using observational analysis and supporting modeling. In particular, the rapid decay of isoprene after sunset has received considerable attention. Measurements of BVOC ~ 1-2 m above canopy level in a Loblolly pine plantation in Alabama during the 1990 ROSE program



(Goldan et al., 1995) were used to infer a nighttime $NO_3$ mixing ratio of only 0.2 ppt and $NO_3$ lifetime of only 7 s due to high levels of monoterpenes. The 4 hr decay time of isoprene after sunset could not be accounted for by gas reactions with $NO_3$ and $O_3$ although the decrease in the α-/β-pinene ratio at night was consistent with known $NO_3$ and $O_3$ chemistry. As part of the North

American Research Strategy for Tropospheric Ozone - Canada East (NARSTO-CE) campaign, measurements of BVOC were made in Nova Scotia in a heavily forested region (Biesenthal et al., 1998). A box-model simulation based on the observational analysis found that the short lifetime of isoprene at night (τ=1-3 hours) could not be explained by the $NO_3$ radical, which was estimated to be 0.1 ppt maximum at night due to low $NO_x$ and $O_3$ levels and high monoterpene

emissions. When OH yields from ozonolysis of BVOC were included in the model, this nighttime OH oxidant could partially account for the isoprene decay. During the Southern Oxidants Study (SOS) campaign in Nashville, TN (Starn et al., 1998), a chemical box model was used to show that rapid nighttime decays of isoprene were consistent with simulated $NO_3$ but only when the site was impacted by urban $NO_x$ emissions. During the PROPHET study, measurements of VOC

were made in a mixed forest approximately 10 m above the canopy surface (Hurst et al., 2001). Isoprene decays at night had an average lifetime of ~2.7 hrs. Box modeling showed that $O_3$ reactions as well as dry deposition were insufficient to account for the decay and that the $NO_3$ radical was a significant sink only after the majority of isoprene had already decayed. On some nights, oxidation by OH could account for all the decay but the decay rates were over predicted.

The authors concluded that vertical transport of isoprene-depleted air aloft contributes to the fast initial decay of isoprene, followed by nighttime OH, $NO_3$ and $O_3$ chemistry decay. Steinbacher et al. (2005) reported on surface measurements in the Po valley at a site 200-300 m from the closest edge of a deciduous forest. Bimodal diurnal cycles of isoprene were observed with morning and evening maxima that were reproduced by a Eulerian model. Isoprene decay lifetimes of 1-3 hrs

were partially explained by $NO_3$ decay, although a dynamic influence on isoprene decrease seemed to be likely including horizontal and vertical dispersion. During the HOHenpeissenberg Photochemistry Experiment (HOHPEX) field campaign, BVOC were analyzed via 2-D GC at a site located on a hilltop above adjacent rural agricultural/forested area that is frequently in the residual layer at night (Bartenbach et al., 2007). For the reactive monoterpenes, a significant non-

zero dependency of the concentration variability on lifetime was found, indicating that chemistry (as well as transport) was playing a role in determining the ambient VOC concentrations. The nighttime analysis gave an estimate of the $NO_3$ mixing ratio of 6.2 +/- 4.2 ppt, indicating it was a significant chemical factor in depletion of monoterpenes.



While the studies above made indirect conclusions about the role of $NO_3$ in BVOC processing, field studies including direct measurements of $NO_3$ are key to confirming the above findings. Golz et al. (2001) reported measurements of $NO_3$ by long path DOAS at an eucalyptus forest site in Portugal during the FIELDVOC94 campaign in 1994. The DOAS beam passed directly over

the canopy at heights of 15 m and 25 m and as a result, they were unable to measure $NO_3$ above the 6 ppt instrumental detection limit despite $NO_3$ production rates of 0.4 ppb hr$^{-1}$. Rapid reaction with BVOC limited the $NO_3$ lifetime to approximately 20 s such that $NO_3$ reactions dominated other indirect losses such as heterogeneous $N_2O_5$ uptake. Simultaneous measurements of $NO_3$ and VOC during the Berliner Ozonexperiment (BERLIOZ) campaign in 1998 allowed one of the first

assessments of the $NO_3$ budget in comparison to OH and $O_3$ oxidants (Geyer et al., 2001b). Surface measurements at this semi-rural location close to forests found the $NO_3$ radical above detection limit (2.4 ppt) on 15 of 19 nights with a maximum of 70 ppt, a steady state lifetime ranging from 20 s to 540 s and $N_2O_5$ ranging from 2-900 ppt. The two most significant losses of $NO_3$ were found to be its direct reaction with olefins (monoterpenes dominating) and indirect loss

due to heterogeneous hydrolysis of $N_2O_5$. Over the study it was possible for the first time to quantify the relative contribution of the $NO_3$ radical to oxidation of VOCs as 28% (24 hr) and 31% for olefinic VOCs (24 hr) compared to the total oxidation via $NO_3$, OH and $O_3$. As part of the 1999 SOS study, $NO_3$, isoprene and its oxidation products were measured at a suburban forested site in Nashville, TN (Stroud et al., 2002). The nitrate radical measured at multiple beam

heights by DOAS had maximum mixing ratios of 100 ppt that were generally found to anticorrelate with isoprene levels with significant vertical gradients on some nights. Early evening losses of isoprene were attributable to reaction with the $NO_3$ radical. During the Pacific 2001 Air Quality Study (PACIFIC 2001) field campaign, $NO_3$ was measured by long path DOAS at an elevated forested site in the Lower Fraser Valley of British Columbia with beam path

nighttime $NO_3$ levels up to a maximum of 50 ppt (average of NBL and residual layer) (McLaren et al., 2004). Simultaneous analysis of carbonyl compounds in aerosol samples (Liggio and Mclaren, 2003) during the study found that only monoterpene oxidation products pinonaldehyde and nopinone (not reported) were enhanced in aerosol filters collected at night, evidence of the role of $NO_3$ in nighttime oxidation of BVOC in the valley. In 2004 measurements of $NO_3$ and

$N_2O_5$ by CRDS, isoprene and its oxidation products were made on board the NOAA P-3 aircraft as part of the New England Air Quality Study (NEAQS) and International Consortium for Atmospheric Research on Transport and Transformation (ICARTT) campaigns in northeast US (Brown et al., 2009). These studies found a very clear anti-correlation between isoprene levels



after dark and NO$_3$ mixing ratios, which varied as high as 350 ppt when isoprene was absent from the air mass. The loss frequencies (i.e., first order loss rate coefficients) of NO$_3$ were strongly correlated with the loss rate coefficient of NO$_3$ with isoprene for lifetimes less than 20 minutes, clearly showing that isoprene was the most important factor determining the lifetime of NO$_3$. It

was also shown that greater than 20% of emitted isoprene was oxidized at night and that 1-17% of SOA was contributed by NO$_3$-isoprene oxidation. A number of recent studies have also investigated the role of NO$_3$ + BVOC chemistry in more polluted areas. In many urban areas the NO$_3$ + BVOC chemistry occurs in parallel to heterogeneous NO$_3$/N$_2$O$_5$ chemistry and reactions of NO$_3$ with anthropogenic VOC. Examples of such environments have been discussed in Brown et

al. (2011;2013) and Stutz et al. (2010) who presented observations in Houston, TX. Brown et al. (2011) and Stutz et al. (2010) found that up to 50% of the NO$_3$ + VOC reactions in Houston are driven by isoprene, with the other VOC emitted by industrial sources. Surprisingly, heterogeneous NO$_3$/N$_2$O$_5$ chemistry plays a minor role in Houston. Brown et al. (2011) also point out that the nocturnal VOC oxidation by NO$_3$ dominates over that from ozone. Nocturnal NO$_3$

formation rates were rapid and comparable to those of OH during the day. Crowley et al. (2011) compares NO$_3$ chemistry in air masses of marine, continental and urban origin at a field site in southern Spain. Under all conditions, NO$_3$ + BVOC reactions (predominately α-pinene and limonene) contributed to the overall NO$_3$ reactivity, confirming other observations that concluded that this chemistry is important in all environments where BVOC sources are present. In the

southeastern U.S. summer, this importance extends even through the daytime, when photolysis and NO reactions compete (Ayres et al., 2015). The NO$_3$ + BVOC reaction rates observed in these studies imply a high production rate of SOA and organic peroxy radicals.

### 2.5.2 Organic nitrate aerosols

There are many factors that motivate understanding organic nitrate in the particulate phase

through field deployment of a variety of instrumentation, much of which is described in other sections of this review. Nitrogen-containing organic fragments (not necessarily organic nitrates) have been identified in atmospheric particles using mass spectrometric techniques (Reemtsma et al., 2006;Farmer et al., 2010;O'Brien et al., 2014). Total atmospheric organic nitrates, as well as organic nitrates segregated by phase, also have been measured in the atmosphere using techniques

such as TD-LIF, CIMS, etc. (Day et al., 2003;Beaver et al., 2012). Given these observations and the propensity of organic nitrate compounds to partition to the condensed phase to create SOA (Rollins et al., 2013), it is critical to determine the level of organic nitrates that reside specifically


in the atmospheric aerosol phase under typical ambient conditions and to identify the chemical and physical processes that determine their concentrations. It is also important to note that formation of SOA that contains organic nitrate groups has the potential to sequester $NO_x$, thereby influencing the cycling of atmospheric oxidants.

Organic nitrates in urban PM that were identified using functional group analyses such as FTIR spectroscopy have been attributed to emission of nitrogen-containing primary organic aerosol or to involvement of reactive nitrogen compounds in SOA formation chemistry (Mylonas et al., 1991;Garnes and Allen, 2002;Day et al., 2010). Other more advanced techniques, such as TD-LIF enhanced with the ability to separate phases or techniques to obtain high resolution mass spectra

(HR-ToF-AMS) have been utilized to quantify the amount of organic nitrate in particles in areas less likely to be influenced strongly by BVOC emissions, such as urban areas or areas influenced by oil and gas operations (Lee et al., 2015). Of specific interest here, however, are observations of organic nitrate PM in areas with a significant influence of BVOC, especially if co-located measurements allow for insight into the role that $NO_3$ plays in the initial BVOC-oxidation step.

As such we focus here on on-line measurements and on measurements that allow specific attribution to BVOC-$NO_3$ reactions. Such measurements broadly can be categorized by region of sampling: the Eastern United States (US), the Western US, and Europe.

### Eastern United States

The first reports of aerosol organic nitrates in the southeastern (SE) US resulted from composition

analysis of four daily PM filter samples from four Southeastern Aerosol Research and Characterization (SEARCH) network sites during summer 2004. Filters were analyzed for polar compounds, with particular focus on organosulfates, using off-line chromatographic–MS methods (Gao et al., 2006;Surratt et al., 2007;Surratt et al., 2008). Several nitrooxy organosulfates were identified, but the only one quantified (1-2% of organic mass) was associated with α-pinene

photooxidation or reaction with $NO_3$. Several of the nitrooxy organosulfates were likely the same as products from BVOC-oxidant-$NO_x$-seed systems based on comparison to spectra collected from chamber studies.

Brown et al. (2013) examined several nighttime aircraft vertical profiles in Houston (October 2006 during the Texas Air Quality Study 2006) that showed increases of total nitrate aerosol (and

increases in HR-ToF-AMS $NO^+/NO_2^+$, indicative of organic nitrates (Farmer et al., 2010)) and oxygenated organic aerosol (OOA). The OA versus carbon monoxide (CO) slopes at lower altitudes were consistent with SOA sources from $NO_3$-BVOC reactions, with a combination of




observations and zero-dimensional modeling showing 1 to 2 $\mu g \ m^{-3}$ SOA formation from $NO_3$-BVOC oxidation overnight with formation rates of 0.05 to 1 $\mu g \ m^{-3} \ h^{-1}$.

More recently, during the summer Southern Oxidant and Aerosol Study (SOAS; mixed, semi-polluted forest) in Alabama (2013), an unprecedented suite of instruments quantified particle-phase organic nitrates using five different online methods: HR-ToF-AMS ($NO^+/NO_2^+$), HR-ToF-AMS – PILS-ion chromatography (IC), HR-ToF-AMS (PMF), TD-LIF (denuded), and iodide-CIMS. Total particle-phase nitrates increased throughout the night and peaked in early/mid-morning. Xu et al. (2015b) systematically evaluated the three AMS-related methods in estimating ambient particulate organic nitrate concentrations. Analysis presented in Xu et al. (2015a)  using the HR-ToF-AMS – PILS-IC method showed that organic nitrate functional groups comprised ~5-12% of OA mass (Xu et al., 2015b) and correlated with PMF-derived less-oxidized oxygenated OA (LO-OOA). Two-thirds of the LO-OOA was estimated to be formed via $NO_3$-BVOC chemistry (dominantly monoterpenes, ~80%), with the balance due to ozone ($O_3$)-BVOC chemistry. Organic nitrates were calculated to comprise 20-30% of the LO-OOA factor. Ayres et al. (2015) used a measurement-constrained model for nighttime that compared $NO_3$ production/loss to total organic nitrate (HR-ToF-AMS $NO^+/NO_2^+$, TD-LIF) formation to calculate a molar yield of aerosol-phase organic nitrates of 23-44% (organic nitrate formed per $NO_3$-BVOC reaction) that was dominated by monoterpene oxidation. They noted that the estimated yield was low compared to aggregated aerosol-phase organic nitrate yields, possibly due to rapid nitrate losses not considered in the model.  Organic nitrate hydrolysis in the particle phase is one potential loss pathway, although recent laboratory studies suggest this process is slow for $NO_3$ + β-pinene SOA (Boyd et al., 2015). Also, particle-phase organic nitrates were observed to contribute 30-45% to the total $NO_y$ budget. (Lee et al., 2016) quantified speciated particle-phase organic nitrates using iodide-CIMS (88 individual $C_4$-$C_{17}$ mono/di-nitrates). A large fraction was highly functionalized, with six to eight oxygen atoms per molecule. Diurnal cycles of isoprene-derived organic nitrates generally peaked during daytime and monoterpene-derived organic nitrates peaked during night or early/mid-morning. Using an observationally-constrained diurnal zero-dimensional model, they showed that the observations were consistent with fast gas/particle equilibrium and a short particle-phase lifetime (2-4 hours), again possibly due to hydrolysis if the field-derived lifetimes for particle phase organic nitrates can be reconciled with recent laboratory studies (Boyd et al., 2015). The sum of the CIMS particle-phase organic nitrates (mass of nitrate functional groups only) was correlated with the two total aerosol organic nitrate AMS-based methods ($R^2$=0.52, 0.67) with slopes of 0.63 and 0.90 (Lee et al., 2016). The CIMS sum was also





correlated with the total measured with the TD-LIF method (R2=0.55); however, since the TD-LIF measurements were ~2-4 times higher (depending on period) than the AMS-based methods, the CIMS vs TDLIF slope was substantially lower (0.19). Reasons for the differences between the total organic nitrate measured by different methods have been investigated but remain unclear.

5   A seasonal and regional survey of particle-phase organic nitrates is reported by Xu et al. (2015b) using a HR-ToF-AMS and an Aerosol Chemical Speciation Monitor (ACSM) (Ng et al., 2011) at four rural and urban sites in the greater Atlanta area (2012-2013) and in Centreville, AL (summer 2013 only, SOAS). They show strong diurnal cycles during summer, peaking early/mid-morning, and cycles with similar timing but smaller magnitude during winter. The concentrations were slightly higher in summer, which was attributed to compensating effects of source strength and gas/particle partitioning. Shallower boundary layers during winter also may have played a role in making the summer and winter concentrations more similar (Kim et al., 2015).

Fisher et al. (2016) report a broad regional survey of particle-phase (and gas-phase) organic nitrates (HR-ToF-AMS $NO^+/NO_2^+$) during summertime for the Studies of Emissions and Atmospheric Composition, Clouds and Climate Coupling by Regional Surveys (SEAC[4]RS) aircraft campaign (August-September, 2013, SE US only) as well as the ground-based SOAS measurements. A substantial vertical gradient was observed in particle-phase organic nitrates, with concentrations decreasing by several-fold from the boundary/residual layer into the free troposphere. Consistent with SOAS ground observations, 10-20% of observed boundary layer total (gas plus particle) organic nitrates were in the particle phase for the aircraft measurements.

In addition to the measurements made in the SE US, characterization of aerosol organic nitrates has been performed in New England. As part of the New England Air Quality Study (NEAQS) in summer 2002, (Zaveri et al., 2010) observed evolution of aerosols in the nocturnal residual layer with an airborne quadrupole (Q)-AMS in the Salem Harbor power plant plume. The aerosols were acidic and internally mixed, suggesting that the observed nitrate was in the form of organic nitrate and that the enhanced particulate organics in the plume was possibly formed from $NO_3$-initiated oxidation of isoprene present in the residual layer.

***Western United States***

Significant work on understanding ambient organic nitrate formation from BVOC-$NO_3$ has been performed in California. During the California Research at the Nexus of Air Quality and Climate Change (CalNex) field campaign from mid-May through June 2010, Rollins et al. (Rollins et al., 2012;Rollins et al., 2013) measured particulate total alkyl and multifunctional nitrates (pΣAN) with TD-LIF at a ground site in Bakersfield, California. They attributed the increase in pΣAN



concentrations at night to oxidation of BVOC by $NO_3$ forming SOA, with an estimated 27 to 40% of the OA growth due to molecules with nitrate functionalities. On average, 21% of ΣANs were in the particle-phase and increased with OA, which was fit to a volatility basis set in which pΣANs/ΣANs increased from ~10% at < 1 µg m$^{-3}$ and plateaued at ~30% by ~5 µg m$^{-3}$. At the

same site, using PMF analysis of FTIR and HR-ToF-AMS measurements, Liu et al. (2012a) showed that the organic nitrate-containing biogenic SOA condensed onto 400 to 700 nm-sized primary particles at night. As part of the Carbonaceous Aerosol and Radiative Effects Study (CARES) in June 2010, Setyan et al. (2012) observed enhanced SOA formation due to interactions between anthropogenic and biogenic emissions at a forest site in the foothills of the

Sierra Nevada Mountains, approximately 40 km downwind of Sacramento. While nitrate accounted for only ~4% of the particle mass measured by a HR-ToF-AMS, it was attributed potentially to organic nitrates based on the much higher $NO^+/NO_2^+$ ion ratio than observed in pure ammonium nitrate.

During the Rocky Mountain Biogenic Aerosol Study field campaign in Colorado's Front Range

(rural coniferous montane forest) (BEACHON-RoMBAS) from July to August 2011, Fry et al. (2013) observed aerosol-phase organic nitrates by optical spectroscopic (denuded TD-LIF) and mass spectrometric (HR-ToF-AMS, $NO^+/NO_2^+$) instruments. The two methods agreed well on average (AMS/TD-LIF slope of 0.94-1.16, depending on averaging method) with a fair correlation ($R^2$=0.53). Similar to studies in other forested environments, the organic nitrate

concentration was found to peak at night. The organic nitrate concentration was positively correlated with the product of the nitrogen dioxide and $O_3$ mixing ratios but not with that of $O_3$ alone; this suggested nighttime $NO_3$-initiated oxidation of monoterpenes as a significant source of nighttime aerosol organic nitrates. The gas/particle partitioning also showed a strong diurnal cycle, with the fraction in the particle-phase peaking at ~30% at night and decreasing to a broad

minimum of ~5% during daytime, which suggests a change in composition in addition to thermodynamic partitioning effects.

### *Europe*

Iinuma et al. (2007) analyzed ambient aerosol samples collected on filters in a Norway spruce forest in northeastern Germany during the BEWA campaign (Regional biogenic emissions of

reactive volatile organic compounds from forests: Process studies, modeling and validation experiments) and compared the results to those from chamber studies. The filter extracts were analyzed using LC-ESI-ToF-MS in parallel to ion trap MS. Several nitrooxy organosulfates with significant mass in the BEWA ambient samples were enhanced in the nighttime samples relative



to the daytime samples. Their abundance in the nighttime samples strongly suggests that $NO_3$-monoterpene chemistry in the presence of sulfate aerosols has an important role in the formation of these nitrooxy organosulfate aerosols.

A similar study by Gómez-González et al. (2008) focused on isoprene through LC-multi-dimensional MS ($MS^n$) analysis of filter samples from both chamber studies and ambient summer day/night $PM_{2.5}$ samples from K-puszta, Hungary, a mixed deciduous/coniferous forest site. Although not the focus of the study, they confirmed the presence of significant quantities of nitrooxy organosulfates that were enhanced in the nighttime samples over the daytime samples.

Initial on-line evidence of the production of organic nitrate aerosols in Europe was provided by Allan et al. (2006) when studying nucleation events driven by BVOC oxidation in Hyytiala, a (boreal) forested region in Finland. The Q-AMS $NO^+/NO_2^+$ ratio was frequently found to be very high, $\sim 10$, for a distinct organic Aitken mode that became apparent late in the afternoon and increased at night. They hypothesized that the excess $NO^+$ signal was associated with organic nitrates, although could not rule out amine contributions. During the same field study, Vaattovaara et al. (2009) applied two tandem differential mobility analyzer methods to study the evolution of the nucleation- and Aitken-mode-particle compositions at this boreal forest site. The results showed a clear anthropogenic influence on the nucleation- and Aitken-mode-particle compositions during the events and suggested organic nitrate and organosulfate aerosol was generated from monoterpene oxidation. Also it was shown that organic nitrate was enhanced in aerosol exposed to elevated temperatures, implying low volatility of organic nitrates (Häkkinen et al., 2012).

More recently, Hao et al. (2014) used a HR-ToF-AMS on a tower in Kuopio, Finland, 224 m above a lake surrounded by a mixed forest of mostly coniferous (pine and spruce) mixed with deciduous trees (mostly birch) to measure submicron aerosol composition. The site also was influenced by urban emissions. A particular focus of the study was to separate organic and inorganic nitrate using PMF. They found that $\sim 37\%$ of the nitrate mass at this location and time could be allocated to organic nitrate factors, the rest being inorganic nitrate. The organic nitrate aerosol was segregated into two organic factors, less-oxidized OOA (LO-OOA) and more-oxidized (MO-OOA) (previously called SV- and LV-OOA, respectively); the majority (74%) of the organic nitrate was found to be in the more volatile LO-OOA factor. Based on meteorology, the air mass source of the organic nitrate aerosol was from a sector with residential and forested areas. Again, the organic nitrate aerosol showed a diurnal trend that was highest at night.



An analysis of AMS data taken across Europe within EUCAARI and EMEP intensive measurement campaigns (Kulmala et al., 2011;Crippa et al., 2014) has recently shown high organic nitrate contributions to total measured $PM_1$ nitrate (Kiendler-Scharr et al., 2016). The spatial distribution and diurnal pattern of particulate organic nitrate indicate a gradient of concentration with high concentration found in source regions, i.e. regions with high $NO_x$ emissions and during night time, and low concentrations in remote regions and during the day. EURAD-IM simulations for Europe show an increase of SOA by 50 to 70% when considering SOA formation by $NO_3$ oxidation with maximum ground level concentrations of SOA from $NO_3$ oxidation in the range of 2 to 4 µg m$^{-3}$ (Li et al., 2013;Kiendler-Scharr et al., 2016).

***Summary of organic nitrate aerosol observations***

Taken together, the observations of particle-phase organic nitrates in the US and Europe suggest that particle-phase organic nitrates (formed substantially via $NO_3$-BVOC chemistry) are ubiquitous, especially in, but not limited to, summer. Their formation appears to play an important role in SOA formation, which can potentially be underestimated due to short particle-phase lifetimes.. Regions with widespread $NO_x$ and BVOC emissions and a humid climate may create optimal conditions for a rapid lifecycle of particle-phase organic nitrates.

## 2.6 Models of $NO_3$-BVOC chemistry

To understand the implications of $NO_3$-BVOC chemistry on atmospheric chemistry as a whole, under both current and future scenarios, the physical and chemical processes, such as those reported in sections 2.1 through 2.3, must be parameterized in numerical models. In this section, we summarize how these reactions are represented in current air quality models (AQMs).

### 2.6.1 Chemical mechanisms

Organic nitrates are produced from the reactions of VOCs with OH followed by NO as well as with $NO_3$, and both of these pathways are represented in chemical mechanisms albeit at varying levels of detail. The use of the term "model" below refers to the treatment of BVOC + $NO_3$ chemistry in lumped chemical mechanisms. The products formed from the OH-initiated (typically daytime) vs $NO_3$-initiated (typically nighttime) chemistry may or may not be treated separately. The $NO_3$-BVOC reactions result in an $RO_2$ that reacts with $NO_3$, other $RO_2$, $HO_2$, or NO. $RO_2$-NO reactions for $NO_3$-initiated chemistry are relatively unimportant due to rapid reaction of NO with $NO_3$ at night (Perring et al., 2009), but they are included in models. Unimolecular rearrangements of the $NO_3$-initiated $RO_2$ radical are not currently considered in models (Crounse



et al., 2011). The products of the initial $NO_3$-BVOC reaction may retain the nitrate group thus forming an organic nitrate or releasing nitrogen as $NO_2$. The branching between organic nitrate formation and N recycling is parameterized in models. Table 5 summarizes the gas-phase organic nitrate yields for isoprene and monoterpene oxidation by $NO_3$ in a number of currently available

chemical mechanisms. The yields represent the first-generation yields since products may react to form further organic nitrates or release $NO_2$. The organic nitrate yield values span from zero (e.g. SAPRC07 isoprene) to 100% (e.g. MCM isoprene). Although GEOS-Chem v10-01 does not consider gas-phase monoterpene chemistry, the model has recently been updated to consider a 10-50% yield of organic nitrates from monoterpene-$NO_3$ reaction independent of the nitrate-$RO_2$ fate

but dependent on monoterpene identity (Fisher et al., 2016). Differences in the organic nitrate yield from $NO_3$ oxidation result from a number of causes including: treatment of $RO_2$ fate, assumptions about decomposition versus retention, and prioritization of functional group identity. Some models parameterize the yield of organic nitrates as a function of $RO_2$ fate while others such as the carbon bond-based (CB) mechanisms treat all $RO_2$ fates the same. The MCM v3.3.1

also considers the yield of isoprene organic nitrates to be independent of $RO_2$ fate, but monoterpene organic nitrate yields are variable between 0 and 100% depending on $RO_2$ fate. Differences in organic nitrate formation due to treating the organic nitrate yield as a function of $RO_2$ fate may be significant based on atmospheric conditions. Reaction with both $HO_2$ and $RO_2$ are significant at night (Xie et al., 2013;Pye et al., 2015). $RO_2$-$NO_3$ may be important in urban

areas or locations where BVOC concentrations are not so high as to deplete $NO_3$ (Rollins et al., 2012).

Mechanisms differ in their assumptions about whether or not the organic nitrates from $NO_3$-initiated chemistry release $NO_2$ or retain the nitrate group. An example of this difference in treatment of organic nitrates can been seen in the reactions of nitrated peroxy radicals with

different radicals (NO, $HO_2$, $RO_2$) predicted by SAPRC07 and MCM. MCM predicts greater loss of the nitrate group, while SAPRC tends to retain it, leading to either <5% (MCM) or >50% (SAPRC) organic nitrate yields.

In order to predict accurately the fates of $RO_2$ and yield of organic nitrates, models must also include information on $RO_2$ reaction rate constants. Some mechanisms use the same set of $RO_2$

rate constants for all hydrocarbons. However, the MCM (Jenkin et al., 1997;Saunders et al., 2003) indicates that the $RO_2$-$HO_2$ rate constant should vary with carbon number (n) and predict $k = 2.91 \times 10^{-13} \exp(1300/T).[1 - \exp(-0.245n)]$ molec$^{-1}$ cm$^3$ s$^{-1}$. The MCM $RO_2$-$RO_2$ rate constant varies between $2 \times 10^{-12}$ molec$^{-1}$ cm$^3$ s$^{-1}$ (based on $C_1$-$C_3$ primary $RO_2$ with adjacent O or Cl) and $6.7 \times 10^{-}$



$^{15}$ molec$^{-1}$ cm$^3$ s$^{-1}$ for tertiary alkyl RO$_2$ (based on t-C$_4$H$_9$O$_2$). RO$_2$-NO$_3$ and RO$_2$-NO rate constants are estimated as 2.3x10$^{-13}$ and 9.0x10$^{-12}$ molec$^{-1}$ cm$^3$ s$^{-1}$ at 298 K.

Air quality models (AQMs) and chemistry-climate models typically cannot handle the complexity associated with tracking each individual VOC and all its possible reaction products. As a result,

surrogate species are often used to represent classes of compounds (e.g., NTR in CB05 for organic nitrates). This mapping can cause yields of organonitrates to be falsely low in a mechanism if other functional groups are prioritized over nitrate in the mapping of predicted products to mechanism species. Compared to the other mechanisms in Table 5, SAPRC07 monoterpenes tend to have very low organic nitrate yields as a result of prioritization of peroxide

and non-nitrate functional groups. If nitrate groups were prioritized, SAPRC07 would more closely resemble the "other monoterpene" yields from SAPRC07tic. In addition, the diversity across mechanisms in the RO$_2$-HO$_2$ monoterpene organic nitrate yields would be reduced such that they would all indicate >50% organic nitrate yields and all but the CB mechanisms would predict a 100% yield of organic nitrates from RO$_2$-HO$_2$. The RO$_2$-HO$_2$ pathway is relatively

unstudied in laboratory conditions due to difficulties in maintaining sufficient concentrations of both NO$_3$ and HO$_2$ radicals (Boyd et al., 2015;Schwantes et al., 2015).

**Table 5** Gas-phase organic nitrate yields (in percent) from BVOC+NO$_3$ systems in current chemical mechanisms. Gas-phase organic nitrate yields depend on RO$_2$ fate as indicated in the

ternary diagrams: Clockwise from the top, RO$_2$ reacts with NO$_3$, RO$_2$, and HO$_2$.

| Chemical Mechanism | Gas-Phase Yield of Organic Nitrates from Isoprene+NO$_3$ | Gas-Phase Yield of Organic Nitrates from Monoterpenes+NO$_3$ | References |
|---|---|---|---|
| CB05 | | | (Yarwood et al., 2005) |





| | | | |
|---|---|---|---|
| CB6r2 | | same as CB05 | (Perring et al., 2009;Hildebrandt Ruiz and Yarwood, 2013) |
| GECKO-A | up to 100% (see supporting information) | same as isoprene | (Aumont et al., 2005) |
| GEOS-Chem v10-01 | | NA (monoterpene oxidation is offline) | (Mao et al., 2013) |
| GFDL AM3 | | | |
| MCM v3.3.1 | | α-pinene        β-pinene | (Jenkin et al., 1997;Saunders et al., 2003;Jenkin et al., 2015) |
| MOZART | | | (Emmons et al., 2010) with updates on organic nitrates |
| SAPRC07 | | | (Carter, 2010b;Carter, 2010a) Plots of $RO_2 + RO_2$ based on RO2C |



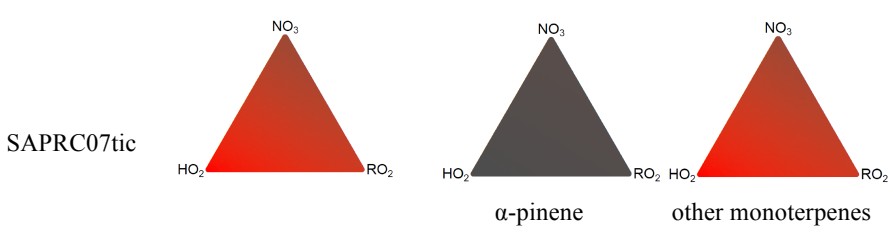

SAPRC07tic — α-pinene — other monoterpenes

(Rollins et al., 2009;Xie et al., 2013)

α-pinene (same as SAPRC07)

Other monoterpenes: (Pye et al., 2015)

### 2.6.2 Influence on organic aerosol

Nitrate radical oxidation can lead to significant amounts of SOA on global and regional scales. Due to a lack of information on the identity and volatility of later-generation BVOC+NO$_3$

products, most models parameterize SOA formation separately from gas-phase chemistry using either and Odum 2-product (Odum et al., 1996) fit, volatility basis set (VBS) (Donahue et al., 2006) fit, or fixed yield (Table 6). Based on the understanding of SOA pathways at the time, Hoyle et al. (2007) found that up to 21% of the global average SOA burden may be due to NO$_3$ oxidation, and Pye et al. (2010) predicted ~10% of global SOA production was due to NO$_3$.

Regional contributions to SOA concentrations can be much higher (Hoyle et al., 2007;Pye et al., 2010). Nitrate radical reactions themselves are estimated to account for less than 3% of isoprene oxidation and less than 2% of sesquiterpene oxidation globally; however, they account for 26% of bicyclic monoterpene oxidation (Pye et al., 2010). Representations of monoterpene-NO$_3$ SOA are more widespread in chemistry-climate models than other BVOC-NO$_3$ SOA parameterizations due

to the relatively early recognition of its high yields (e.g. (Griffin et al., 1999)) and relative importance for SOA. Inclusion of SOA from isoprene-NO$_3$ is more variable as reflected in Table 6.

SOA from BVOC-NO$_3$ reactions traditionally has been parameterized on the initial hydrocarbon reaction assuming semivolatile products and an Odum two-product approach (e.g. (Chung and

Seinfeld, 2002)). This treatment is often implemented in parallel to the gas-phase chemistry, meaning that later generation products leading to SOA are not identified. Information is still emerging on the fate of organic nitrates, and that information is just beginning to be included in models. Hydrolysis of particle-phase organic nitrates is one such process more recently considered with impacts for both O$_3$ and PM in models (Hildebrandt Ruiz and Yarwood,

2013;Browne et al., 2014;Pye et al., 2015;Fisher et al., 2016).



**Table 6** Treatment of SOA formation from BVOC-NO$_3$ systems in current 3-D models.

| Model | Gas-phase Chemistry | Isoprene + NO$_3$ SOA Parameterization | Monoterpene + NO$_3$ Parameterization | References |
|---|---|---|---|---|
| CAMx v6.20 with SOAP | CB05, CB6, or SAPRC99 | no SOA from this path | NO$_3$ SOA yields same as photooxidation (OH+ozone) yields | SOA: (Strader et al., 1999) |
| CAMx v6.20 with 1.5 D VBS | CB05, CB6, or SAPRC99 | NO$_3$ SOA yields same as photooxidation (OH+ozone) yields | NO$_3$ SOA yields same as photooxidation (OH+ozone) yields | SOA: (Koo et al., 2014) |
| CMAQ v5.1 cb05e51-AERO6 | CB05 with additional modification | Odum 2-product based on Kroll et al. 2006 photooxidation (OH) yields | Odum 2-product based on Griffin et al. 1999a photooxidation (OH+ozone) yields | Monoterpene SOA: (Carlton et al., 2010a) |
| CMAQ v5.1 SAPRC07tic-AERO6 | SAPRC07 with 2 monoterpenes: a-pinene (APIN) and other monoterpenes (TERP) | Odum 2-product based on Kroll et al. 2006 photooxidation (OH) yields | Odum 2-product based *on* Griffin et al. 1999a photooxidation (OH+ozone) yields | Chemistry: (Hutzell et al., 2012) Monoterpene SOA: (Carlton et al., 2010a) |
| CMAQ v5.1 SAPRC07tic-AERO6i | SAPRC07tic | based on semivolatile organic nitrate from isoprene dinitrate in (Rollins et al., 2009) | no SOA from a-pinene+NO$_3$, SOA from other monoterpenes based on semivolatile organic nitrates from (Fry et al., 2009) | Isoprene: (Rollins et al., 2009;Xie et al., 2013) a-pinene (same as SAPRC07): (Carter, 2010b) (Carter, 2010a) Other monoterpenes: (Pye et al., 2015) |





| | | | | |
|---|---|---|---|---|
| EURAD-IM | RACM | Odum 2-product based on (Ng et al., 2008) | Odum 2-product based on (Griffin et al., 1999); T-dependent implementation | (Li et al., 2013;Kiendler-Scharr et al., 2016) |
| GEOS-Chem v10-01 | GEOS-Chem v10-01 | VBS fit to *Ng et al.* 2008 experiment | VBS fit to Griffin et al. 1999a b-pinene+$NO_3$ experiment | Isoprene: (Rollins et al., 2009;Xie et al., 2013) VBS SOA fits: (Pye et al., 2010) |
| GFDL AM3 | GFDL AM3 | no SOA from this pathway | Odum 2-product based on Griffin et al. 1999a b-pinene+$NO_3$ | (Chung and Seinfeld, 2002) |
| GISS-GCM II' | NA (offline oxidants) | no SOA from this pathway | Odum 2-product based on Griffin et al. 1999 b-pinene+$NO_3$ | (Chung and Seinfeld, 2002) |
| GLOMAP/ UKESM-1 | VOC+$NO_3$ | Based on *Kroll et al.* experiments (2006), set to 3% | Based on Tunved et al. (2004), set to 13% | (Scott et al., 2014) |
| STOCHEM -CRI | MCM | CRI species fit to MCMv3.1 simulations following approach of Utembe et al., 2009 and Johnson et al., 2006. | CRI species fit to MCMv3.1 simulations following approach of Utembe et al., 2009 and Johnson et al., 2006. | (Utembe et al., 2011;Khan et al., 2015) |
| WRF-Chem V3.6.1 | MOZART-MOSAIC | no SOA from this pathway | VBS fit to Griffin et al. 1999 ß-pinene+$NO_3$ experiment | (Knote et al., 2014) |





### 2.6.3 Influence on reactive nitrogen and ozone

The influence of BVOC nighttime oxidation on the nitrogen budget remains unclear. Current modeling efforts have mainly focused on the nighttime oxidation of isoprene, which is dominated by isoprene-$NO_3$ reaction. This pathway is initialized via addition of $NO_3$ to one of the double

bonds, as discussed in section 2.1.2. Due to the additional stabilization from alkoxy radical and nitrate functional groups (Paulson and Seinfeld, 1992), the yield of first-generation carbonyl nitrates is relatively high (65-85%); they may react with $NO_3$ again to produce secondary dinitrates (Perring et al., 2009;Rollins et al., 2009;Rollins et al., 2012). Assuming little $NO_x$ is recycled from these carbonyl nitrates, most models suggest that nighttime oxidation of isoprene

by $NO_3$ contributes significantly to the budget of organic nitrates (von Kuhlmann et al., 2004;Horowitz et al., 2007;Mao et al., 2013;Xie et al., 2013). Two recent studies (Suarez-Bertoa et al., 2012;Müller et al., 2014), however, suggest fast photolysis of these carbonyl nitrates with high efficiency of $NO_x$ recycling, which could lead to release of $NO_x$ in the next day. Further modeling is required to investigate the importance of nighttime isoprene oxidation on the nitrogen

budget.

Very little modeling effort has been dedicated to the influence of nighttime terpene oxidation on the nitrogen budget, mainly due to the lack of laboratory data on oxidation products and their fate. In contrast to isoprene, terpene emissions are temperature sensitive but not light sensitive (Guenther et al., 1995), leading to a significant portion of terpenes emissions being released at

night. The high yield of organic nitrates and SOA from the terpene-$NO_3$ reaction (Fry et al., 2009;Fry et al., 2011;Fry et al., 2014;Boyd et al., 2015), provides an important sink for $NO_x$ at night, likely larger than for isoprene-$NO_3$ over the eastern US (Warneke et al., 2004). Recent laboratory experiments suggest that aerosol organic nitrates can be either a permanent or temporary $NO_x$ sink depending on their monoterpene precursors (and hence nature of the

resulting $RO_2$) as well as ambient RH (Boyd et al., 2015;Nah et al., 2016b). In order to understand the impact of terpenes on nighttime chemistry, a fully-coupled model of terpene-$NO_x$ chemistry will be required, as monoterpenes can be the dominant loss process for $NO_3$ and $N_2O_5$ at night (Ayres et al., 2015).

While a significant portion of nitrogen is emitted at night (Boersma et al., 2008), the impact of

nighttime chemistry on the initiation of the following daytime chemistry has received little attention in regional and global models. Different treatments of $NO_3$ chemistry can result in 20% change in the following daytime $O_3$ concentration, as shown by a 1-D model study (Wong and



Stutz, 2010) and box model simulations (Millet et al., 2016). This impact can be further complicated by uncertainty in emissions of BVOC and model resolutions. For example, a recent study by Millet et al. (2016) shows that in a city downwind of an isoprene-rich forest, daytime $O_3$ can be largely modulated by the chemical removal of isoprene throughout the night. Such local

scale event may only be captured by a very high resolution model with detailed characterization of emission sources. It is important to assess this impact on a global scale using 3D chemistry models, owing to the profound coupling of boundary layer dynamics and chemistry. Quantifying the impact of BVOC-$NO_3$ chemistry on $NO_x$ fate is important given the long-standing problem in current global and regional AQMs of a large overestimate of $O_3$ over eastern US in summer

(Fiore et al., 2009).

### 2.6.4 Comparison of field data with air quality models

Recent field campaigns (SOAS, SEAC4RS, EUCAARI, EMEP) have allowed for the attribution of SOA to $NO_3$ oxidation to provide model constraints not previously available. Pye et al. (2015) and Fisher et al. (2016) implemented updated BVOC+$NO_3$ chemistry in CMAQ and GEOS-

Chem respectively to interpret data in the southeast U.S. during the summer of 2013 (SOAS and SEAC$^4$RS). Model predictions of gas-phase monoterpene nitrates (primarily $NO_3$ derived) were higher than the sum of $C_{10}H_{17}NO_4$ and $C_{10}H_{17}NO_5$ (Nguyen et al., 2015) by a factor of 2-3 (Fisher et al., 2016) and 7 (Pye et al., 2015), consistent with a significant fraction of the monoterpene nitrates being highly functionalized (Lee et al., 2016). The studies identified particle-phase

hydrolysis as an important modulator of particulate organic-nitrate concentrations and organic nitrate lifetime. The GEOS-Chem simulation reproduced the particle-phase organic nitrate diurnal cycles (SOAS), boundary layer concentrations, and gas-particle partitioning reasonably well; however, underestimated concentrations in the free troposphere, possibly due to measurement limitations and/or the implementation of rapid uptake followed by hydrolysis of all gas-phase

organic nitrates in the model, which may not be valid for non-tertiary organic nitrates (Fisher et al., 2016).

### 3 Perspectives and outlook

Section 3 outlines perspectives on the implications of $NO_3$-BVOC atmospheric chemistry with respect to 1) aerosol optical and physical properties; 2) health effects; 3) trends in $NO_x$ emissions

and organic aerosols and their implications for control strategies related to particulate matter; 4)



critical needs for analytical methods; 5) critical needs for models; and 6,7) critical issues to address in future field and laboratory measurements in light of current understanding of this chemistry and trends in emissions.

### 3.1 Aerosol optical and physical properties

The climatic effects of atmospheric aerosols depend on their various physical and chemical properties. Hygroscopicity, cloud condensation nuclei (CCN) activity, optical properties (namely light absorption and scattering) and ability to act as CCN and ice nuclei (IN) are the key aerosol properties that would determine their ability to affect climate. Additional properties such as aerosol number size distribution, chemical composition, mixing state and morphology will

determine whether the aerosols will be optically important or whether they would affect cloud properties. These aerosol properties depend on the sources, aging processes, and removal pathways that aerosols experience in the atmosphere (Boucher, 2013).

Absorption by aerosol may affect the cloud lifetime and altitude due to heating of the atmosphere (Mishra et al., 2014). They can also change the atmospheric lapse rate, which in turn can result in

modification in aerosol microphysics in mixed-phase, ice and convective clouds (Boucher, 2013). In addition to direct emissions of known absorbing particles (black carbon, mineral dust, biomass burning aerosols), SOA may also have absorption properties. The absorbing component of organic carbon (OC), namely "brown carbon" (BrC), is associated with OC found in both primary and secondary OC and has a spectral dependent absorption that smoothly increases from short

visible to UV wavelengths (Bond and Bergstrom, 2006). It has been suggested that BrC is a component of SOA that is composed of high molecular weight and multifunctional species such as humic like substances, organonitrates and organosulfate species (Andreae and Gelencser, 2006;Bond and Bergstrom, 2006;Ramanathan et al., 2007b;Laskin et al., 2015;Moise et al., 2015). Many modeling studies often assume that BC and mineral dust are the only two significant

types of light-absorbing aerosols on the global scale. Therefore, they treat SOA as a purely scattering component that leads to climate cooling (Stier et al., 2007;Bond et al., 2011;Ma et al., 2012). However, observations suggest that BrC is widespread mostly around and downwind urban centers (Jacobson, 1999). In such places, BrC may have significant contribution, and in some cases it may dominate the total aerosol absorption at specific (short) wavelengths

(Ramanathan et al., 2007a;Bahadur et al., 2012;Chung et al., 2012;Feng et al., 2013).



Based on observations, Chung et al. (2012) recently suggested that the direct radiative forcing of carbonaceous aerosols is +0.65 (0.5 to about 0.8) $Wm^{-2}$, comparable to that of methane, the second most important greenhouse gas. This study emphasizes the important role of BrC and calls for better measurements of the absorption properties of BrC, specifically at short wavelengths where the absorption is most significant. Many previous studies have concentrated on primary particulate matter, mostly from biomass burning. However, these studies often neglected contributions to absorption due to BrC in SOA. There is ample laboratory and field evidence for the formation of such absorbing material in SOA (Chung et al., 2012;Lack et al., 2012). This absorbing component is the least characterized component of the atmospheric absorbing aerosols and constitutes a major knowledge gap, calling for an urgent need to identify the optical properties of the organic (BrC) component in SOA, and the chemical pathways leading to its formation and losses (Laskin et al., 2015;Lin et al., 2015;Moise et al., 2015).

Recently, Washenfelder et al. (2015) measured aerosol optical extinction and absorption in rural Alabama during the SOAS campaign. While they found that the majority of BrC aerosol mass was associated with biomass burning, a smaller, but not negligible contribution was attributed to biogenically-derived SOA. This fraction reached a daily maximum at night and correlated with particle-phase organic nitrates, and is associated with nighttime reactions between monoterpenes and the $NO_3$ radical (Xu et al., 2015a). Based on the above, it is concluded that SOA produced from reactions of $NO_3$ with BVOC can be a major source of SOA during the night that may affect daytime aerosol loading. This important fraction of $NO_3$-derived SOA can contribute to the direct radiative effect of SOA through scattering and absorption of incoming solar radiation.

Nitration of aromatic compounds (oxidation via $NO_2$, $NO_3$, $N_2O_5$) has a potential to form chromophores that can absorb solar radiation. Theoretical and experimental studies have shown that nitration of polycyclic aromatic hydrocarbons (PAH) leads to nitro-PAH and their derivatives such as nitrophenols (Jacobson, 1999;Harrison et al., 2005;Lu et al., 2011). The nitro substituents on the aromatic ring in compounds enhance and shift the absorption to longer wavelengths (>350 nm). Field studies report that nitrogen-containing mono- and polyaromatic SOA constituents absorb light at short (near-UV and visible) wavelengths. The reaction products between $NO_3$ and BVOC have the potential to form effective chromophores. Multifunctional organic nitrates and organosulfate compounds formed during the nighttime suggest that the SOA produced from $NO_3$ reactions leads to formation of BrC that can absorb solar radiation (Iinuma et al., 2007).



Only a few studies have investigated optical properties of SOA partially composed of organonitrates (Moise et al., 2015). Most existing literature on optical properties of organonitrates in SOA has been focused on oxidation of anthropogenic precursor compounds (Jacobson, 1999;Nakayama et al., 2010;Lu et al., 2011;Liu et al., 2012b), while a few partially contradictory studies have examined SOA formed from $NO_3$ reaction with biogenic precursors (Song et al., 2013;Varma et al., 2013). The typically high mass absorption coefficient (MAC) that was observed for anthropogenic high-$NO_x$ SOA can be partially attributed to the presence of nitroaromatic groups, for example via the nitration of PAHs (Jacobson, 1999;Lu et al., 2011). Song et al. (2013) examined optical properties of SOA formed by $NO_3 + O_3 + \alpha$-pinene. With neutral seed aerosol, organonitrates were present but observed to be non-absorbing; however, with acidic seed aerosol, SOA were strongly light absorbing, which the authors attributed to nitrooxy organosulfates formed via aldol condensation. Varma et al. (2013) measured absorption of $NO_3 + \beta$-pinene SOA and found a higher refractive index than when oxidation was via OH or $O_3$, and attributed to the difference to the low HC/$NO_x$ ratio and presence of organic nitrates in the particle phase.

Laboratory and field studies suggest that SOA formed by nighttime chemistry can have profound regional and possible global climatic effects via their absorbing properties. However, the optical properties of $NO_3$-containing SOA are not well known. Varma et al. (2013) measured a high value for the refractive index real part value of 1.61 ($\pm$ 0.03) at $\lambda = 655-687$ nm following reactions of $NO_3$ with $\beta$–pinene. This value is significantly higher than values observed following OH- and ozone-initiated terpene oxidation (Figure 5) (Moise et al., 2015). This has been attributed to the high content (up to 45%) of organic nitrates in the particle phase (Varma et al., 2013).





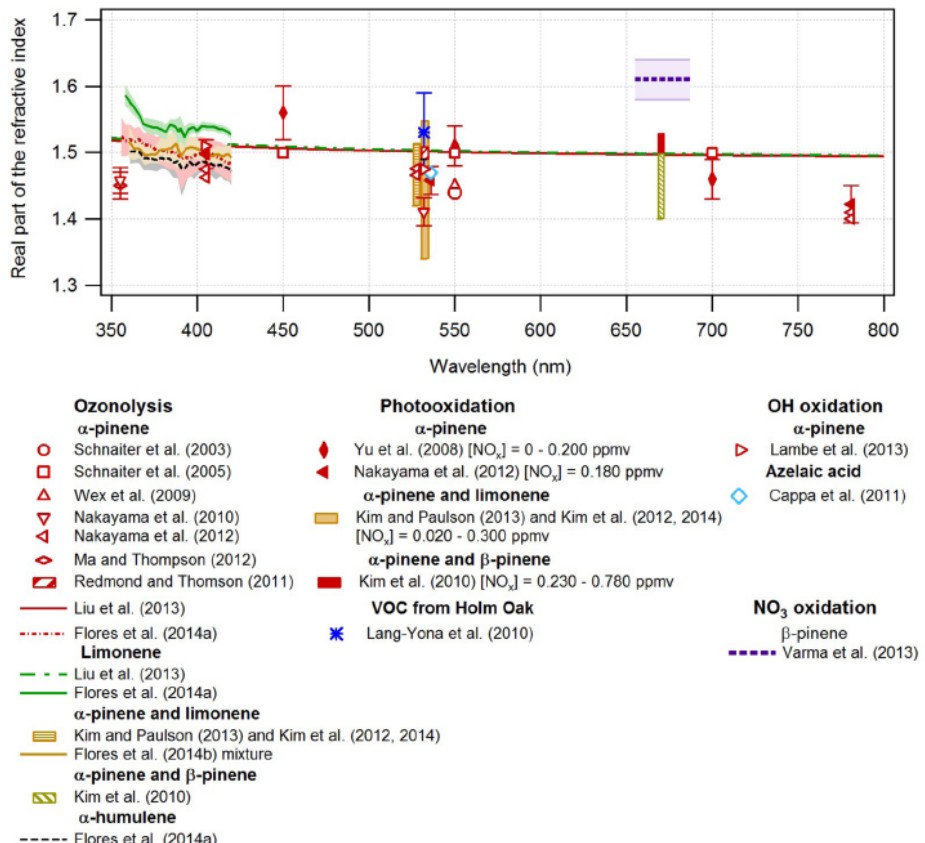

**Figure 5**. The real part of RI (mr) for biogenic SOAs compiled from several chamber studies. The legend specifies the precursor type and oxidation pathway as well as the reference. Taken from (Moise et al., 2015).

Key physical parameters of aerosols include particle size and number, volatility, viscosity, hygroscopicity and CCN activity. While it is clear that atmospheric particle size increases through condensation of BVOC + $NO_3$ oxidation products, the effect of $NO_3$ oxidation on particle number is not usually studied in laboratory experiments. Very little is known about the volatility of SOA

10    from $NO_3$ with field studies from Hyytiälä indicating that organic nitrates may have low volatility (Häkkinen et al., 2012). Viscosity is not known. Few studies report the hygroscopicity and CCN activity of SOA from $NO_3$ oxidation of BVOC. A study by Suda et al. (2014) show that organic compounds with nitrate functionality (compared to other functional groups such as hydroxyl, carbonyl, hydroperoxide) has the lowest hygroscopicity and CCN efficiency. Recently, (Cerully


et al., 2015) reported that the hygroscopicity of less-oxidized OOA (LO-OOA, mostly from BVOC+NO$_3$) is lower than other OA subtypes (MO-OOA and isoprene-OA) resolved by PMF analysis of AMS data from SOAS campaign. As monterpenes+NO$_3$ reactions can contribute ~50% of nighttime OA production (Xu et al., 2015a), results from (Cerully et al., 2015) suggested that it is possible that SOA formed from NO$_3$ oxidation of BVOC is less hygroscopic than OA formed from other oxidation pathways.

## 3.2 Health effects

Nitrated organic compounds also pose adverse health effects (Franze et al., 2003;Franze et al., 2005;Pöschl, 2005;Gruijthuijsen et al., 2006;Pöschl and Shiraiwa, 2015). In particular, several studies have reported that biological particles such as birch pollen protein can be nitrated by O$_3$ and NO$_2$ in polluted urban air (Franze et al., 2005;Reinmuth-Selzle et al., 2014). The mechanism of protein nitration involves the formation of long-lived reactive oxygen intermediates, which are most likely tyrosyl radicals (phenoxy radical derivatives of tyrosine) (Shiraiwa et al., 2011). The resulting organonitrates were found to enhance the immune response and the allergenicity of proteins and biomedical data suggest strong links between protein nitration and various diseases (Gruijthuijsen et al., 2006). Inhalation and deposition of organic nitrates into lung lining fluid in the human respiratory tract may lead to hydrolysis of organic nitrates forming HNO$_3$, which may reduce pulmonary functions (Koenig et al., 1989). Consequently, inhalation of aerosols partially composed of nitrated proteins or nitrating reagents might promote i) immune reactions, ii) the genesis of allergies, iii) the intensity of allergic diseases, and iv) airway inflammation. Toxicity of nitrated SOA compounds is still unclear. In the light of these observations and remaining uncertainties, the effect of organonitrates present in biogenic SOA on human health should be a focus of future studies.

Formaldehyde is an important source of atmospheric radicals as well as a major hazardous air pollutant (HAP). It is a degradation product of almost every VOC in the atmosphere, and BVOC are known to contribute substantially to ambient concentrations of formaldehyde (Luecken et al., 2012). The overall yield of formaldehyde from BVOC-NO$_3$ reactions is lower than from corresponding OH reactions, indicating that any changes in the relative distribution of oxidation routes will have a corresponding change in formaldehyde (and thus oxidant regeneration and HAP exposure).



### 3.3 Trends in NOₓ emissions and organic aerosols – implications for air quality control strategies

Nitrogen oxide emissions are converted to $NO_3$ and thus affect nitrate-derived SOA. In the United States, where NOₓ emissions are dominated by fuel combustion, regulatory actions have resulted
5. in decreasing NOₓ levels after increases from 1940 to 1970 (Nizich et al., 2000) and relatively stable levels between ~1970 and ~2000 (Richter et al., 2005). NOₓ emissions in the US are estimated to have decreased by roughly 30-40% in the recent past (between 2005 and 2011/12), as reflected in satellite observed $NO_2$, ground-based measurements, and the Environmental Protection Agency (EPA) National Emission Inventory (NEI) (Russell et al., 2012;Xing et al.,
10. 2013;Hidy et al., 2014;Tong et al., 2015;Xing et al., 2015). Recent decreases in NOₓ have been attributed to the mobile sector, and power plant controls including the EPA NOₓ State Implementation Plan Call implemented between 2003 and 2004 (Kim et al., 2006;Russell et al., 2012;Hidy et al., 2014;Foley et al., 2015;Lu et al., 2015). In the United States, NOₓ emissions are expected to continue to decrease and reach 72% and 61% of their 2011 levels in 2018 and 2025
15. respectively (Eyth et al., 2014). Furthermore, recent work indicates that NOₓ emissions may be overestimated in models for the United States (Travis et al., 2016) particularly for on-road gasoline vehicles (McDonald et al., 2012).

Globally, the Representative Concentration Pathway trajectories indicate that NOₓ emissions will decrease below year 2000 levels by the middle of the 21st century (Lamarque et al., 2011).
20. Europe has experienced declines in NOₓ with $NO_2$ concentrations decreasing by 20% over Western Europe between 1996 and 2002 (Richter et al., 2005) and decreasing by an additional ~20% in the more recent past (2004-2010) (Castellanos and Boersma, 2012). In contrast, NOₓ emissions in China have increased by large amounts since 1996 (Richter et al., 2005;Stavrakou et al., 2008;Verstraeten et al., 2015) with a more recent leveling out or decrease of $NO_2$
25. concentrations (Krotkov et al., 2015). $NO_2$ concentrations in India have continued to increase (Krotkov et al., 2015;Duncan et al., 2016).

These large past and expected future changes in anthropogenic NOₓ emissions indicate that analysis of historical data could reveal how NOₓ emissions affect organic aerosol formation and more specifically SOA from $NO_3$-initiated chemistry. Long-term monitoring networks often
30. measure NOₓ and OC, which could allow for correlation analysis. In addition, air quality trends in organic aerosol from traditionally less-sampled locations (e.g. (Streets et al., 2008)) and emissions for locations such as China have been characterized and could be used for analysis.





In addition to examining measurement data for relationships between $NO_3$-derived SOA and $NO_x$, chemical transport modeling with emissions sensitivity simulations can be used to provide estimates of how various SOA pathways respond to changes in $NO_x$ emissions. For example, Carlton et al. (2010b) used the CMAQ model to determine that controllable $NO_x$ emissions were

responsible for just over 20% of total SOA in the United States based on the $NO_3$-BVOC mechanism available at the time. Pye et al. (2015) predicted nitrate-derived SOA concentrations would decrease by 25% due to a 25% reduction in $NO_x$ emissions, but the overall change including all organic aerosol components would be only 9% as a result of other less sensitive (or increasing) components. Other modeling studies (Lane et al., 2008;Zheng et al., 2015;Fisher et

al., 2016) have shown that total organic aerosol or particle-phase organic nitrates may not respond strongly to decreased $NO_x$ emissions, but significant spatial and composition changes can occur.

### 3.4 Organic nitrate standards

The CIMS technique allows for highly time-resolved, chemically-speciated measurements of multifunctional organic nitrates (Beaver et al., 2012;Paulot et al., 2012;Lee et al., 2014a;Xiong et

al., 2015). Synthesis, purification and independent quantification of an individual, isomerically-specific organic nitrate is, however, required for calibration because standards are not commercially available, except for a few monofunctional alkyl nitrates.

The synthesis of monofunctional alkyl nitrates can be performed via several methods (Boschan et al., 1955), including nitration of alkyl halides with silver nitrate, direct nitration of alcohols or

alkanes with nitric acid (Luxenhofer et al., 1996;Woidich et al., 1999) or treatment of alcohols with dinitrogen pentoxide (Kames et al., 1993). Techniques for the synthesis of multifunctional nitrates, in particular hydroxynitrates, have been described in previous reports (Muthuramu et al., 1993;Kastler and Ballschmiter, 1998;Werner et al., 1999;Treves et al., 2000). Carbonyl nitrates have also been synthesized using the same protocol, i.e. nitration of hydroxyketones with

dinitrogen pentoxide (Kames et al., 1993;Suarez-Bertoa et al., 2012).

Most recently, three isomers of isoprene hydroxynitrates were synthesized (Lockwood et al., 2010;Lee et al., 2014b). As the precursor ingredient is an organic epoxide on which a hydroxy and nitrate functional groups are attached, the same protocol (Nichols et al., 1953;Cavdar and Saracoglu, 2008) can be applied to synthesize hydroxynitrates of various VOC backbones

assuming availability of precursor compounds. Oxidation of a single parent compound can yield numerous isomerically unique byproducts possessing various functional groups, including one or





more nitrates. As such, synthesis of and calibration for each rapidly become prohibitive. Given that multifunctional organic nitrates possessing more than four oxygen atoms, for which synthesis protocols currently do not exist, dominate the particulate nitrate mass of submicron particles (Lee et al., 2016), a more comprehensive calibration technique is needed. Three broad approaches are

currently utilized. One is to collect cryogenically a suite of oxidation byproducts (present in the atmosphere, formed in a simulation chamber or flow-tube, etc.) on a GC column. The desorbing eluent, separated in time by volatility/polarity as it is thermally desorbed, is measured simultaneously by CIMS and a quantitative instrument such as the TD-LIF (Day et al., 2002;Lee et al., 2014b). The corresponding eluting peaks detected by both instruments allow for calibration

of each surviving, isobarically-unique (at least for unit mass resolution spectrometers) organic nitrate (Bates et al., 2014;Schwantes et al., 2015;Teng et al., 2015). The second approach employed for the iodide-adduct ionization technique is to deduce the instrument response from a comparison of the binding energies of the numerous iodide-organic nitrate clusters to those of compounds with known sensitivities by applying variable voltages in the ion molecule reaction

region to break up charged clusters systematically. The rate at which the signal of an organic nitrate cluster decays with voltage is a function of its binding energy, which governs its transmission efficiency through the electric fields and thus its sensitivity (Lopez-Hilfiker et al., 2015). Lastly, quantum chemical calculations of specific compounds allow the determination of the sensitivity of their iodide-adduct (Iyer et al., 2016) and $CF_3O^-$ (Kwan et al., 2012;Paulot et al.,

2012) ionizations.

### 3.5 Critical needs for models

### 3.5.1 Robust and efficient representation of gas-phase chemistry

Previous sections have detailed the reactions of BVOC with $NO_3$ and the need to include this chemistry to represent more accurately processes that control $O_3$ and SOA formation. But

applying that information in a way that can be used for air quality studies presents a serious challenge. As highlighted in Section 2.6.1, the chemical mechanisms currently being used in AQMs are limited in their representation of $NO_3$-BVOC chemistry, largely lumping all monoterpenes together, and with no agreement on yields.   The lack of detail in current mechanisms is reflected in the variety of methods by which SOA formation from BVOC-$NO_3$

chemistry is estimated (Section 2.6.2).





Typically the NO₃-BVOC chemistry is implemented in AQMs into the existing system of organic and inorganic chemical reactions that occur in the atmosphere. Because there may be hundreds or thousands of different chemical reactions occurring simultaneously and the computational efforts required to solve those on a 3D grid are onerous, the chemical mechanisms used in AQMs are

typically condensed to a certain extent. The greatest challenges in modeling the reactions initiated by NO₃ and BVOC in AQMs are 1) deciding how much detail must be included to represent accurately the chemistry; 2) estimating intermediate reactions and/or products when direct experimental observations are not available; 3) integrating the new reactions into existing chemical mechanisms; and 4) validating the complete schemes against observational data.

Including all of the attack pathways and isomers that are formed in the reactions of NO₃ and BVOC and their subsequent products rapidly becomes an intractable problem, as the number of species and reactions produced from a VOC grows exponentially with the number of carbons in the compound (Aumont et al., 2005), resulting in an estimate of almost 400 million products from a single C₁₀ hydrocarbon. Even restricting the chemistry solely to the RO₂ formed from α-pinene,

β-pinene and limonene via addition of NO₃ to the double bond results in 861 unique product species and 2646 reactions as estimated from the MCM (http://mcm.leeds.ac.uk/MCM-devel/home.htt, (Saunders et al., 2003)). In comparison, the chemical mechanisms used in AQMs typically consider a total of 100-200 species and less than 400 reactions to model the entire gas-phase chemistry occurring in the troposphere. One challenge is to find a balance between

complexity and computational efficiency that involves both deriving complete mechanisms as well as condensing them to the extent possible.

The second major challenge is that many of the chemical pathways must be estimated given the limited experimental measurements of intermediate reaction rate constants and products. Structure-activity predictions have been used heavily in the past, but these have been formulated

for a limited number of compounds. Their predictions become less accurate as the complexity of the molecule increases (Calvert et al., 2015). When heterogeneous reactions play a significant role in the transport and fate of reaction products, as they do in monoterpene chemistry, the challenge becomes even greater. With recent research, new product structures that contribute to SOA have been identified (Boyd et al., 2015). However, these are not covered by existing

predictive theory, and these new pathways must be characterized including reaction rate constants, co-reactants and products. Physical parameters of all of these new species, such as solubility, radiative properties, emission rates and deposition velocities also are required, but data are often unavailable for these or even comparable species.


The last challenge is integrating the chemistry within the rest of the chemistry occurring in the atmosphere. The major chemical mechanisms used in AQMs today were developed primarily to address episodes of elevated $O_3$ under conditions of high $NO_x$ and have been evaluated for this purpose. Thus the mechanisms often do not lend themselves well to predicting the chemistry of

complex VOCs or other air quality endpoints (Kaduwela et al., 2015). Minor pathways with respect to $O_3$ formation have been removed from the mechanisms to reduce the computational burden, but these pathways may be important for formation of SOA. In addition, the detailed chemistry of multi-step alkoxy and peroxy radical chemistry is condensed into a single step in some mechanisms, but identifying whether these radicals react with $NO_x$ or $HO_x$ or isomerize is

critical for predicting the types of organic molecules that are formed. As described in the previous section 2.6, existing mechanisms include the capability for a limited number of nitrates and in many cases the links to facilitate expansion to more detailed representations are missing.

Significant work must be done to allow modelers to implement this new information in AQMs and thus use this updated knowledge to develop improved predictions of future air quality. One

approach is to focus on key chemicals of interest, derive mechanisms that are suitable for specialized applications and append these on to existing frameworks (for example, (Xie et al., 2013)). The longer term view requires a more comprehensive approach that draws on the development of community archives that can better accommodate rapidly changing information and better represent the interactions of biogenic with anthropogenic chemistry. Here we put

forward our recommendations for future work in the following areas:

1. Development of tools for the semi-automated production of the reaction pathways and products of later-generation products resulting from alternate pathways of radical reactions with BVOC. These tools should be able to incorporate experimental data when available. In conjunction with the automated development, we require advanced methods for condensing these large

mechanisms into computationally feasible reaction schemes.

2. Improvements in estimation techniques for uncertain pathways, including reaction rate constants for multifunctional stable compounds and radicals for which measurements are not available and the quantification of the errors associated with these estimation methods.

3. Development of theory and techniques for integrating gas-phase products with SOA

production, in this case, describing the transformation of gas-phase organic nitrates to their SOA products.





4. Development of more versatile base mechanisms that have the flexibility to accept increased detail in VOC description and the continuing validation of the complete tropospheric chemical mechanisms against observational data.

### 3.5.2 Improved techniques and protocols for evaluation of complex and reduced gas-phase mechanisms

Generally speaking, once detailed mechanisms are developed they are evaluated through some form of benchmarking. Systematic strategies for mechanism evaluation include validation of highly detailed mechanisms unable to be run in 3D models against benchmark data from well-characterized simulation chamber experiments (Jenkin et al., 1997;Aumont et al., 2005) and the incorporation of these mechanisms into box or 1D models to validate radical and short lived species against field campaign observations. Less detailed AQ mechanisms can then be compared to these reference mechanisms by way of sensitivity experiments in idealized modelling studies – often aimed at assessing the sensitivity in $O_3$ to changing $NO_x$ and VOC emissions (Archibald et al., 2010;Squire et al., 2015). AQ mechanisms are often also then re-evaluated against chamber and or field experiment data before they are implemented into 3D models and then undergo evaluation against extensive measurements in the residual layer.

One of the greatest challenges in the BVOC-$NO_3$ system is that current night-time measurements are mainly collected from surface sites, which are confined to a shallow surface layer at night and not representative of the whole night-time boundary layer. The impact of night-time chemistry on daytime ozone and nitrogen/aerosol budget would require careful investigation of night-time chemistry in the residual layer, which contains >80% of air masses at night.

Moreover, the benchmarking activities mentioned above and the development process discussed in section 3.5.1 are not well aligned. A more unified approach that identifies some key mechanistic problems and identifies strategies to evaluate them is required in order to make improved progress on simulating the changing composition of the atmosphere.

### 3.5.3 Reduce uncertainties in sub-grid scale processes

Uncertainties in AQM predictions also arise from the representation of physical sub-grid scale processes. The ones particularly relevant for the $NO_3$-BVOC chemistry include, but are not limited to the following.



***Nighttime boundary layer (NBL) mixing***

The spatial distribution of BVOC and $NO_x$ precursors is highly variable, but the current AQMs neglect these heterogeneities and assume perfect mixing within grid cells of typically 3-10 kilometers in the horizontal. At those resolutions, models are unable to resolve the localized

surface emission sources and the microscale structure of boundary layer turbulence, and therefore cannot resolve spatial heterogeneities in chemistry, partitioning and mixing of chemicals, which are essential for predicting the concentrations of secondary pollutants.

Typically, the freshly emitted monoterpenes species have a tendency to accumulate in the shallow nighttime boundary layer (typically < 200 m), and can react with $NO_3$ if available. However,

often $NO_3$ is located in the residual layer that is decoupled from the NBL, and the BVOC+$NO_3$ reactions would depend on the model's ability to mix the two layers. Thus, mixing within and out of the boundary layer provides a key challenge for modeling the impacts of BVOC-$NO_3$ chemistry, as the measured gradients of $NO_3$ and BVOC are very strong in the vertical (e.g. (Brown et al., 2007b;Fuentes et al., 2007)).

A large focus on model evaluation has been on the impacts of higher horizontal resolution (Jang et al., 1995). It has been shown in several cases that owing to the complex interplay of chemical families, the sensitivity of the chemical system isn't captured at lower resolution (e.g. (Cohan et al., 2006)). However, very little work has focused on the role of improvements in vertical resolution, despite the fact that intra model differences in properties like the height of the

boundary layer vary by over a factor of 2 in cases (e.g. (Hu et al., 2010)) Moreover, the NBL is not well mixed, so evaluation of nocturnal physics requires more than just evaluating the NBL height.

***Plume parameterizations***

Typically, parameterizations have been applied to anthropogenic emission sources (e.g. aircraft

plumes, urban plumes) and not to biogenic sources. Partly this is a result of the differences in the source terms, anthropogenic emissions often being well represented as point sources in space, whereas biogenic emissions are often large area sources. However, as the emissions of BVOC are often very species specific, and observations highlight large spatial variability over small areas (e.g. (Niinemets et al., 2010)), the adoption of the anthropogenic plume parameterizations to

BVOC emissions could lead to improvements in model performance.

One approach is the plume in grid (PiG) parameterization (Karamchandani et al., 2002). This aims to solve the problem of sub-grid scale chemical processes by implementing ensembles of Gaussian puffs within the AQM (e.g. (Vijayaraghavan et al., 2006)). Other approaches include



hybrid Eulerian-Lagrangian models (Alessandrini and Ferrero, 2009). These differ from the PiG models by simulating large numbers of stochastic trajectories that can make use of variable reactive volumes to simulate their diffusion into background air masses simulated on Eulerian grids.

Global models have generally used a different approach to the problem of plumes. Broadly, following one of the two paradigms (Paoli et al., 2011): (i) modify the emissions of the reaction mix (using so-called effective emissions or applying emission conversion factors) (ii) modify the rates of reaction (effective reaction rates).

### 3.6 Field studies in the developing world and under-studied areas

In light of the questions raised earlier in this review, assessing the role of $NO_3$-BVOC chemistry will require field experiments over a wide range of ratios of isoprene to monoterpene emissions and of $NO_3$ to BVOC. Future studies of $NO_3$-BVOC chemistry are in planning stages for North America. These studies will provide access to environments with different $NO_x$ levels and over a modest range of isoprene and monoterpene emission rates. A wider range of these parameters can

be accessed in countries where $NO_x$ emissions controls are not as completely implemented and where BVOC emissions are abundant. Bringing the state-of-the-art capabilities developed for study of $NO_3$-BVOC chemistry in the last few years to locations in China and India would allow insight not only into the role of that chemistry in those countries now but also into the role this chemistry played in Europe and the US prior to implementation of current emission standards.

Experiments in the tropics potentially would allow observations of the confluence of BVOC and very low $NO_x$ to be explored, thus providing insight into BVOC-$NO_3$ as a sink of $NO_x$.

### 3.7 Future needs for chamber studies

Field studies by definition include the entire complexity of the real atmosphere, so that the identification of single processes and quantification of their impact is challenging. Specific

experiments in chambers allow investigating processes without effects from meteorology, which largely impacts observations in the real atmosphere specifically during nighttime, when the lower troposphere is not as well mixed as it is during daytime. In chamber experiments, specific compounds of interest can be isolated and studied under well-controlled oxidation environments, allowing a more detailed and direct characterization of the composition, chemical, and physical

properties of aerosols.    Because such laboratory and chamber data provide the basic





understanding for predicting SOA formation, it is important that the design of such experiments mimic the oxidation environments in the atmosphere to the greatest extent possible. Several important needs for understanding $NO_3$-BVOC chemistry in chambers include: 1) elucidation of kinetic and mechanistic information for $NO_3$-BVOC reactions; 2) characterization of wall losses

for low-volatility products in the $NO_3$-BVOC system; 3) understanding the fate of peroxy radicals in the nighttime atmosphere and its influence on this chemistry; 4) hydrolysis and photooxidation of BVOC-derived organic nitrates from specific BVOC plus specific oxidant pairs over a range of appropriate conditions; 5) optical properties of aerosol organic nitrate; and 6) intercomparison of instrumental methods for key species in the $NO_3$-BVOC system.

***Kinetic and mechanistic elucidation***

The number of chamber studies investigating $NO_3$ chemistry is small compared to the number of studies for photochemical oxidation and ozonolysis. In most of the studies, gas-phase oxidation products and SOA yields from the oxidation of BVOC have been measured. Studies include the investigation of SOA from monoterpenes (Wangberg et al., 1997;Griffin et al., 1999;Hallquist et

al., 1999;Spittler et al., 2006;Fry et al., 2009;Fry et al., 2011;Boyd et al., 2015;Nah et al., 2016b), methyl butenol (Fantechi et al., 1998b) and isoprene (Rollins et al., 2009;Ng et al., 2010;Schwantes et al., 2015). A few more studies investigated gas-phase reaction kinetics, including the reactions of $NO_3$ with aldehydes (Clifford et al., 2005;Bossmeyer et al., 2006), amines (Zhou and Wenger, 2013), or cresol (Olariu et al., 2013). As a consequence of the small

number of studies, the oxidation mechanisms of organic compounds by $NO_3$ and the yields of oxidation products in the gas phase and particle phase have larger uncertainties. The well-controlled oxidation environments in chamber experiments, coupled with complimentary gas-phase and particle-phase measurements (online and offline), allow for elucidating detailed oxidation mechanisms under varying reaction conditions (Ng et al., 2008;Boyd et al.,

2015;Schwantes et al., 2015). Identification of gas- and particle-phase reaction products from $NO_3$-BVOC chemistry within controlled chamber environments can also greatly aid in the interpretation of field data in which multiple oxidants and BVOC are present. Future chamber experiments will naturally take advantage of new advanced gas/aerosol instrumentation and aim to constrain the formation yields of gas-phase oxidation products and establish a fundamental

understanding of aerosol formation mechanisms from $NO_3$-BVOC under a wide range of oxidation conditions.

***Wall losses***





Although chamber studies allow separating processes driven by chemistry and physics from transport processes that occur in the real atmosphere, careful characterization of the behavior of $NO_3$ in chambers as well as the organic products of the $NO_3$ oxidation remains a research priority. Yields of gas-phase oxidation products can be influenced by chamber specific loss processes

(surface loss on the chamber wall) and SOA yields can be impacted by both direct loss of particles and loss of species that can condense on particle or chamber wall surfaces (McMurry and Grosjean, 1985;Loza et al., 2010;Matsunaga and Ziemann, 2010;Yeh and Ziemann, 2014;Zhang et al., 2014a;Zhang et al., 2015;Krechmer et al., 2016;La et al., 2016;Nah et al., 2016a;Ye et al., 2016). The extent to which vapor wall loss affects SOA yields appears to be

dependent on the VOC system, from small effects to as high as a factor of four (Zhang et al., 2014a;Nah et al., 2016a). Studies on the effects of vapor loss on SOA formation from BVOC+$NO_3$ are limited. With minimal or no competing gas-particle partitioning processes, substantial vapor wall loss of organonitrates have been observed in experiments not specific to $NO_3$ oxidation (Yeh and Ziemann, 2014;Krechmer et al., 2016). However, the use of excess

oxidant concentrations and rapid SOA formation in BVOC+$NO_3$ experiments (hence shorter experiments) could potentially mitigate the effects of vapor wall loss on SOA yields in chamber studies (Boyd et al., 2015;Nah et al., 2016a). In light of the developing understanding of this issue, an important consideration for the design of any future systematic chamber studies is the influence of vapor wall loss on SOA formation from nitrate radical oxidation under different

reaction conditions, such as peroxy radical fates, relatively humidity, seeds, oxidant level, chamber volume, etc.

***Peroxy radical fate***

As discussed above, the fate of peroxy radicals directly governs the product distribution in the $NO_3$-BVOC system, including SOA yields and composition. Dark reactions of peroxy radicals

differ significantly from their photochemical analogs, and are directly related to the development of mechanistic understanding in the $NO_3$-BVOC system. There is a need to systematically investigate reaction products and SOA formation from $NO_3$-BVOC reactions under different peroxy radical reaction regimes, but this aspect of this system has only recently become of focus of chamber studies (Ng et al., 2008;Boyd et al., 2015;Schwantes et al., 2015). Rapid formation of

highly oxygenated organic nitrates has been observed in laboratory studies of β-pinene+$NO_3$ and α-pinene+$NO_3$; these products could be formed by unimolecular isomerization of peroxy radicals or autoxidation (Nah et al., 2016b). The importance of this peroxy radical reaction channel in $NO_3$-BVOC chemistry warrants further studies. Future chamber studies will need to be explicit in





their specification of the peroxy radical chemistry regime that is investigated in a particular experiment, and will need to relate that regime to the conditions of the ambient nighttime atmosphere.

*Organic nitrate hydrolysis and photooxidation*

Recent field studies have shown that organic nitrates formed from $NO_3$-BVOC are important components of ambient OA. However, the reactivity in both gaseous and condensed phases of these biogenic nitrates, in particular of polyfunctional nitrates, has been subject to few studies and requires better characterization to evaluate the role of these compounds as reservoirs/sinks of $NO_x$. Field results suggest that the fate of organic nitrates in both the gas and aerosol phase have

variable lifetimes with respect to hydrolysis. The difference in the relative amount of primary/secondary/tertiary organic nitrates (which hydrolyze with different rates) from nitrate radical oxidation vs. photochemical oxidation needs to be constrained. Most of the hydrolysis studies thus far are conducted in bulk, except for a few recent studies on monoterpene organic nitrates (e.g., (Boyd et al., 2015; Rindelaub et al., 2015)). The solubility of multifunctional organic

nitrates in water and the extent to which hydrolysis occur in aerosol water warrant future studies. The effect of particle acidity on hydrolysis might also be important for organic nitrates formed in different BVOC systems.

While there are extensive studies on photochemical aging of ozonolysis SOA, studies on photochemical aging of $NO_3$-initated SOA and organic nitrates are extremely limited. A recent

study shows that the particle-phase organic nitrates from $NO_3$+β-pinene and $NO_3$+α-pinene reactions exhibit completely different behavior upon photochemical aging during the night-to-day transition, and act as permanent and temporary $NO_x$ sinks, respectively (Nah et al., 2016b). With the ~1 week lifetime of aerosols in the atmosphere and that majority of $NO_3$-BVOC organic nitrates are formed at night, the photochemical fates of these organic nitrates could impact next-

day $NO_x$ cycling and ozone formation. Therefore, there is a critical need to understand the mutigenerational chemistry and characterize the evolution of organic nitrates over its diurnal life cycle, including aging $NO_3$-initiated SOA and organic nitrates by photolysis and/or OH radicals.

*Aerosol optical properties*

The optical properties, especially in the short wavelength region, of $NO_3$-derived SOA may be

most conveniently measured during coordinated chamber studies that also include detailed measurements of gas phase oxidation chemistry and aerosol composition. Such studies could also serve to isolate the specific optical properties of $NO_3$-BVOC derived aerosol to obtain better optical closure in the interpretation of field data. Field studies that include aerosol optical





properties measurements in conjunction with other instrumentation can help quantify the bulk organic nitrate abundance and identify organic nitrate molecular composition in the SOA.

### Instrument intercomparisons

The discussion above shows that recent advances in analytical instrumentation are key to the

developing science of $NO_3$-BVOC chemistry. Chamber studies provide an excellent opportunity for the comparison and validation of such instrumentation. State-of-the-art and developing instruments for measurement of $NO_3$ and $N_2O_5$ were compared approximately a decade ago (Fuchs et al., 2012;Dorn et al., 2013). These instruments have improved and proliferated since that time, and further validation studies are needed. Measurements of total and speciated gas and

aerosol-phase organic nitrates, as well as other oxygenated compounds that result from $NO_3$-BVOC reactions, have not been the subject of a specific intercomparison study. Their comparison and validation will be a priority in future coordinated chamber studies.

### Utility of coordinated chamber studies

Because of the need for a better understanding of $NO_3$ oxidation and because of the challenges of

chamber studies investigating $NO_3$ chemistry, coordination between studies carried out in different chambers, and between chamber and field studies, can augment efforts of single or stand-alone chamber studies. Coordinated studies that would include several chambers could increase the accuracy and reliability of results and quantify realistic errors associated with product yields estimates. This can be achieved by determining the same quantities in similar

experiments in different chambers. Studies could benefit from complementary capabilities and properties of chambers. Chambers that typically operate at higher concentration ranges and therefore increased oxidation rates are suitable to perform a larger number of experiments and are therefore useful for screening experiments and a series of experiments with systematic variations of chemical conditions. Other chambers are suited to perform experiments at atmospheric reactant

concentrations. Experiments in these chambers may take place on a longer time scale, for example a scale characteristic of the duration of at least one night. Analytical instrumentation and capability also differs considerably among chambers, so that coordinated chamber studies can make use of the determination of complementary quantities such as product yields of different organic compounds and characterization of various properties of particles for the same chemical

system. For instance, it would be invaluable to conduct coordinated studies where a variety of instrument techniques are used to measure total and speciated gas- and particle-phase organic nitrates, as well as aerosol physical and chemical properties in the same chamber.





Substantial insights into aerosol sources, formation, and processing can be gained from coordinated laboratory chamber and field studies. Laboratory chamber experiments provide the fundamental data to interpret field measurements. The analysis of field data in turn can provide important insights for constraining chamber experiment parameters so that the oxidation conditions in chambers can be as representative as possible of those in the atmosphere. Two recent sets of experiments serve as examples of this approach. Fundamental chamber studies on β-pinene+$NO_3$ in the Georgia Tech Environmental Chamber (GTEC) facility under conditions relevant to the southeast US provided constraints on the contribution of monoterpenes+$NO_3$ to ambient OA during the 2013 SOAS campaign (Boyd et al., 2015;Xu et al., 2015b). The Focused Isoprene eXperiment at California Institute of Technology (FIXCIT) chamber study following SOAS advanced the understanding of isoprene oxidation chemistry relevant to the SE US (Nguyen et al., 2014). In all essence, it is important not to consider fundamental laboratory studies as isolated efforts, but they should be an integrated part of field studies. Similarly, having the modeling community involved in early planning stages of laboratory and field studies will greatly aid in the identification of critically-needed measurement data.

## 4 Impacts of $NO_3$-BVOC chemistry on air quality

The previous sections have demonstrated that understanding how $NO_3$ reacts with BVOC, including the ultimate fate of products, encompasses all aspects of atmospheric physics, chemistry, and transport. These sections have raised numerous complex and fascinating science questions and highlighted the critical need for much more basic science to fill in unknown aspects of this system. However "getting this system right" is not just an interesting science problem because it has direct implications for policy decisions that governments across the world are taking to protect citizens and ecosystems from harmful effects of air pollutants. Addressing the uncertainties raised in the previous chapters is critical for developing efficient, accurate, and cost-effective strategies to reduce the harmful effects of air pollution.

BVOC have long been predicted to be significant contributors to regional and global $O_3$ (e.g. (Pierce et al., 1998;Curci et al., 2009)) and $PM_{2.5}$ (Pandis et al., 1991), with $NO_3$ reactions providing a major pathway for loss of ambient BVOC (Winer et al., 1984;Pye et al., 2010;Xie et al., 2013). If BVOC react with $NO_3$ instead of OH, the $O_3$ production of the BVOC can be reduced relative to reactions through OH, although in some instances they may slightly increase $O_3$ by reducing next-day $NO_x$. For example, measurements in St. Louis (Millet et al., 2016),



demonstrate that nights with lower levels of NO$_3$ resulted in higher isoprene concentrations the following morning, producing higher and earlier O$_3$ peaks. Recent insights into the role of biogenic nitrates, which are produced in large quantities through the reactions of NO$_3$ with primary emitted BVOC and subsequent reactions of their stable products, demonstrate that these

compounds can substantially alter the availability of NO$_x$ (Perring et al., 2013). This highlights the importance of accurate treatment of fates of organic nitrates form from nighttime chemistry in models, which will impact the next day NO$_x$ and ozone levels. Organic nitrates from BVOC+NO$_3$ also can contribute to nitrogen deposition (Nguyen et al., 2015), which adversely impacts ecosystems. The ways in which the patterns of deposition for biogenic nitrates affect inorganic

nitrate deposition remain poorly understood.

*Implications for spatial distribution of ozone and PM2.5*

While it is clear that NO$_3$-BVOC reactions affect oxidant availability and SOA, it remains unclear how large that role is in the ambient atmosphere relative to other VOCs and other oxidants and where it occurs. The extent of O$_3$ formation downwind of sources is influenced by the transport of

NO$_y$ species, including organic nitrates, which can release NO$_x$ downwind, where O$_3$ may be formed more efficiently. Biogenically-derived nitrates are the dominant organic nitrates in many places (Pratt et al., 2012). A variety of different organic nitrates are formed from different BVOC, with some being short lived (releasing NO$_2$ locally) and others being long lived (releasing NO$_2$ downwind unless they are removed in the meantime). Errors in our attribution of the lifetime of

individual biogenic nitrate compounds can cause errors in predicted NO$_x$ redistributions regionally and globally and modify the spatial distributions of O$_3$ (Perring et al., 2013). Updates to the chemistry of BVOC-NO$_3$ also could alter calculations of the relative role of biogenic species versus anthropogenic pollutants to O$_3$ and PM$_{2.5}$ formation.

*Implications for control strategy development*

Air quality models are used not only to understand the production of air pollutants in the current atmosphere, but also to guide the development of strategies to reduce the future pollution burden. Uncertainties in the chemistry and removal of BVOC can contribute to uncertainties in the sensitivity of O$_3$ and PM to emission reduction strategies. This increases the risk of implementing expensive control strategies that are found later to be inefficient (more control specified than

needed) or ineffective (do not meet the air quality goals for which they were developed). As noted by Millet et. al (Millet et al., 2016), in urban areas downwind of high isoprene emissions, the loss of isoprene by NO$_3$ at night can produce the opposite O$_3$-NO$_x$ behavior that would normally be expected in urban areas, potentially causing a reassessment of optimum control





strategies. In addition, the early $O_3$ peaks noted on low $NO_3$ nights expands the high ozone time window, resulting in higher 8-hour $O_3$ averages, on which regulatory compliance in the US is based.

The uncertainties in our understanding of $NO_3$-BVOC chemistry propagate into chemical mechanisms, as described in section 3. Past work has shown that vastly different chemical mechanisms may predict similar $O_3$ in current atmospheres but show huge differences for intermediate species (e.g., (Luecken et al., 2008)) and different potential responses to precursor reductions, including different indicators of $O_3$ sensitivity to VOC versus $NO_x$ reductions (Knote et al., 2015). The presence of large weekend effects in $NO_x$ makes identifying such errors more likely in current analyses.

Incorporating new information on biogenic chemistry within a chemical mechanism will impact the availability of $NO_x$, (e.g. (Archibald et al., 2010;Xie et al., 2013)) and modify the predicted effectiveness of anthropogenic $NO_x$ controls. Incorporating new chemical information into models can also impact $PM_{2.5}$ sensitivities to $NO_x$ reductions. In one example, organic $PM_{2.5}$ was almost twice as responsive to a $NO_x$ reduction than in older mechanisms (Pye et al., 2015). Because much of the $NO_x$ dependence of $O_3$ and aerosols from $NO_3$-BVOC reactions is inadequately accounted for in models, the few examples we have hint that current $NO_x$ control strategies might result in more significant improvements to air quality than currently assumed. Retrospective analyses should focus on elucidating the elements of this hypothesis that are represented in the historical record.

The role of climate change in modifying air quality is also a highly uncertain issue and may be particularly sensitive to the characterization of BVOC. Biogenic emissions may increase or decrease in the future, depending on many factors including increased temperatures, changes in water availability, occurrence of biotic and abiotic stress (e.g. (Kleist et al., 2012;Wu et al., 2015), $CO_2$ fertilization, $CO_2$ inhibition, and land use changes (Chen et al., 2009;Squire et al., 2014). Uncertainties in biogenic reactions may be amplified as they become a larger share of the VOC burden in some places. The predicted response of $O_3$ to future climate has been found to be especially sensitive to assumptions about the chemical pathways of BVOC reactions, in particular the treatment of nitrates. Mao et al. (2013) and several earlier researchers found that predictions of the $O_3$ response to $NO_x$ reductions change from negative to positive depending solely on how the isoprene chemistry was represented. Similarly, a comparison of several widely-used chemical mechanisms with varied descriptions of BVOC-derived nitrates (Squire et al., 2015) found that description of BVOC chemistry significantly alters not only the amount of oxidant change


predicted under future scenarios but also the direction of the change. Direct measurements of the key steps in isoprene oxidation should eliminate the ambiguity in such model calculations. Nonetheless, the exquisite sensitivity of model predictions of ozone trends to the representation of isoprene and $NO_x$ indicates that ambient observations of those trends are an excellent strategy for

evaluating the accuracy of mechanisms.

The relative distribution of emissions among different types of BVOC may also shift as climate and land use changes, emphasizing the need to understand differences among terpenes in their chemistry, transport, and fate (Pratt et al., 2012). While most of the research to date has been done on isoprene, with some on α-pinene and β-pinene, little has been done on products or

reaction parameters of other terpenes. The previous sections have demonstrated that different terpenoid structures can have vastly different atmospheric chemistry and physical properties, so it is unclear whether assuming one "representative" species or distribution, as is done in most chemical mechanisms, will adequately account for future impacts of BVOC on $O_3$ and PM.

***Summary of impacts***

This review has illustrated that accurate characterization of $NO_3$-BVOC chemistry is critical to our understanding of both the air quality and climate impacts of $NO_x$ emissions. Our knowledge of the complexity of $NO_3$-BVOC reaction pathways and multi-generational products has advanced rapidly, especially in the last decade. Despite the fact that much of that information is not yet in a form that can be included in current air quality models, we anticipate improved predictive

capabilities in models in the coming years through sustained laboratory and field studies coupled to model development. While the current levels of uncertainty make it difficult to accurately quantify the impact of $NO_3$-BVOC chemistry on air pollutant concentrations, we expect that developments in this field will improve the effectiveness of air pollution control strategies going forward. The limited studies available demonstrate that even small changes to BVOC chemistry

modify the production of oxidants ($NO_3$, OH and $O_3$) and change the transport of $NO_y$. Therefore, $NO_3$-BVOC oxidation modifies the chemical regime in which additional BVOC oxidation occurs. Of most importance will be the studies that indicate changes in the direction of predicted future pollutant concentrations as chemical mechanisms of BVOC are updated. Emissions control strategies and attainment of air quality goals rely on the best possible chemical models. Current

and future laboratory and field research is critical to the improvement of chemical mechanisms that account for biogenic chemical processes and products which will augment efforts to reduce harmful air pollutants.





**Acknowledgements**

The authors acknowledge support from the International Global Atmospheric Chemistry project (IGAC), the U.S. National Science Foundation (NSF grant 1541331), and Georgia Tech College of Engineering and College of Sciences for support of the workshop on Nitrate Radicals and

5  Biogenic Hydrocarbons that led to this review article. SSB acknowledges support from the NOAA Atmospheric Chemistry, Carbon Cycle and Climate program.

The U.S. Environmental Protection Agency (EPA), through its Office of Research and Development (ORD), collaborated in the research described herein. It has not been subject to EPA review and therefore does not necessarily reflect the views of EPA. No official endorsement

10  should be inferred.





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
