# Peer review of "Nitrate radicals and biogenic volatile organic compounds: oxidation, mechanisms and organic aerosol"

_Atmospheric Chemistry and Physics, 2016_

## Referee Comment (RC1) · Anonymous Referee #1 · 14 Sep 2016

1) General Comments The manuscript acp-2016-734 submitted by Ng et al. presents a critical and extensive review on nighttime chemistry derived from the workshop held in Georgia Institute of Technology past June. The paper includes the state-of-the-art on the reactivity of NO3 radical with BVOC both in gas and condense phases, the mechanisms of gaseous products and particulate matter formation in the reactions reviewed, and secondary organic aerosol yields from them. It also makes a revision of the instrumental techniques employed in the detection of NO3, organic nitrates and particulate matter. A compilation of the recent field measurements of NO3 and BVOC is presented as well, together with a synthesized description of the air quality models that try to interpret the observations in the atmosphere. Finally, the last section about

future perspectives is a good indication that there are still many uncertainties in the NO3 chemistry of biogenic VOCs. In my opinion, the paper under discussion is very well structured, showing the data in a concise manner. The great complexity of the NO3-BVOC chemistry is highlighted through the paper and I agree with the authors that coordinated research projects on this subject is strongly recommended to provide a more complete view of the nighttime chemistry of areas with high levels of biogenic emissions. After addressing the comments/suggestions given below, this review is publishable in the Atmospheric Chemistry and Physics journal.

2) Specific Comments To be consistent through the paper the term "rate constant" or "rate coefficient" should be unified. Comments on Table 1 As this table compiles all data reported for the gas-phase rate constants of NO3+BVOC reactions, alignment of columns is needed to correlate the value of k with the reference. Is it possible to include the temperature range in the T-dependence expressions? Is the second value for k in isoprene 1.21e-13 cm3 molecule-1 s-1? In alpha-pinene, what is the uncertainty in 5.82e-12 value? What about the stated uncertainties? I guess they are those reported by the authors, in some cases one standard deviation and in others, twice the standard deviation. Add a footnote to clarify this aspect. In my opinion, in a review the presentation of data should be done in order of publication (or inverse order, if you wish), but not mixed. Comment on Table 2 In my opinion, there is a lot of information of the last column. Can it be split in two columns: OA loading and relevant information? Comment on Table 6 As in the heading of this table it is stated that the data presented are relative to SOA formation, delete "SOA" from the third column. What was exactly intended to highlight in the last column named "References"? What do the author want to state by SOA/monoterpenes/chemistry/etc included in the last column? Is it possible to include a column with references for isoprene separately from monoterpenes? Comment on Figure 5. Include the permission of the journal in the caption. 3) Suggestions/ Typographical errors Suggestions of text addition/change are written in capital characters. Abstract, Line 17: "The first section. . .." could be replaced by "The first PART OF THIS REVIEW summarizes" or "The first SECTIONS. . .". In fact, the first section is solely

the Introduction co Page 5, line 8: "…..BVOC, such as…..monoterpenes, are…" Page 5, line23: "BVOC-NO3-derived organic nitrates" could be replaced by "organic nitrates derived from BVOC-NO3 reaction" Page 7, line 10-11: "field observations relevant to the understanding of NO3 and BVOC". This sentence is weird or seems to be incomplete. The understanding of? Page 14, line 26: "…hydroxyl nitrates FORMED from…"; "hydroperoxides FORMED from…" Page 16, line 23: (RH) should be placed after "Relative humidity" in line 21. Page 16, line 22: Add "…heterogeneous uptake of N2O5, PRECURSOR OF NO3." Page 24, line 6: Replace "peroxy-radical" by "peroxy radical". Page 24, line 17: Replace "H-atom" by "H atom" Page 25, line 16-17: "…predicts [NO3] between…" Use the multiplication symbol in the concentration values. Remove the semi-colon after the last concentration. The sentence "The higher values are associated with urban clouds, with rural and marine clouds an order of magnitude lower" can be rephrases as "High NO3 concentration levels are associated with urban clouds, while in rural and marine clouds these levels are an order of magnitude lower". Is this a general trend? Page 26, line 15. "Eq. 5" should be "Eq. 2" Page 27, line 14: hydroxyl and nitrate radicals have been already defined previously in the manuscript. Page 28, line 10: Delete the hyphen after 10-2. Page 39, line 32: Replace "+/-" by "$\pm$" Page 40, line 13: "…from 2-900 ppt" is better to be written as "…from 2 to 900 ppt" Page 47, line 10: In my opinion, the heading is not necessary. Page 47, line 15: Delete an extra period. Page 48, lines 33-34: The rate constant units in cm3 molecule-1 s-1 for consistency with the rest of the manuscript. Page 49, line 1: The rate constant units in cm3 molecule-1 s-1 for consistency with the rest of the manuscript. Page 51, line 6: Replace "and Odum" by "AN (or THE) Odum"

---

## Referee Comment (RC2) · Anonymous Referee #2 · 16 Sep 2016

Nitrate radicals and biogenic volatile organic compounds: oxidation, mechanisms and organic aerosol

N.L. Ng et al.

The manuscript by Ng et al. provides an overview of NO3-BVOC chemistry in the atmosphere, and how this impacts atmospheric aerosols. The review has stemmed from a workshop on NO3-BVOC chemistry, and provides a review of recent laboratory studies of kinetics and reaction mechanisms, field measurements/techniques, leading to a series of recommendations for future work. However, the manuscript provides little by

way of critical review, and, perhaps owing to the comprehensive nature of the review, in several places lacks the detail required for a tutorial-style review. For example, the rate coefficients for NO3+BVOC reactions determined in previous experimental work are listed in Table 1, and in many cases, the rate coefficients have already been reviewed by IUPAC, with no new recommendations/review of the data. It is not entirely clear what is gained by the inclusion of these data in this review. Similarly, there is little information regarding the mechanisms of NO3-BVOC chemistry – some figures displaying mechanisms for some reactions/models at least may be helpful to aid comparison. In addition, the overview of experimental techniques used to measure NO3 (and N2O5) in the atmosphere is covered in greater depth in the 2012 review by Brown & Stutz, with little/no new insights presented here.

While the review does cover areas of interest to many atmospheric scientists, it does read somewhat as a summary of presentations at a meeting/workshop covering a broad range of topics, with the detail regarding each of the topics to be found elsewhere.

In general, the tables are poorly formatted and difficult to read. It may be helpful to remove the references and list them numerically with a list of numerical references in supplementary information (similar to the style adopted for Figure 2 perhaps). Numerous acronyms are used without definition. This review may be particularly useful to graduate students and researchers new to the area, and definitions should be provided at the point of first use to aid those researchers. A glossary would also be helpful.

Minor comments:

Page 3, line 24: Terms such as 'enormous' should be avoided, quantify the statement or at least provide a relative comparison to anthropogenic emissions.

Page 4, line 6: Should this be 'alkenes'? The reactions of NO3 with alkanes are very slow.

[Figure]

Page 7, lines 17-19: Are the two statements regarding isoprene rate coefficients contradictory? If the results from different studies for isoprene differ by over a factor of two, how is the IUPAC recommendation precise?

Page 16, line 4: '. . . was investigated in the context of. . .', perhaps a brief discussion of the results of this investigation?

Page 16, ;one 8: 'at conditions' to 'under conditions'.

Page 16, line 11: '. . . fairly constant. . .', please quantify.

Page 16, line 27: 'reaction' to 'reactions'.

Page 17, line 18: '. . . observed a tight correlation. . .', provide the correlation coefficient.

Page 17, line 21: Remove 'the' in '. . . the nucleation and . . .'.

Page 20, line 18-19: 'Our present understanding. . .', please provide a reference.

Page 21, line 20: Please clarify the term 'oxidized once', what does this mean? A single oxidation step?

Page 21, line 26: '. . . effectively limit rate. . .' to '. . . effectively limit the rate. . .'.

Page 22, line 1: 'a phase transition' or 'phase transitions'.

Page 23, line 23: lower case 'u' in undec-10-ene-1-thiol.

Page 25, line 11: Provide a definition for 'SOx-'.

Page 25, line 14: Perhaps 'in situ' in place of 'instantaneous'?

Page 25, line 30: Can the rate coefficients used in CAPRAM be linked to those recommended by IUPAC?

Page 27, line 10: 'similar rate constants'.

Page 27, line 11: 'well correlated', provide the correlation coefficients.

Page 29, line 13: How long is 'long path', the 5 km referred to above?

Page 30, line 3: Perhaps worth noting that a single wavelength measurement is more difficult to identify potential interferences?

Page 31, line 2: Please provide some information regarding the 'Langley-Plot method'.

Page 32, line 11: Note that DOAS is also an absolute technique.

Page 33, line 17: Quantify the statement 'less good'.

Page 40, line 6: What was the integration time associated with the 6 ppt detection limit?

Page 44, line 1: Superscript in R^2, note that 'R' has been quoted elsewhere, it would be good to maintain consistency throughout.

Page 46, line 6: 'K-puszta' to 'Puszta'?

Page 46, line 29: Why not maintain consistency and use either LO-OOA or SV-OOA throughout?

Page 48, line 15: Please briefly outline the potential fates of RO2 radicals and how these each impact nitrate formation and the nitrate yield.

Page 48, line 31: 'k' in italics.

Page 48, line 33: Multiplication symbol in equation rather than letter 'x' (and in following equations). Units for these equations should be presented in a consistent way with those elsewhere in the manuscript (e.g. in Table 1).

Page 60, line 3: 'the SOAS campaign'.

Page 60, line 4: Remove the brackets around the reference.

Page 66, line 12: Inconsistency between 'modeling' and 'modelling' throughout the manuscript.

Page 76, line 22: Space in 'BVOCchemistry'.

References: Chemical symbols – some use subscripts, others don't, while some display markup language formatting instructions (e.g. page 92, line 25 onwards).
* * *

---

## Author Comment (AC1) · 29 Dec 2016

We thank the reviewers for their comments. Our specific response can be found below. The reviewers' comments are in italics and changes made to the manuscript are in quotation marks. All changes made are minor and do not affect the conclusions in the manuscript.

**Response to Reviewer 1**

*1. In my opinion, the paper under discussion is very well structured, showing the data in a concise manner. The great complexity of the NO3-BVOC chemistry is highlighted through the paper and I agree with the authors that coordinated research projects on this subject is strongly recommended to provide a more complete view of the nighttime chemistry of areas with high levels of biogenic emissions. After addressing the comments/suggestions given below, this review is publishable in the Atmospheric Chemistry and Physics journal.*

Response: We thank the reviewer for the positive comments.

*2. To be consistent through the paper the term "rate constant" or "rate coefficient" should be unified.*

Response: In the revised manuscript, we will use term "rate constant".

*3. Comments on Table 1: As this table compiles all data reported for the gas-phase rate constants of NO3+BVOC reactions, alignment of columns is needed to correlate the value of k with the reference. Is it possible to include the temperature range in the T-dependence expressions? Is the second value for k in isoprene 1.21e-13 cm3 molecule-1 s-1? In alpha-pinene, what is the uncertainty in 5.82e-12 value? What about the stated uncertainties? I guess they are those reported by the authors, in some cases one standard deviation and in others, twice the standard deviation. Add a footnote to clarify this aspect. In my opinion, in a review the presentation of data should be done in order of publication (or inverse order, if you wish), but not mixed.*

Response: Rows in the table have now been aligned. Ranges of temperature have been specified for all rate constants, including single temperature measurements. The second value for k in isoprene has been corrected to $1.3\times10^{-12}$. Uncertainty in the $\alpha$-pinene value is now specified. The stated uncertainties are as given in the references in the table, and the reader is referred to the individual studies. Rate constants are now also given in order of publication.

*4. Comment on Table 2: In my opinion, there is a lot of information of the last column. Can it be split in two columns: OA loading and relevant information?*

Response: We agree that this column is messy, but unfortunately this is unavoidable because it arises from the information available in the papers cited. The "relevant information" included in this column often substitutes for direct knowledge of the OA (e.g., where reacted $N_2O_5$ was reported but not total OA), so we don't think that splitting the column will necessarily help – where OA information is

available, that is listed; where not, other relevant information substitutes. Nevertheless, prompted by this reviewer comment and in an effort to make the table more readable, we have moved some information that could better be characterized as "additional notes" on the analysis methods to footnotes. We have also added another study for the $NO_3 + \alpha$-pinene yields for completeness, and have added some additional notes to that footnote.

*5. Comment on Table 6: As in the heading of this table it is stated that the data presented are relative to SOA formation, delete "SOA" from the third column. What was exactly intended to highlight in the last column named "References"? What do the author want to state by SOA/monoterpenes/chemistry/etc included in the last column? Is it possible to include a column with references for isoprene separately from monoterpenes?*

Response: "SOA" deleted from third column as suggested by the reviewer. For clarity, last column had been eliminated in favor of footnotes at the bottom of the table to clarify which aspect of the mechanism each references is for.

*6. Comment on Figure 5. Include the permission of the journal in the caption.*

Response: Permission has been updated.

*7. Abstract, Line 17: "The first section. . .." could be replaced by "The first PART OF THIS REVIEW summarizes" or "The first SECTIONS. . .". In fact, the first section is sole the Introduction.*

Response: The sentence "The first section summarizes the current literature on $NO_3$-BVOC chemistry, with a particular focus on recent advances in instrumentation and models, and in organic nitrate and secondary organic aerosol (SOA) formation chemistry" is changed to

"The first half of the review summarizes the current literature on $NO_3$-BVOC chemistry, with a particular focus on recent advances in instrumentation and models, and in organic nitrate and secondary organic aerosol (SOA) formation chemistry"

*8. Page 5, line 8: ". . ..BVOC, such as. . ..monoterpenes, are. . ."*

Response: Commas are added as suggested.

*9. Page 5, line23: "BVOC-NO3-derived organic nitrates" could be replaced by "organic nitrates derived from BVOC-NO3 reaction"*

Response: Modified.

*10. Page 7, line 10-11: "field observations relevant to the understanding of NO3 and BVOC". This sentence is weird or seems to be incomplete. The understanding of?*

Response: The sentence is changed to "field observations relevant to the understanding of $NO_3$ and BVOC chemistry".

*11. Page 14, line 26: "…hydroxyl nitrates FORMED from…"; "hydroperoxides FORMED from…"*

Response: Corrected.

*12. Page 16, line 23: (RH) should be placed after "Relative humidity" in line 21.*

Response: Modified.

*13. Page 16, line 22: Add "…heterogeneous uptake of N2O5, PRECURSOR OF NO3."*

Response: The sentence refers to the competition between $NO_3$ and $N_2O_5$. Since $N_2O_5$ is not a precursor of $NO_3$ (rather it is in equilibrium with $NO_3$), we feel that the suggested change would not be accurate.

*14. Page 24, line 6: Replace "peroxy-radical" by "peroxy radical".*

Response: Modified.

*15. Page 24, line 17: Replace "H-atom" by "H atom"*

Response: Modified.

*16. Page 25, line 16-17: ". . .predicts [NO3] between. . ." Use the multiplication symbol in the concentration values. Remove the semi-colon after the last concentration. The sentence "The higher values are associated with urban clouds, with rural and marine clouds an order of magnitude lower" can be rephrases as "High NO3 concentration levels are associated with urban clouds, while in rural and marine clouds these levels are an order of magnitude lower". Is this a general trend?*

Response: The sentence is modified to read "Model studies with the CAPRAM mechanism (Chemical Aqueous Phase RAdical Mechanism (CAPRAM; (Herrmann et al., 2005;Tilgner et al., 2013)) predict [$NO_3$] between $1.6 \times 10^{-16}$ mol $L^{-1}$ to $2 \times 7 \cdot 10^{-13}$ mol $L^{-1}$. High $NO_3$ concentration levels are associated with urban clouds, while in rural and marine clouds these levels are an order of magnitude lower. Since the $NO_3$ concentrations are related to the $NO_x$ budget, typically higher $NO_3$ concentrations are present under urban cloud conditions compared to rural and marine cloud regimes."

*17. Page 26, line 15. "Eq. 5" should be "Eq. 2"*

Response: The reviewer is right. This is corrected in the revised manuscript.

*18. Page 27, line 14: hydroxyl and nitrate radicals have been already defined previously in the manuscript.*

Response: The sentence "Figure 4 shows a comparison of the modeled chemical turnovers of reactions of organic compounds with hydroxyl (OH) versus nitrate ($NO_3$) radicals distinguished for different compound classes" is changed to

 "Figure 4 shows a comparison of the modeled chemical turnovers of reactions of organic compounds with OH versus $NO_3$ radicals distinguished for different compound classes"

*19. Page 28, line 10: Delete the hyphen after 10-2.*

Response: There is no hyphen after $10^{-2}$ in line 10. The review could be referring to the hyphen in line 12. It is now formatted to be the same as in the previous two lines to show the range of the $NO_3$/OH ratios.

*20. Page 39, line 32: Replace "+/-" by "±"*

Response: Modified.

*21. Page 40, line 13: "…from 2-900 ppt" is better to be written as "…from 2 to 900 ppt"*

Response: Modified.

*22. Page 47, line 10: In my opinion, the heading is not necessary.*

Response: We respectfully disagree, and have retained the heading in this instance, since it separates the summary of the text from the specific descriptions of different regions.

*23. Page 47, line 15: Delete an extra period.*

Response: Corrected.

*24. Page 48, lines 33-34: The rate constant units in cm3 molecule-1 s-1 for consistency with the rest of the manuscript.*

Response: The rate constant units has been changed from "molec$^{-1}$ cm$^3$ s$^{-1}$" to "cm$^3$ molec$^{-1}$ s$^{-1}$".

*25. Page 49, line 1: The rate constant units in cm3 molecule-1 s-1 for consistency with the rest of the manuscript.*

Response: The rate constant units has been changed from "molec$^{-1}$ cm$^3$ s$^{-1}$" to "cm$^3$ molec$^{-1}$ s$^{-1}$".

*26. Page 51, line 6: Replace "and Odum" by "AN (or THE) Odum"*

Response: Corrected. The sentence now reads as "…most models parameterize SOA formation separately from gas-phase chemistry using either the Odum 2-product…".

---

## Author Comment (AC2) · 29 Dec 2016

We thank the reviewers for their comments. Our specific response can be found below. The reviewers' comments are in italics and changes made to the manuscript are in quotation marks. All changes made are minor and do not affect the conclusions in the manuscript.

**Response to Reviewer 2**

*1.  The manuscript by Ng et al. provides an overview of NO3-BVOC chemistry in the atmosphere, and how and how this impacts atmospheric aerosols. The review has stemmed from  a workshop on NO3-BVOC chemistry, and provides a review of recent laboratory studies of kinetics and reaction mechanisms, field measurements/techniques, leading to a  series of recommendations for future work.*

Response: We thank the reviewer for these comments.

*2.  However, the manuscript provides little by way of critical review, and, perhaps owing to the comprehensive nature of the review, in  several places lacks the detail required for a tutorial-style review.*

Response: Please see responses below for addition of details that aid in the tutorial-style review.  In addition to answering several of the more detailed comments, we have added a new figure to the introduction to illustrate features of nighttime chemistry and anthropogenic-biogenic interactions that are central themes of the review.

[Figure]

**Figure 1.** Schematic of nighttime NO$_3$-BVOC chemistry

We have also added two new figures in the section on organic nitrates to summarize observations of these species in aerosols in the U.S. and Europe.

[Figure]

**Figure 2.5.2a** Average mass concentrations (in mg m-3, ambient temperature and pressure) of submicrometer particulate organic nitrates (NO3, org) and particulate inorganic nitrates (NO3, inorg) in different months at multiple sites. The concentrations correspond to mass concentrations of −ONO2 functionality. Note that the y-axis is different for site with total nitrate greater than 1 mg m-3 (shaded). Detailed information and measurements for each site are provided in Table SI-5 in the Supplement.

[Figure]

**Figure 2.5.2b** Percentage (by mass, cyan) of submicrometer particulate organic nitrate aerosols in ambient organic aerosols in different months at multiple sites. Detailed information and measurements for each site are provided in Table SI-5 in the Supplement.

*3.  For example, the rate  coefficients for NO3+BVOC reactions determined in previous experimental work  are  listed  in  Table 1, and in many cases, the rate coefficients have already been reviewed  by IUPAC, with  no  new  recommendations/review of the data.  It is not entirely clear  what is gained by the inclusion of these data in this review.*

Response:  The reviewer is correct that some rate constant data are available through IUPAC, and where such data have been reviewed by IUPAC they are specified in table 1.  However, the current compilation is more comprehensive and includes a larger array than is available through the IUPAC database.  For example, IUPAC reviews rate constant data for 15 NO$_3$-BVOC reaction, whereas the current table 1 reviews 44 reactions, almost 3 times as many.   Furthermore, this review collects all relevant rate constants in one place so that the interested investigator has a single reference to understand what is, and what is not, available from previous investigations of NO$_3$-BVOC rate constants.

*4.     Similarly, there is little information regarding the mechanisms of NO3-BVOC chemistry – some figures displaying  mechanisms for some reactions/models at least may be helpful to aid comparison.*

Response:   We agree.   Section 2.1.2 on mechanisms has now been re-written and expanded to include a discussion of the mechanistic features of $NO_3$ + isoprene and $NO_3$ + monoterpenes.  A new figure 2.1 now replaces the older figure 1.  The new text and figures are as follows:

[revised manuscript text omitted]

*5. In addition, the overview of experimental techniques used to measure NO3 (and N2O5) in the atmosphere is covered in greater depth in the 2012 review by Brown & Stutz, with little/no new insights presented here.*

Response:  We agree that other reviews, including Brown & Stutz 2012, discuss experimental techniques for $NO_3$ and $N_2O_5$.  However, this review also includes a discussion of experimental

methods for other compounds relevant to $NO_3$-BVOC chemistry, such as gas and aerosol phase organic nitrates.  It would be remiss of the authors within that context *not* to include a discussion of $NO_3$ and $N_2O_5$, even if such a discussion exists in a separate context elsewhere in the literature.

*6.  While the review does cover areas of interest to many atmospheric scientists, it does read somewhat as a summary of presentations at a meeting/workshop covering a broad range of topics, with the detail regarding each of the topics to be found elsewhere.*

Response: We respectfully disagree.  The topics and outline for this review were agreed upon through a series of breakout meetings and group discussions rather than through a compliation of individual presentations.  The outline and flow of the manuscript is logical, beginning with the current state of the science and proceeding to discuss research needs within this topic.

*7.  In general, the tables are poorly formatted and difficult to read.  It may be helpful to remove the references and list them numerically with a list of numerical references in supplementary information (similar to the style adopted for Figure 2 perhaps).*

Response: We agree, the table formatting, which was done to match ACP standards, has made some of the presented information difficult to read.  We have made an effort to improve the formatting, borders and spacing of all tables to make them more readable.

*8.  Numerous acronyms are used without definition. This review may be particularly useful to graduate students and researchers new to the area, and definitions should be provided  at the point of first use to aid those researchers. A glossary would also be helpful.*

Response:  We thank the reviewer for this suggestion.  A glossary of acronyms and common chemical formulas has been included at the end.

*9.  Page 3, line 24: Terms such as 'enormous' should be avoided, quantify the statement  or at least provide a relative comparison to anthropogenic emissions.*

Response: Agreed.  "enormous" has been replaced by "large".

*10. Page 4, line 6: Should this be 'alkenes'? The reactions of NO3 with alkanes are very slow.*

Response: The review is right. This has been corrected in the revised manuscript.

*11. Page 7, lines 17-19: Are the two statements regarding isoprene rate coefficients contradictory? If*

*the results from different studies for isoprene differ by over a factor of two, how is the IUPAC recommendation precise?*

Response:  The reviewer is correct that the statements are contradictory.  The first sentence has been modified to state that BVOC with multiple $NO_3$ rate constant studies have been evaluated by IUPAC to produce recommended values.  The sentence now reads as "The most emitted/important BVOC have been subjected to several kinetic studies, using both absolute and relative methods, which are evaluated to determine recommended rate constants by IUPAC".

*12. Page 16, line 4: '. . . was investigated in the context of. . .', perhaps a brief discussion of the results of this investigation?*

Response:  We have added a second clause to this sentence to briefly describe the results of Draper et al: "This difference among monoterpenes was investigated in the context of the competition between $O_3$ and $NO_3$ oxidation (Draper et al., 2015), in which shifting from $O_3$-dominated to $NO_3$-dominated oxidation was observed to suppress SOA formation from $\alpha$-pinene, but not from $\beta$-pinene, $\Delta$-carene, or limonene."

*13. Page 16, line 8: 'at conditions' to 'under conditions'.*

Response: Modified.

*14. Page 16, line 11: '. . . fairly constant. . .', please quantify.*

Response: The sentence is modified to "Recent experiments showed that the particulate organic nitrates formed from $\beta$-pinene-$NO_3$ are resilient to photochemical aging, while those formed from $\alpha$-pinene-$NO_3$ evaporate readily".

*15. Page 16, line 27: 'reaction' to 'reactions'.*

Response: Corrected.

*16. Page 17, line 18: '. . . observed a tight correlation. . .', provide the correlation coefficient.*

Response:  The correlation coefficient was 0.99 ($r^2$ = 0.98) and has been added to the text.

*17. Page 17, line 21: Remove 'the' in '. . . the nucleation and . . .'.*

Response: Corrected.

18. *Page 20, line 18-19: 'Our present understanding...', please provide a reference.*

Response: We have added Hallquist et al (ACP, 2009) and Tsigaridis et al. (ACP, 2014) as references.

19. *Page 21, line 20: Please clarify the term 'oxidized once', what does this mean? A single oxidation step?*

Response: The phrase has been clarified to read "… undergo a single reaction with $NO_3$."

20. *Page 21, line 26: '… effectively limit rate…' to '… effectively limit the rate…'.*

Response: Corrected.

21. *Page 22, line 1: 'a phase transition' or 'phase transitions'.*

Response: The sentence is changed to "…..owing to a phrase transition…".

22. *Page 23, line 23: lower case 'u' in undec-10-ene-1-thiol.*

Response: Corrected.

23. *Page 25, line 11: Provide a definition for 'SOx-'.*

Response: The sentence is changed to "….. $SO_x^-$ (sulfur-containing radical anions)…".

24. *Page 25, line 14: Perhaps 'in situ' in place of 'instantaneous'?*

*Response: Phrase has been changed to "measured in-situ"*

25. *Page 25, line 30: Can the rate coefficients used in CAPRAM be linked to those recommended by IUPAC?*

Response: Unfortunately, there are no IUPAC recommendations for the aqueous phase available right now. However, this task is currently ongoing. In the future, all measured CAPRAM rate coefficients will be linked to IUPAC data. At current, CAPRAM has to be considered as a database with carefully evaluated

kinetic data for aqueous phase reactions.

*26. Page 27, line 10: 'similar rate constants'.*

Response: Corrected.

*27. Page 27, line 11: 'well correlated', provide the correlation coefficients.*

Response: The correlation of OH and $NO_3$ rate expressions and statistical analysis data calculated from kinetic data of hydroxyl radical and nitrate radical of the reactions with organic compounds for the various compound classes are given in Table SI-3 of the Supplement. We have made this clear in the revised manuscript.

*28. Page 29, line 13: How long is 'long path', the 5 km referred to above?*

Response: Quotations around long path have been removed, and the typical path length on the order of several km for the active techniques has now been specified.

*29. Page 30, line 3: Perhaps worth noting that a single wavelength measurement is more difficult to identify potential interferences?*

Response: This depends on how the measurement is done. For $NO_3$, with chemical titration by NO combined with 662 nm detection, the measurement is quite specific. The phrase has been modified to include "specificity can be achieved through chemical titration with NO (Brown *et al*. 2001)."

*30. Page 31, line 2: Please provide some information regarding the 'Langley-Plot method'.*

*Response:* The Langley-Plot method is now defined in the text. "… by the Langley-Plot method (Noxon et al., 1980), which takes advantage of the different dependence of tropospheric and stratospheric $NO_3$ slant column density on the Lunar Zenith Angle."

*31. Page 32, line 11: Note that DOAS is also an absolute technique.*

Response: We agree that DOAS is absolute, but the paragraph in question is not about DOAS, but rather about potential wall losses in CEAS or CRDS instruments with closed sample cells. Since DOAS does not fall in this category of instruments, it is best left out of this list.

*32. Page 33, line 17: Quantify the statement 'less good'.*

Response: We add a sentence at Page 33, Line 19: "Whereas differences between $N_2O_5$ measurements were less than 20% in the absence of aerosol, measurements differed up to factor of 2.5 for highest aerosol surface concentrations of $5x10^8$ $nm^2cm^{-3}$. Also differences between $NO_3$ measurements showed an increasing trend (up to 50%) with increasing aerosol surface concentration for some instruments."

*33. Page 40, line 6: What was the integration time associated with the 6 ppt detection limit?*

Response: Unfortunately, the integration time is not given in the Golz et al. (2001) reference.

*34. Page 44, line 1: Superscript in Rˆ2, note that 'R' has been quoted elsewhere, it would be good to maintain consistency throughout.*

Response: Superscript has been corrected, and the manuscript has been checked to ensure that $R^2$ is used for correlations throughout.

*35. Page 46, line 6: 'K-puszta' to 'Puszta'?*

Response: K-puszta is the name of the field station.

*36. Page 46, line 29: Why not maintain consistency and use either LO-OOA or SV-OOA throughout?*

Response: We note that LO-OOA has been used consistently throughout the manuscript. LO-OOA is used instead of SV-OOA as multiple studies have found that the degree of oxidation of OA does not necessarily correlate with its volatility.

*37. Page 48, line 15: Please briefly outline the potential fates of RO2 radicals and how these each impact nitrate formation and the nitrate yield.*

Response: The potential fates of the $RO_2$ radical are listed on page 47, line 28 (they are reaction with $NO_3$, $RO_2$, $HO_2$, or NO). Table 5 summarizes the nitrate yield in current models by $RO_2$ fate which is sometimes identical for all $RO_2$ fates. The yield spans from 0 to 100% (page 48, line 6-7). Observed organonitrate yields are covered earlier in the manuscript, and Table 2 summarizes observed organonitrate molar yields.
*38. Page 48, line 31: 'k' in italics.*

Response: Corrected.

*39. Page 48, line 33: Multiplication symbol in equation rather than letter 'x' (and in following*

*equations). Units for these equations should be presented in a consistent way with those elsewhere in the manuscript (e.g. in Table 1).*

Response:  Multiplication sign has been corrected, and units have been standardized to $cm^3$ $molecule^{-1}$ $s^{-1}$ throughout

*40. Page 60, line 3: 'the SOAS campaign'.*

Response: Corrected.

*41. Page 60, line 4: Remove the brackets around the reference.*

Response: Corrected.

*42. Page 66, line 12: Inconsistency between 'modeling' and 'modelling' throughout the manuscript.*

Response: This is the only place where "modelling" is used. We have changed it to "modeling" to be consistent with the rest of the manuscript.

*43. Page 76, line 22: Space in 'BVOCchemistry'.*

Response: Corrected.

*44. References: Chemical symbols – some use subscripts, others don't, while some display markup language formatting instructions (e.g. page 92, line 25 onwards).*

Response: The subscripts in the references are now properly formatted.

---

## Author Response (AR1)

We thank the reviewers for their comments. Our specific response can be found below. The reviewers' comments are in italics and changes made to the manuscript are in quotation marks. All changes made are minor and do not affect the conclusions in the manuscript.

**Response to Reviewer 1**

*1. In my opinion, the paper under discussion is very well structured, showing the data in a concise manner. The great complexity of the NO3-BVOC chemistry is highlighted through the paper and I agree with the authors that coordinated research projects on this subject is strongly recommended to provide a more complete view of the nighttime chemistry of areas with high levels of biogenic emissions. After addressing the comments/suggestions given below, this review is publishable in the Atmospheric Chemistry and Physics journal.*

Response: We thank the reviewer for the positive comments.

*2. To be consistent through the paper the term "rate constant" or "rate coefficient" should be unified.*

Response: In the revised manuscript, we will use term "rate constant".

*3. Comments on Table 1: As this table compiles all data reported for the gas-phase rate constants of NO3+BVOC reactions, alignment of columns is needed to correlate the value of k with the reference. Is it possible to include the temperature range in the T-dependence expressions? Is the second value for k in isoprene 1.21e-13 cm3 molecule-1 s-1? In alpha-pinene, what is the uncertainty in 5.82e-12 value? What about the stated uncertainties? I guess they are those reported by the authors, in some cases one standard deviation and in others, twice the standard deviation. Add a footnote to clarify this aspect. In my opinion, in a review the presentation of data should be done in order of publication (or inverse order, if you wish), but not mixed.*

Response: Rows in the table have now been aligned. Ranges of temperature have been specified for all rate constants, including single temperature measurements. The second value for k in isoprene has been corrected to $1.3 \times 10^{-12}$. Uncertainty in the $\alpha$-pinene value is now specified. The stated uncertainties are as given in the references in the table, and the reader is referred to the individual studies. Rate constants are now also given in order of publication.

*4. Comment on Table 2: In my opinion, there is a lot of information of the last column. Can it be split in two columns: OA loading and relevant information?*

Response: We agree that this column is messy, but unfortunately this is unavoidable because it arises from the information available in the papers cited. The "relevant information" included in this column often substitutes for direct knowledge of the OA (e.g., where reacted $N_2O_5$ was reported but not total OA), so we don't think that splitting the column will necessarily help – where OA information is

available, that is listed; where not, other relevant information substitutes. Nevertheless, prompted by this reviewer comment and in an effort to make the table more readable, we have moved some information that could better be characterized as "additional notes" on the analysis methods to footnotes. We have also added another study for the $NO_3 + \alpha$-pinene yields for completeness, and have added some additional notes to that footnote.

*5. Comment on Table 6: As in the heading of this table it is stated that the data presented are relative to SOA formation, delete "SOA" from the third column. What was exactly intended to highlight in the last column named "References"? What do the author want to state by SOA/monoterpenes/chemistry/etc included in the last column? Is it possible to include a column with references for isoprene separately from monoterpenes?*

Response: "SOA" deleted from third column as suggested by the reviewer.  For clarity, last column had been eliminated in favor of footnotes at the bottom of the table to clarify which aspect of the mechanism each references is for.

*6. Comment on Figure 5. Include the permission of the journal in the caption.*

Response:  Permission has been updated.

*7. Abstract, Line 17: "The first section. . .." could be replaced by "The first PART OF THIS REVIEW summarizes" or "The first SECTIONS…". In fact, the first section is sole the Introduction.*

Response: The sentence "The first section summarizes the current literature on $NO_3$-BVOC chemistry, with a particular focus on recent advances in instrumentation and models, and in organic nitrate and secondary organic aerosol (SOA) formation chemistry" is changed to

"The first half of the review summarizes the current literature on $NO_3$-BVOC chemistry, with a particular focus on recent advances in instrumentation and models, and in organic nitrate and secondary organic aerosol (SOA) formation chemistry"

*8. Page 5, line 8: "….BVOC, such as….monoterpenes, are…"*

Response: Commas are added as suggested.

*9. Page 5, line23: "BVOC-NO3-derived organic nitrates" could be replaced by "organic nitrates derived from BVOC-NO3 reaction"*

Response: Modified.

*10. Page 7, line 10-11: "field observations relevant to the understanding of NO3 and BVOC". This sentence is weird or seems to be incomplete. The understanding of?*

Response: The sentence is changed to "field observations relevant to the understanding of $NO_3$ and BVOC chemistry".

*11. Page 14, line 26: "…hydroxyl nitrates FORMED from…"; "hydroperoxides FORMED from…"*

Response: Corrected.

*12. Page 16, line 23: (RH) should be placed after "Relative humidity" in line 21.*

Response: Modified.

*13. Page 16, line 22: Add "…heterogeneous uptake of N2O5, PRECURSOR OF NO3."*

Response: The sentence refers to the competition between $NO_3$ and $N_2O_5$. Since $N_2O_5$ is not a precursor of $NO_3$ (rather it is in equilibrium with $NO_3$), we feel that the suggested change would not be accurate.

*14. Page 24, line 6: Replace "peroxy-radical" by "peroxy radical".*

Response: Modified.

*15. Page 24, line 17: Replace "H-atom" by "H atom"*

Response: Modified.

*16. Page 25, line 16-17: ". . .predicts [NO3] between. . ." Use the multiplication symbol in the concentration values. Remove the semi-colon after the last concentration. The sentence "The higher values are associated with urban clouds, with rural and marine clouds an order of magnitude lower" can be rephrases as "High NO3 concentration levels are associated with urban clouds, while in rural and marine clouds these levels are an order of magnitude lower". Is this a general trend?*

Response: The sentence is modified to read "Model studies with the CAPRAM mechanism (Chemical Aqueous Phase RAdical Mechanism (CAPRAM; (Herrmann et al., 2005;Tilgner et al., 2013)) predict [$NO_3$] between $1.6 \times 10^{-16}$ mol $L^{-1}$ to $2 \times 7 \cdot 10^{-13}$ mol $L^{-1}$. High $NO_3$ concentration levels are associated with urban clouds, while in rural and marine clouds these levels are an order of magnitude lower. Since the $NO_3$ concentrations are related to the $NO_x$ budget, typically higher $NO_3$ concentrations are present under urban cloud conditions compared to rural and marine cloud regimes."

*17. Page 26, line 15. "Eq. 5" should be "Eq. 2"*

Response: The reviewer is right. This is corrected in the revised manuscript.

*18. Page 27, line 14: hydroxyl and nitrate radicals have been already defined previously in the manuscript.*

Response: The sentence "Figure 4 shows a comparison of the modeled chemical turnovers of reactions of organic compounds with hydroxyl (OH) versus nitrate ($NO_3$) radicals distinguished for different compound classes" is changed to

"Figure 4 shows a comparison of the modeled chemical turnovers of reactions of organic compounds with OH versus $NO_3$ radicals distinguished for different compound classes"

*19. Page 28, line 10: Delete the hyphen after 10-2.*

Response: There is no hyphen after $10^{-2}$ in line 10. The review could be referring to the hyphen in line 12. It is now formatted to be the same as in the previous two lines to show the range of the $NO_3$/OH ratios.

*20. Page 39, line 32: Replace "+/-" by "±"*

Response: Modified.

*21. Page 40, line 13: "…from 2-900 ppt" is better to be written as "…from 2 to 900 ppt"*

Response: Modified.

*22. Page 47, line 10: In my opinion, the heading is not necessary.*

Response: We respectfully disagree, and have retained the heading in this instance, since it separates the summary of the text from the specific descriptions of different regions.

*23. Page 47, line 15: Delete an extra period.*

Response: Corrected.

*24. Page 48, lines 33-34: The rate constant units in cm3 molecule-1 s-1 for consistency with the rest of the manuscript.*

Response: The rate constant units has been changed from "molec$^{-1}$ cm$^3$ s$^{-1}$" to "cm$^3$ molec$^{-1}$ s$^{-1}$".

*25. Page 49, line 1: The rate constant units in cm3 molecule-1 s-1 for consistency with the rest of the manuscript.*

Response: The rate constant units has been changed from "molec$^{-1}$ cm$^3$ s$^{-1}$" to "cm$^3$ molec$^{-1}$ s$^{-1}$".

*26. Page 51, line 6: Replace "and Odum" by "AN (or THE) Odum"*

Response: Corrected. The sentence now reads as "…most models parameterize SOA formation separately from gas-phase chemistry using either the Odum 2-product…".

**Response to Reviewer 2**

*1.   The manuscript by Ng et al. provides an overview of NO3-BVOC chemistry in the atmosphere, and how and how this impacts atmospheric aerosols. The review has stemmed from  a workshop on NO3-BVOC chemistry, and provides a review of recent laboratory studies of kinetics and reaction mechanisms, field measurements/techniques, leading to a  series of recommendations for future work.*

Response: We thank the reviewer for these comments.

*2.   However, the manuscript provides little by way of critical review, and, perhaps owing to the comprehensive nature of the review, in  several places lacks the detail required for a tutorial-style review.*

Response: Please see responses below for addition of details that aid in the tutorial-style review.  In addition to answering several of the more detailed comments, we have added a new figure to the introduction to illustrate features of nighttime chemistry and anthropogenic-biogenic interactions that are central themes of the review.

[Figure]

**Figure 1.** Schematic of nighttime NO$_3$-BVOC chemistry

We have also added two new figures in the section on organic nitrates to summarize observations of these species in aerosols in the U.S. and Europe.

[Figure]

**Figure 2.5.2a** Average mass concentrations (in mg m-3, ambient temperature and pressure) of submicrometer particulate organic nitrates (NO3, org) and particulate inorganic nitrates (NO3, inorg) in different months at multiple sites. The concentrations correspond to mass concentrations of −ONO2 functionality. Note that the y-axis is different for site with total nitrate greater than 1 mg m-3 (shaded). Detailed information and measurements for each site are provided in Table SI-5 in the Supplement.

[Figure]

**Figure 2.5.2b** Percentage (by mass, cyan) of submicrometer particulate organic nitrate aerosols in ambient organic aerosols in different months at multiple sites. Detailed information and measurements for each site are provided in Table SI-5 in the Supplement.

*3.    For example, the rate  coefficients for NO3+BVOC reactions determined in previous experimental work  are  listed in Table 1, and in many cases, the rate coefficients have already been reviewed  by IUPAC, with no new recommendations/review of the data.  It is not entirely clear  what is gained by the inclusion of these data in this review.*

Response:  The reviewer is correct that some rate constant data are available through IUPAC, and where such data have been reviewed by IUPAC they are specified in table 1.  However, the current compilation is more comprehensive and includes a larger array than is available through the IUPAC database.  For example, IUPAC reviews rate constant data for 15 $NO_3$-BVOC reaction, whereas the current table 1 reviews 44 reactions, almost 3 times as many.   Furthermore, this review collects all relevant rate constants in one place so that the interested investigator has a single reference to understand what is, and what is not, available from previous investigations of $NO_3$-BVOC rate constants.

*4.    Similarly, there is little information regarding the mechanisms of NO3-BVOC chemistry – some figures displaying  mechanisms for some reactions/models at least may be helpful to aid comparison.*

Response: We agree. Section 2.1.2 on mechanisms has now been re-written and expanded to include a discussion of the mechanistic features of NO$_3$ + isoprene and NO$_3$ + monoterpenes. A new figure 2.1 now replaces the older figure 1. The new text and figures are as follows:

[revised manuscript text omitted]

*5.    In addition, the overview of experimental techniques used to measure NO3 (and N2O5) in the atmosphere is covered in greater depth in the 2012 review by Brown & Stutz, with little/no new insights presented here.*

Response:  We agree that other reviews, including Brown & Stutz 2012, discuss experimental techniques for $NO_3$ and $N_2O_5$.  However, this review also includes a discussion of experimental

methods for other compounds relevant to $NO_3$-BVOC chemistry, such as gas and aerosol phase organic nitrates. It would be remiss of the authors within that context *not* to include a discussion of $NO_3$ and $N_2O_5$, even if such a discussion exists in a separate context elsewhere in the literature.

*6. While the review does cover areas of interest to many atmospheric scientists, it does read somewhat as a summary of presentations at a meeting/workshop covering a broad range of topics, with the detail regarding each of the topics to be found elsewhere.*

Response: We respectfully disagree. The topics and outline for this review were agreed upon through a series of breakout meetings and group discussions rather than through a compliation of individual presentations. The outline and flow of the manuscript is logical, beginning with the current state of the science and proceeding to discuss research needs within this topic.

*7. In general, the tables are poorly formatted and difficult to read. It may be helpful to remove the references and list them numerically with a list of numerical references in supplementary information (similar to the style adopted for Figure 2 perhaps).*

Response: We agree, the table formatting, which was done to match ACP standards, has made some of the presented information difficult to read. We have made an effort to improve the formatting, borders and spacing of all tables to make them more readable.

*8. Numerous acronyms are used without definition. This review may be particularly useful to graduate students and researchers new to the area, and definitions should be provided at the point of first use to aid those researchers. A glossary would also be helpful.*

Response: We thank the reviewer for this suggestion. A glossary of acronyms and common chemical formulas has been included at the end.

*9. Page 3, line 24: Terms such as 'enormous' should be avoided, quantify the statement or at least provide a relative comparison to anthropogenic emissions.*

Response: Agreed. "enormous" has been replaced by "large".

*10. Page 4, line 6: Should this be 'alkenes'? The reactions of NO3 with alkanes are very slow.*

Response: The review is right. This has been corrected in the revised manuscript.

*11. Page 7, lines 17-19: Are the two statements regarding isoprene rate coefficients contradictory? If*

*the results from different studies for isoprene differ by over a factor of two, how is the IUPAC recommendation precise?*

Response:  The reviewer is correct that the statements are contradictory.  The first sentence has been modified to state that BVOC with multiple $NO_3$ rate constant studies have been evaluated by IUPAC to produce recommended values.  The sentence now reads as "The most emitted/important BVOC have been subjected to several kinetic studies, using both absolute and relative methods, which are evaluated to determine recommended rate constants by IUPAC".

*12. Page 16, line 4: '. . . was investigated in the context of. . .', perhaps a brief discussion of the results of this investigation?*

Response:  We have added a second clause to this sentence to briefly describe the results of Draper et al: "This difference among monoterpenes was investigated in the context of the competition between $O_3$ and $NO_3$ oxidation (Draper et al., 2015), in which shifting from $O_3$-dominated to $NO_3$-dominated oxidation was observed to suppress SOA formation from $\alpha$-pinene, but not from $\beta$-pinene, $\Delta$-carene, or limonene."

*13. Page 16, line 8: 'at conditions' to 'under conditions'.*

Response: Modified.

*14. Page 16, line 11: '. . . fairly constant. . .', please quantify.*

Response: The sentence is modified to "Recent experiments showed that the particulate organic nitrates formed from $\beta$-pinene-$NO_3$ are resilient to photochemical aging, while those formed from $\alpha$-pinene-$NO_3$ evaporate readily".

*15. Page 16, line 27: 'reaction' to 'reactions'.*

Response: Corrected.

*16. Page 17, line 18: '. . . observed a tight correlation. . .', provide the correlation coefficient.*

Response:  The correlation coefficient was 0.99 ($r^2$ = 0.98) and has been added to the text.

*17. Page 17, line 21: Remove 'the' in '. . . the nucleation and . . .'.*

Response: Corrected.

18. *Page 20, line 18-19: 'Our present understanding...', please provide a reference.*

Response: We have added Hallquist et al (ACP, 2009) and Tsigaridis et al. (ACP, 2014) as references.

19. *Page 21, line 20: Please clarify the term 'oxidized once', what does this mean? A single oxidation step?*

Response: The phrase has been clarified to read "... undergo a single reaction with $NO_3$."

20. *Page 21, line 26: '... effectively limit rate...' to '... effectively limit the rate...'.*

Response: Corrected.

21. *Page 22, line 1: 'a phase transition' or 'phase transitions'.*

Response:  The sentence is changed to "…..owing to a phrase transition…".

22. *Page 23, line 23: lower case 'u' in undec-10-ene-1-thiol.*

Response: Corrected.

23. *Page 25, line 11: Provide a definition for 'SOx-'.*

Response: The sentence is changed to "….. $SO_x^-$ (sulfur-containing radical anions)…".

24. *Page 25, line 14: Perhaps 'in situ' in place of 'instantaneous'?*

*Response:  Phrase has been changed to "measured in-situ"*

25. *Page 25, line 30: Can the rate coefficients used in CAPRAM be linked to those recommended by IUPAC?*

Response: Unfortunately, there are no IUPAC recommendations for the aqueous phase available right now. However, this task is currently ongoing. In the future, all measured CAPRAM rate coefficients will be linked to IUPAC data. At current, CAPRAM has to be considered as a database with carefully evaluated

kinetic data for aqueous phase reactions.

*26. Page 27, line 10: 'similar rate constants'.*

Response: Corrected.

*27. Page 27, line 11: 'well correlated', provide the correlation coefficients.*

Response: The correlation of OH and $NO_3$ rate expressions and statistical analysis data calculated from kinetic data of hydroxyl radical and nitrate radical of the reactions with organic compounds for the various compound classes are given in Table SI-3 of the Supplement. We have made this clear in the revised manuscript.

*28. Page 29, line 13: How long is 'long path', the 5 km referred to above?*

Response: Quotations around long path have been removed, and the typical path length on the order of several km for the active techniques has now been specified.

*29. Page 30, line 3: Perhaps worth noting that a single wavelength measurement is more difficult to identify potential interferences?*

Response: This depends on how the measurement is done.  For $NO_3$, with chemical titration by NO combined with 662 nm detection, the measurement is quite specific.  The phrase has been modified to include "specificity can be achieved through chemical titration with NO (Brown *et al*. 2001)."

*30. Page 31, line 2: Please provide some information regarding the 'Langley-Plot method'.*

*Response:*   The Langley-Plot method is now defined in the text.  "… by the Langley-Plot method (Noxon et al., 1980), which takes advantage of the different dependence of tropospheric and stratospheric $NO_3$ slant column density on the Lunar Zenith Angle."

*31. Page 32, line 11: Note that DOAS is also an absolute technique.*

Response:  We agree that DOAS is absolute, but the paragraph in question is not about DOAS, but rather about potential wall losses in CEAS or CRDS instruments with closed sample cells.  Since DOAS does not fall in this category of instruments, it is best left out of this list.

*32. Page 33, line 17: Quantify the statement 'less good'.*

Response: We add a sentence at Page 33, Line 19: "Whereas differences between $N_2O_5$ measurements were less than 20% in the absence of aerosol, measurements differed up to factor of 2.5 for highest aerosol surface concentrations of $5x10^8$ $nm^2cm^{-3}$. Also differences between $NO_3$ measurements showed an increasing trend (up to 50%) with increasing aerosol surface concentration for some instruments."

*33. Page 40, line 6: What was the integration time associated with the 6 ppt detection limit?*

Response: Unfortunately, the integration time is not given in the Golz et al. (2001) reference.

*34. Page 44, line 1: Superscript in Rˆ2, note that 'R' has been quoted elsewhere, it would be good to maintain consistency throughout.*

Response: Superscript has been corrected, and the manuscript has been checked to ensure that $R^2$ is used for correlations throughout.

*35. Page 46, line 6: 'K-puszta' to 'Puszta'?*

Response: K-puszta is the name of the field station.

*36. Page 46, line 29: Why not maintain consistency and use either LO-OOA or SV-OOA throughout?*

Response: We note that LO-OOA has been used consistently throughout the manuscript. LO-OOA is used instead of SV-OOA as multiple studies have found that the degree of oxidation of OA does not necessarily correlate with its volatility.

*37. Page 48, line 15: Please briefly outline the potential fates of RO2 radicals and how these each impact nitrate formation and the nitrate yield.*

Response: The potential fates of the $RO_2$ radical are listed on page 47, line 28 (they are reaction with $NO_3$, $RO_2$, $HO_2$, or NO). Table 5 summarizes the nitrate yield in current models by $RO_2$ fate which is sometimes identical for all $RO_2$ fates. The yield spans from 0 to 100% (page 48, line 6-7). Observed organonitrate yields are covered earlier in the manuscript, and Table 2 summarizes observed organonitrate molar yields.

*38. Page 48, line 31: 'k' in italics.*

Response: Corrected.

*39. Page 48, line 33: Multiplication symbol in equation rather than letter 'x' (and in following*

*equations). Units for these equations should be presented in a consistent way with those elsewhere in the manuscript (e.g. in Table 1).*

Response:  Multiplication sign has been corrected, and units have been standardized to $cm^3$ $molecule^{-1}$ $s^{-1}$ throughout

*40. Page 60, line 3: 'the SOAS campaign'.*

Response: Corrected.

*41. Page 60, line 4: Remove the brackets around the reference.*

Response: Corrected.

*42. Page 66, line 12: Inconsistency between 'modeling' and 'modelling' throughout the manuscript.*

Response: This is the only place where "modelling" is used. We have changed it to "modeling" to be consistent with the rest of the manuscript.

*43. Page 76, line 22: Space in 'BVOCchemistry'.*

Response: Corrected.

*44. References: Chemical symbols – some use subscripts, others don't, while some display markup language formatting instructions (e.g. page 92, line 25 onwards).*

Response: The subscripts in the references are now properly formatted.

[revised manuscript text omitted]

| | | | |
|---|---|---|---|
| α-terpinene  | $(1.82 \pm 0.07) \times 10^{-10}$
 $(1.03 \pm 0.06) \times 10^{-10}$
 **$1.8 \times 10^{-10}$ (Δlog k : ± 0.25)** | 294
 298
 **298** | RR/(Atkinson et al., 1985)
 RR/(Berndt et al., 1996)
 **IUPAC** |
| γ-terpinene  | $(2.94 \pm 0.05) \times 10^{-11}$
 $(2.4 \pm 0.7) \times 10^{-11}$
 **$2.9 \times 10^{-11}$ (Δlog k : ± 0.12)** | 294
 298
 **298** | RR/(Atkinson et al., 1985)
 DF-LIF/(Martínez et al., 1999)
 **IUPAC** |
| Terpinolene  | $(9.67 \pm 0.51) \times 10^{-11}$
 $(5.2 \pm 0.9) \times 10^{-11}$
 $(6.12 \pm 0.52) \times 10^{-11}$
 **$9.7 \times 10^{-11}$ (Δlog k : ± 0.25)** | 295
 298
 298
 **298** | RR/(Corchnoy and Atkinson, 1990)
 DF-LIF/(Martínez et al., 1999)
 RR/(Stewart et al., 2013)
 **IUPAC** |
| Ocimene (cis, trans)  | $(2.23 \pm 0.06) \times 10^{-11}$
 **$2.2 \times 10^{-11}$ (Δlog k : ± 0.15)** | 294
 **298** | RR/(Atkinson et al., 1985)
 **IUPAC** |
| Myrcene  | $(1.06 \pm 0.02) \times 10^{-11}$
 $(1.28 \pm 0.11) \times 10^{-11}$
 $(2.2 \pm 0.2) \times 10^{-12} \exp[(523 \pm 35)/T]$
 **$1.1 \times 10^{-11}$ (Δlog k : ± 0.12)** | 294
 298
 298-433
 **298** | RR/(Atkinson et al., 1985)
 DF-LIF/(Martínez et al., 1999)
 DF-LIF/(Martínez et al., 1999)
 **IUPAC** |
| α-cedrene  | $(0.82 \pm 0.30) \times 10^{-11}$ | 296 | RR/(Shu and Atkinson, 1995) |
| α-copaene  | $(1.6 \pm 0.6) \times 10^{-11}$ | 296 | RR/(Shu and Atkinson, 1995) |
| β-caryophyllene  | $(1.9 \pm 0.8) \times 10^{-11}$ | 296 | RR/(Shu and Atkinson, 1995) |
| α-humulene  | $(3.5 \pm 1.3) \times 10^{-11}$ | 296 | RR/(Shu and Atkinson, 1995) |

| Compound | $k$ (cm³ molecule⁻¹ s⁻¹) | T (K) | Method/(Reference) |
|---|---|---|---|
| Longifolene | $(6.8 \pm 2.1) \times 10^{-13}$ | 296 | RR/(Shu and Atkinson, 1995) |
| Isolongifolene | $(3.9 \pm 1.6) \times 10^{-12}$ | 298 | RR/(Canosa-Mas et al., 1999b) |
| Alloisolongifolene | $(1.4 \pm 0.7) \times 10^{-12}$ | 298 | RR/(Canosa-Mas et al., 1999b) |
| α-neoclovene | $(8.2 \pm 4.6) \times 10^{-12}$ | 298 | RR/(Canosa-Mas et al., 1999b) |
| 2-methyl-3-buten-2-ol | $4.6 \times 10^{-14} \exp[-(400 \pm 35)/T]$ | 267-400 | F-A/(Rudich et al., 1996) |
|  | $(1.21 \pm 0.09) \times 10^{-14}$ | 298 | F-A/(Rudich et al., 1996) |
|  | $(2.1 \pm 0.3) \times 10^{-14}$ | 294 | DF-A/(Hallquist et al., 1996) |
|  | $(1.55 \pm 0.55) \times 10^{-14}$ | 294 | RR/(Hallquist et al., 1996) |
|  | $(8.7 \pm 3.0) \times 10^{-15}$ | 298 | RR/(Fantechi et al., 1998b) |
|  | $(1.0 \pm 0.2) \times 10^{-14}$ | 297 | RR/(Noda et al., 2002) |
|  | $(1.1 \pm 0.1) \times 10^{-14}$ | 297 | RR/(Noda et al., 2002) |
|  | **$1.2 \times 10^{-14}$ (Δlog k : ± 0.2)** | **298** | **IUPAC** |
| 3-methyl-2-buten-1-ol | $(1.0 \pm 0.1) \times 10^{-12}$ | 297 | RR/(Noda et al., 2002) |
| 3-methyl-3-buten-1-ol | $(2.7 \pm 0.2) \times 10^{-13}$ | 297 | RR/(Noda et al., 2002) |
| cis-3-hexen-1-ol | $(2.72 \pm 0.83) \times 10^{-13}$ | 296 | RR/(Atkinson et al., 1995) |
|  | $(2.67 \pm 0.42) \times 10^{-13}$ | 298 | DF-CEAS/(Pfrang et al., 2006) |
| trans-3-hexen-1-ol | $(4.43 \pm 0.91) \times 10^{-13}$ | 298 | DF-CEAS/(Pfrang et al., 2006) |

| | | | |
|---|---|---|---|
| *cis*-4-hexen-1-ol | $(2.93 \pm 0.48) \times 10^{-13}$ | 298 | DF-CEAS/(Pfrang et al., 2006) |
| *trans*-2-hexen-1-ol | $(1.30 \pm 0.24) \times 10^{-13}$ | 298 | DF-CEAS/(Pfrang et al., 2006) |
| *cis*-2-hexen-1-ol | $(1.56 \pm 0.24) \times 10^{-13}$ | 298 | DF-CEAS/(Pfrang et al., 2006) |
| *trans*-2-hexenal | $(1.21 \pm 0.44) \times 10^{-14}$
$(1.36 \pm 0.29) \times 10^{-14}$
$(4.7 \pm 1.5) \times 10^{-15}$ | 296
295
294 | RR/(Atkinson et al., 1995)
RR/(Zhao et al., 2011b)
AR/(Kerdouci et al., 2012) |
| 4-methylenehex-5-enal | $(4.75 \pm 0.35) \times 10^{-13}$ | 296 | RR/(Baker et al., 2004) |
| (3Z)-4-methylhexa-3,5-dienal | $(2.17 \pm 0.30) \times 10^{-12}$ | 296 | RR/(Baker et al., 2004) |
| (3E)-4-methylhexa-3,5-dienal | $(1.75 \pm 0.27) \times 10^{-12}$ | 296 | RR/(Baker et al., 2004) |
| 4-methylcyclohex-3-en-1-one | $(1.81 \pm 0.35) \times 10^{-12}$ | 296 | RR/(Baker et al., 2004) |
| *cis*-3-hexenyl acetate | $(2.46 \pm 0.75) \times 10^{-13}$ | 296 | RR/(Atkinson et al., 1995) |
| methyl vinyl ketone | $< 1.2 \times 10^{-16}$
$< 6 \times 10^{-16}$
$(3.2 \pm 0.6) \times 10^{-16}$
$(5.0 \pm 1.2) \times 10^{-16}$
**$< 6 \times 10^{-16}$** | 298
296
296
296
**298** | F-A/(Rudich et al., 1996)
DF- RR/(Kwok et al., 1996)
LIF/(Canosa-Mas et al., 1999a)
RR/(Canosa-Mas et al., 1999a)

[revised manuscript text omitted]